# Lending Interaction Wings to Recommender Systems with Conversational Agents

**Jiarui Jin**[1,†]**, Xianyu Chen**[1]**, Fanghua Ye**[2]**, Mengyue Yang**[2]**, Yue Feng**[2]**,**
**Weinan Zhang**[1,‡]**, Yong Yu**[1]**, Jun Wang**[2,‡]
[1]Shanghai Jiao Tong University, [2]University College London
{jinjiarui97,wnzhang,xianyujun,yyu}@sjtu.edu.cn
{fanghua.ye.19,mengyue.yang.20,yue.feng.20,jun.wang}@ucl.ac.uk

## Abstract

Recommender systems trained on *offline* historical user behaviors are embracing conversational techniques to *online* query user preference. Unlike prior conversational recommendation approaches that systemically combine conversational and recommender parts through a reinforcement learning framework, we propose `CORE`, a new *offline-training and online-checking* paradigm that bridges a COnversational agent and REcommender systems via a unified *uncertainty minimization* framework. It can benefit *any* recommendation platform in a plug-and-play style. Here, `CORE` treats a recommender system as an *offline relevance score estimator* to produce an estimated relevance score for each item; while a conversational agent is regarded as an *online relevance score checker* to check these estimated scores in each session. We define *uncertainty* as the summation of *unchecked* relevance scores. In this regard, the conversational agent acts to minimize uncertainty via querying either *attributes* or *items*. Based on the uncertainty minimization framework, we derive the *expected certainty gain* of querying each attribute and item, and develop a novel *online decision tree* algorithm to decide what to query at each turn. We reveal that `CORE` can be extended to query attribute values, and we establish a new Human-AI recommendation simulator supporting both open questions of querying attributes and closed questions of querying attribute values. Experimental results on 8 industrial datasets show that `CORE` could be seamlessly employed on 9 popular recommendation approaches, and can consistently bring significant improvements, compared against either recently proposed reinforcement learning-based or classical statistical methods, in both hot-start and cold-start recommendation settings. We further demonstrate that our conversational agent could communicate as a human if empowered by a pre-trained large language model, e.g., `gpt-3.5-turbo`.

## 1 Introduction

Recommender systems are powerful tools to facilitate users' information seeking [28, 38, 4, 20, 22, 21, 52, 51]; however, most prior works solely leverage offline historical data to build a recommender system. The inherent limitation of these recommendation approaches lies in their offline focus on users' historical interests, which would not always align with users' present needs. As intelligent conversational assistants (a.k.a., chat-bots) such as ChatGPT and Amazon Alexa, have entered the daily life of users, these conversational techniques bring an unprecedented opportunity to online obtain users' current preferences via conversations. This possibility has been envisioned as conversational

---

[†]Work done during Jiarui Jin's visit at University College London.
[‡]Correspondence: Weinan Zhang, Jun Wang.

37th Conference on Neural Information Processing Systems (NeurIPS 2023).

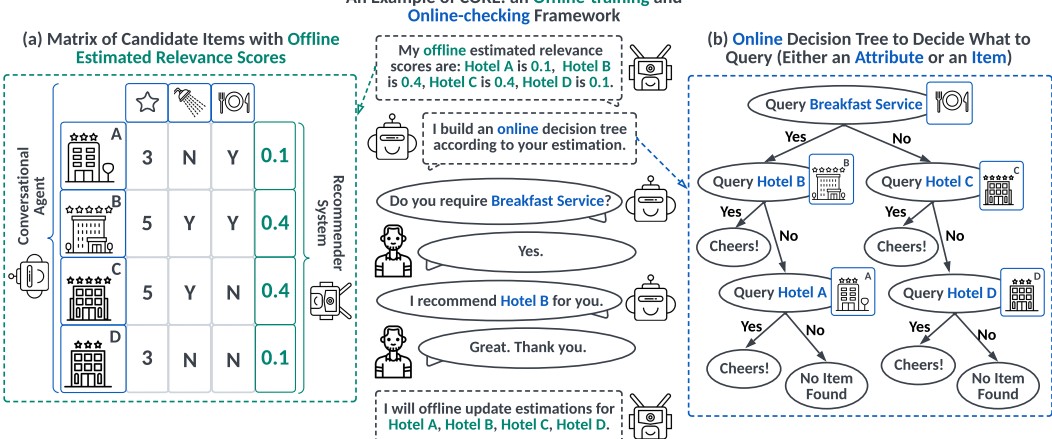

Figure 1: An illustrated example of CORE, an *offline-training and online-checking* framework, where a recommender system operates as *an offline relevance score estimator* (colored in green), while a conversational agent acts as *an online relevance score checker* (colored in blue). Concretely, given a matrix of candidate items, as shown in (a), the recommender system could *offline* assign an estimated relevance score to each item, and then the conversational agent would *online* check these scores by querying either items or attributes, depicted in (b).

recommender systems and has inspired a series of conversational recommendation methods [26, 30, 44]. Unfortunately, all of these approaches try to model the interactions between users and systems using a reinforcement learning-based framework, which inevitably suffers from data insufficiency and deployment difficulty, because most recommendation platforms are based on supervised learning.

In this paper, we propose CORE that can bridge a COnversational agent and REcommender systems in a plug-and-play style. In our setting, a conversational agent can choose either to query (a.k.a., to recommend) an item (e.g., Hotel A) or to query an attribute (e.g., Hotel Level), and the user should provide their corresponding preference. Here, the goal of the conversational agent is to find (a.k.a., to query) an item that satisfies the user, with a minimal number of interactions.

We formulate the cooperation between a conversational agent and a recommender system into a novel *offline-training and online-checking* framework. Specifically, CORE treats a recommender system as an *offline relevance score estimator* that offline assigns a relevance score to each item, while a conversational agent is regarded as an *online relevance score checker* that online checks whether these estimated relevance scores could reflect the relevance between items and the user's current needs. Here, "checked items" means those items, we can certainly say that they can not satisfy the user according to already queried items and attributes. We introduce a new *uncertainty* metric defined as the summation of estimated relevance scores of those unchecked items. Then, the goal of our conversational agent can be formulated as minimizing the uncertainty via querying items or attributes during the interactions. To this end, we derive *expected certainty gain* to measure the expectation of uncertainty reduction by querying each item and attribute. Then, during each interaction, our conversational agent selects an item or an attribute with the maximum certainty gain, resulting in *an online decision tree* algorithm. We exemplify the above process in Figure 1.

Notice that users usually do not hold a clear picture of their preferences on some attributes (i.e., attribute IDs), e.g., what Hotel Level they need, instead, they could have a clear preference on a specific value of an attribute (i.e., attribute value), e.g., Hotel Level=5 is too expensive for a student user. Also, asking an open question of querying attributes could result in an unexpected answer, e.g., a user answers 3.5 to Hotel Level. In this regard, querying attribute values leading to closed questions (i.e., Yes or No questions) could be a better choice. We reveal that CORE could be directly applied to the above querying strategies. We also develop a new Human-AI recommendation simulator that supports both querying attributes and attribute values.

In practice, we extend CORE to handle continuous attributes and to consider the dependence among attributes. Moreover, we demonstrate that our conversational agent could straightforwardly be empowered by a pre-trained language model, e.g., gpt-3.5-turbo, to communicate as a human. Note that CORE poses no constraint on recommender systems, only requiring the estimated relevance scores. Therefore, CORE can be seamlessly applied to *any* recommendation platform. We conduct experiments on 8 industrial datasets (including both tabular data, sequential behavioral data and graph-structured

data) with 9 popular recommendation approaches (e.g., DeepFM [20], DIN [52]). Experimental results show that `CORE` can bring significant improvements in both hot-start recommendation (i.e., the recommender system is offline trained) and cold-start recommendation (i.e., the recommender system is not trained) settings. We compare `CORE` against recently proposed reinforcement learning based methods and classical statistical methods, and `CORE` could consistently show better performance.

## 2 Bridging Conversational Agents and Recommender Systems

### 2.1 Problem Formulation

Let $\mathcal{U}$ denote a set of users, $\mathcal{V} = \{v_1, \ldots, v_M\}$ be a set of $M$ items, $\mathcal{X} = \{x_1, \ldots, x_N\}$ be a set of $N$ attributes (a.k.a., features) of items. We consider a recommender system as a mapping function, denoted as $\Psi_{\texttt{RE}} : \mathcal{U} \times \mathcal{V} \to \mathbb{R}$ that assigns an estimated relevance score to each item regarding a user. Then, during each online session, a conversational agent also can be formulated as a mapping function, denoted as $\Psi_{\texttt{CO}} : \mathcal{U} \times \mathcal{A} \to \mathbb{R}$ that chooses either an item or an attribute to query user, and $\mathcal{A} = \mathcal{V} \cup \mathcal{X}$ denotes the action space of the conversational agent. For convenience, we focus on the *most* favorite item of the given user in each session. In other words, our goal is to identify the item that will receive the *first* click when all items are equally presented. We call this item as *target item*.

For this purpose, $\Psi_{\texttt{RE}}(\cdot)$ acts as an *offline estimator* to produce an estimated relevance distribution of items regarding each user through offline training on previous behavioral data; while $\Psi_{\texttt{CO}}(\cdot)$ operates as an *online checker* to check whether these estimated scores fit the user's current needs through online interactions. Here, "checked items" denote those items that can not be the target item according to queried items and attributes. For example, as Figure 1 illustrates, after querying `Breakfast Service`, we have items `Hotel C` and `Hotel D` checked. We introduce *uncertainty* as the summation of estimated relevance scores of unchecked items. Formally, we have:

**Definition 1** (**Uncertainty and Certainty Gain**). *For the $k$-th turn, we define uncertainty, denoted as $\texttt{U}_k$, to measure how many estimated relevance scores are still unchecked, i.e.,*

$$\texttt{U}_k := \texttt{SUM}(\{\Psi_{\texttt{RE}}(v_m) | v_m \in \mathcal{V}_k\}), \tag{1}$$

*where $\Psi_{\texttt{RE}}(v_m)$[1] outputs the estimated relevance score for item $v_m$, $\mathcal{V}_k$ is the set of all the unchecked items after $k$ interactions, and $\mathcal{V}_k$ is initialized as $\mathcal{V}_0 = \mathcal{V}$. Then, the certainty gain of $k$-th interaction is defined as $\Delta \texttt{U}_k := \texttt{U}_{k-1} - \texttt{U}_k$, i.e., how many relevance scores are checked at the $k$-th turn. Since our goal is to find the target item, if $\Psi_{\texttt{CO}}(\cdot)$ successfully finds one at the $k$-th turn, then we set all the items checked, namely $\texttt{U}_k = 0$.*

In this regard, our conversational agent is to minimize $\texttt{U}_k$ at each $k$-th turn via removing those checked items from $\mathcal{V}_k$. Considering that online updating $\Psi_{\texttt{RE}}(\cdot)$ is infeasible in practice due to the high latency and computation costs, the objective of $\Psi_{\texttt{CO}}(\cdot)$ can be expressed as:

$$\min_{\Psi_{\texttt{RE}}^*} K, \text{ s.t., } \texttt{U}_K = 0, \tag{2}$$

where $K$ is the number of turns, and $\Psi_{\texttt{RE}}^*$ means that the recommender system is frozen. To this end, the design of our conversational agent could be organized as an uncertainty minimization problem.

### 2.2 Comparisons to Previous Work

Bridging conversational techniques and recommender systems has become an appealing solution to model the dynamic preference and weak explainability problems in recommendation task [15, 26], where the core sub-task is to dynamically select attributes to query and make recommendations upon the corresponding answers. Along this line, the main branch of previous studies is to combine the conversational models and the recommendation models from a systematic perspective. Namely conversational and recommender models are treated and learned as two individual modules [30, 2, 44, 48] in the system. The system is developed from a single-turn conversational recommender system [5, 6, 47] to a multiple-turn one [49]. To decide when to query attributes and when to make recommendations (i.e., query items), recent papers [30, 31, 44, 30] develop reinforcement

---

[1]In this paper, as our conversational agent only faces one user $u \in \mathcal{U}$ in each session, we omit the input $u$ in mapping functions $\Psi_{\cdot}(\cdot)$s for simplicity.

learning-based solutions, which are innately suffering from insufficient usage of labeled data and high complexity costs of deployment. However, reinforcement learning approaches often demand a substantial number of training samples, a condition referred to as data insufficiency, and require operating within a relatively compact action space. Instead, our conversational agent can be regarded as a generalist agent that can query either items or attributes. In addition, our querying strategy is derived based on the uncertainty minimization framework, which only requires estimated relevance scores from the recommender system. Hence, CORE can be straightforwardly applied to *any* supervised learning-based recommendation platform, in a plug-and-play way, free of meticulous reward function design in reinforcement learning methods. CORE is more friendly to open-world scenarios especially when training data is limited.

We present the connections to other previous work (e.g., decision tree algorithms) in Appendix A5.

## 3 Making the Conversational Agent a Good Uncertainty Optimizer

### 3.1 Building an Online Decision Tree

As described in Section 2.1, the aim of our conversational agent is to effectively reduce uncertainty via querying either items or attributes. The core challenge is how to decide which item or attribute to query. To this end, we begin by introducing *expected certainty gain* to measure the expectation of how much uncertainty could be eliminated by querying each item and each attribute. Then, we can choose an item or an attribute with the maximum expected certainty gain to query.

Formally, let $\mathcal{X}_k$ denote the set of unchecked attributes after $k$ interactions. Then, for each $k$-th turn, we define $a_{\text{query}}$ as an item or an attribute to query, which is computed following:

$$a_{\text{query}} = \underset{a \in \mathcal{V}_{k-1} \cup \mathcal{X}_{k-1}}{\arg\max} \ \Psi_{\text{CG}}(\text{query}(a)), \tag{3}$$

where $\Psi_{\text{CG}}(\cdot)$ denotes the *expected certainty gain* of querying $a$, and $a$ can be either an unchecked item (from $\mathcal{V}_{k-1}$) or an unchecked attribute (from $\mathcal{X}_{k-1}$).

$\Psi_{\text{CG}}(\cdot)$ **for Querying an Item.** We first consider the case where $a \in \mathcal{V}_{k-1}$. Let $v^*$ denote the target item in the session. Since we only need to find *one* target item, therefore, if $a \in \mathcal{V}_{k-1}$, we can derive:

$$\begin{aligned}
\Psi_{\text{CG}}(\text{query}(a)) &= v^*) \cdot \text{Pr}(a = v^*) + \Psi_{\text{CG}}(a \neq v^*) \cdot \text{Pr}(a \neq v^*) \\
&= \Big( \sum_{v_m \in \mathcal{V}_{k-1}} \Psi_{\text{RE}}(v_m) \Big) \cdot \text{Pr}(a = v^*) + \Psi_{\text{RE}}(a) \cdot \text{Pr}(a \neq v^*),
\end{aligned} \tag{4}$$

where $a = v^*$ and $a \neq v^*$ denote that queried $a$ is the target item and not. If $a = v^*$, the session is done, and therefore, the certainty gain (i.e., $\Psi_{\text{CG}}(a = v^*)$) is the summation of all the relevance scores in $\mathcal{V}_{k-1}$. Otherwise, only $a$ is checked, and the certainty gain (i.e., $\Psi_{\text{CG}}(a \neq v^*)$) is the relevance score of $a$, and we have $\mathcal{V}_k = \mathcal{V}_{k-1} \backslash \{a\}$ and $\mathcal{X}_k = \mathcal{X}_{k-1}$.

Considering that $a$ being the target item means $a$ being the most relevant item, we leverage the user's previous behaviors to estimate the user's current preference. With relevance scores estimated by $\Psi_{\text{RE}}(\cdot)$, we estimate $\text{Pr}(a = v^*)$ as:

$$\text{Pr}(a = v^*) = \frac{\Psi_{\text{RE}}(a)}{\text{SUM}(\{\Psi_{\text{RE}}(v_m) | v_m \in \mathcal{V}_{k-1}\})}, \tag{5}$$

and $\text{Pr}(a \neq v^*) = 1 - \text{Pr}(a \neq v^*)$.

$\Psi_{\text{CG}}(\cdot)$ **for Querying an Attribute.** We then consider the case where $a \in \mathcal{X}_{k-1}$. For each queried attribute $a$, let $\mathcal{W}_a$ denote the set of all the candidate attribute values, and let $w_a^* \in \mathcal{W}_a$ denote the user preference on $a$, e.g., $a$ is Hotel Level, $w_a^*$ is 3. Then, if $a \in \mathcal{X}_{k-1}$, we have:

$$\Psi_{\text{CG}}(\text{query}(a)) = \sum_{w_a \in \mathcal{W}_a} \Big( \Psi_{\text{CG}}(w_a = w_a^*) \cdot \text{Pr}(w_a = w_a^*) \Big), \tag{6}$$

where $w_a = w_a^*$ means that when querying $a$, the user's answer (represented by $w_a^*$) is $w_a$, $\Psi_{\text{CG}}(w_a = w_a^*)$ is the certainty gain when $w_a = w_a^*$ happens, and $\text{Pr}(w_a = w_a^*)$ is the probability of $w_a = w_a^*$ occurring. If $w_a = w_a^*$ holds, then all the unchecked items whose value of $a$ is not equal to $w_a$ should be removed from $\mathcal{V}_{k-1}$, as they are certainly not satisfying the user's needs.

Formally, let $\mathcal{V}_{a_{\text{value}}=w_a}$ denote the set of all the items whose value of $a$ is equal to $w_a$, and let $\mathcal{V}_{a_{\text{value}}\neq w_a}$ denote the set of rest items. Then, $\Psi_{\text{CG}}(w_a = w_a^*)$ can be computed as:

$$\Psi_{\text{CG}}(w_a = w_a^*) = \text{SUM}(\{\Psi_{\text{RE}}(v_m)|v_m \in \mathcal{V}_{k-1} \cap \mathcal{V}_{a_{\text{value}}\neq w_a}\}), \tag{7}$$

which indicates that the certainty gain, when $w_a$ is the user's answer, is the summation of relevance scores of those items not matching the user preference.

To finish $\Psi_{\text{CG}}(\text{query}(a))$, we also need to estimate $\text{Pr}(w_a = w_a^*)$. To estimate the user preference on attribute $a$, we leverage the estimated relevance scores given by $\Psi_{\text{RE}}(\cdot)$ as:

$$\text{Pr}(w_a = w_a^*) = \frac{\text{SUM}(\{\Psi_{\text{RE}}(v_m)|v_m \in \mathcal{V}_{k-1} \cap \mathcal{V}_{a_{\text{value}}=w_a}\})}{\text{SUM}(\{\Psi_{\text{RE}}(v_m)|v_m \in \mathcal{V}_{k-1}\})}. \tag{8}$$

In this case, we remove $\mathcal{V}_{k-1} \cap \mathcal{V}_{a_{\text{value}}\neq w_a^*}$ from $\mathcal{V}_{k-1}$, namely we have $\mathcal{V}_k = \mathcal{V}_{k-1} \setminus \mathcal{V}_{a_{\text{value}}\neq w_a^*}$. As attribute $a$ is checked, we have $\mathcal{X}_k = \mathcal{X}_{k-1}\setminus\{a\}$. Here, $w_a^*$ is provided by the user after querying $a$.

By combining Eqs. (4), (6), and (7), we can derive a completed form of $\Psi_{\text{CG}}(\text{query}(a))$ for $a \in \mathcal{V}_{k-1} \cup \mathcal{X}_{k-1}$ (See Appendix A1.1 for details). Then, at each $k$-th turn, we can always follow Eq. (3) to obtain the next query $a_{\text{query}}$. As depicted in Figure 1(b), the above process results in an online decision tree, where the nodes in each layer are items and attributes to query, and the depth of the tree is the number of turns (see Appendix A4.3 for visualization of a real-world case).

## 3.2 From Querying Attributes to Querying Attribute Values

We note that the online decision tree introduced above is a general framework; while applying it to real-world scenarios, there should be some specific designs.

$\Psi_{\text{CG}}(\cdot)$ **for Querying an Attribute Value.** One implicit assumption in the above online decision tree is that the user's preference on queried attribute $a$ always falls into the set of attribute values, namely $w_a^* \in \mathcal{W}_a$ holds. However, it can not always hold, due to (i) a user would not have a clear picture of an attribute, (ii) a user's answer would be different from all the candidate attribute values, e.g., $a$ is Hotel Level, $w_a^* = 3.5$, and $\mathcal{W}_a = \{3, 5\}$, as shown in Figure 1(a). In these cases, querying attributes would not be a good choice. Hence, we propose to query attribute values instead of attribute IDs, because (i) a user is likely to hold a clear preference for a specific value of an attribute, e.g., a user would not know an actual Hotel Level of her favoring hotels, but she clearly knows she can not afford a hotel with Hotel Level=5, and (ii) since querying attribute values leads to closed questions instead of open questions, a user only needs to answer Yes or No, therefore, avoiding the user's answer to be out of the scope of all the candidate attribute values.

Formally, in this case, $\mathcal{A} = \mathcal{W}_x \times \mathcal{X}_{k-1}$ which indicates we need to choose a value $w_x \in \mathcal{W}_x$ where $x \in \mathcal{X}_{k-1}$. In light of this, we compute the expected certainty gain of querying attribute value $w_x$ as:

$$\Psi_{\text{CG}}(\text{query}(x) = w_x) = \Psi_{\text{CG}}(w_x = w_x^*) \cdot \text{Pr}(w_x = w_x^*) + \Psi_{\text{CG}}(w_x \neq w_x^*) \cdot \text{Pr}(w_x \neq w_x^*), \tag{9}$$

where $w_x^* \in \mathcal{W}_x$ denotes the user preference on attribute $x$. Here, different from querying attributes, a user would only respond with Yes (i.e., $w_x = w_x^*$) or No (i.e., $w_x \neq w_x^*$). Therefore, we only need to estimate the certainty gain for the above two cases. $\Psi_{\text{CG}}(w_x = w_x^*)$ can be computed following Eq. (7) and $\Psi_{\text{CG}}(w_x \neq w_x^*)$ can be calculated by replacing $\mathcal{V}_{x_{\text{value}}\neq w_x}$ with $\mathcal{V}_{x_{\text{value}}=w_x}$. $\text{Pr}(w_x = w_x^*)$ is estimated in Eq. (8) and $\text{Pr}(w_x \neq w_x^*) = 1 - \text{Pr}(w_x = w_x^*)$. In this case, if all the values of $x$ have been checked, we have $\mathcal{X}_k = \mathcal{X}_{k-1}\setminus\{x\}$; otherwise, $\mathcal{X}_k = \mathcal{X}_{k-1}$; and $\mathcal{V}_k = \mathcal{V}_{k-1} \setminus \mathcal{V}_{x_{\text{value}}\neq w_x}$ if receiving Yes from the user, $\mathcal{V}_k = \mathcal{V}_{k-1} \setminus \mathcal{V}_{x_{\text{value}}=w_x}$, otherwise.

We reveal the connection between querying attributes (i.e., querying attribute IDs) and querying attribute values in the following proposition.

**Proposition 1.** *For any attribute $x \in \mathcal{X}_{k-1}$, $\Psi_{\text{CG}}(\text{query}(x)) \geq \Psi_{\text{CG}}(\text{query}(x) = w_x)$ holds for all the possible $w_x \in \mathcal{W}_x$.*

This proposition shows that if users could give a clear preference for the queried attribute and their preferred attribute value is one of the candidate attribute values, then querying attributes would be an equivalent or a better choice than querying attribute values. In other words, querying attributes and querying attribute values can not operate on the same attributes (otherwise, $\Psi_{\text{CG}}(\cdot)$ would always choose to query attributes). Therefore, we can combine querying items and querying attribute values

**Algorithm 1** CORE for Querying Items and Attributes

---

**Input:** A recommender system $\Psi_{\mathtt{RE}}(\cdot)$, an item set $\mathcal{V}$, an attribute set $\mathcal{X}$, an offline dataset $\mathcal{D}$.
**Output:** Updated recommender system $\Psi_{\mathtt{RE}}(\cdot)$, up-to-date dataset $\mathcal{D}$.

 1: Train $\Psi_{\mathtt{RE}}(\cdot)$ on $\mathcal{D}$.                                      ▷ Offline-Training
 2: **for** each session (i.e., the given user) **do**
 3:     Initialize $k = 1$ and $\mathcal{V}_0 = \mathcal{V}$, $\mathcal{X}_0 = \mathcal{X}$.
 4:     **repeat**
 5:         Compute $a_{\mathtt{query}}$ following Eq. (3) for querying items and attributes or following Eq. (10) for querying items and attribute values.                                    ▷ Online-Checking
 6:         Query $a_{\mathtt{query}}$ to the user and receive the answer.                  ▷ Online-Checking
 7:         Generate $\mathcal{V}_k$ and $\mathcal{X}_k$ from $\mathcal{V}_{k-1}$ and $\mathcal{X}_{k-1}$.                        ▷ Online-Checking
 8:         Go to next turn: $k \leftarrow k + 1$.
 9:     **until** Querying the target item or $k > K_{\mathtt{MAX}}$ where $K_{\mathtt{MAX}}$ is the maximal number of turns.
10:     Collect session data and add to $\mathcal{D}$.
11: **end for**
12: Update $\Psi_{\mathtt{RE}}(\cdot)$ using data in $\mathcal{D}$.                           ▷ Offline-Training

---

by setting the action space to $\mathcal{A} = \mathcal{W}_x \times \mathcal{X}_{k-1} \cup \mathcal{V}_{k-1}$. Then, we can re-formulate Eq. (3) as:

$$a_{\mathtt{query}} = \underset{a \in \{w_x, v\}}{\arg \max} \Big( \max_{w_x \in \mathcal{W}_x \text{ where } x \in \mathcal{X}_{k-1}} \Psi_{\mathtt{CG}}(\mathtt{query}(x) = w_x), \max_{v \in \mathcal{V}_{k-1}} \Psi_{\mathtt{CG}}(\mathtt{query}(v)) \Big). \quad (10)$$

In the context of querying attribute values, we further reveal what kind of attribute value is an ideal one in the following theorem.

**Proposition 2.** *In the context of querying attribute values, an ideal choice is always the one that can partition all the unchecked relevance scores into two equal parts (i.e., the ideal $w_x \in \mathcal{W}_x, x \in \mathcal{X}_{k-1}$ is the one that makes $\Psi_{\mathtt{CG}}(w_x = w_x^*) = \mathtt{SUM}(\{\Psi_{\mathtt{RE}}(v_m) | v_m \in \mathcal{V}_{k-1}\})/2$ hold), if it is achievable. And the certainty gain in this case is $\Psi_{\mathtt{CG}}(\mathtt{query}(x) = w_x) = \mathtt{SUM}(\{\Psi_{\mathtt{RE}}(v_m) | v_m \in \mathcal{V}_{k-1}\})/2$.*

Then, we consider the bound of the expected number of turns. To get rid of the impact of $\Psi_{\mathtt{RE}}(\cdot)$, we introduce a cold-start setting [43], where $\Psi_{\mathtt{RE}}(\cdot)$ knows nothing about the user, and equally assigns relevance scores to all $M$ items, resulting in $\Psi_{\mathtt{RE}}(v_m) = 1/M$ holds for any $v_m \in \mathcal{V}$.

**Lemma 1.** *In the context of querying attribute values, suppose that $\Psi_{\mathtt{RE}}(v_m) = 1/M$ holds for any $v_m \in \mathcal{V}$, then the expected number of turns (denoted as $\widehat{K}$) is bounded by $\log_2^{M+1} \leq \widehat{K} \leq (M+1)/2$.*

Here, the good case lies in that our conversational agent is capable of finding an attribute value to form an ideal partition at each turn, while the bad case appears when we can only check one item at each turn. We provide detailed proofs of Propositions 1 and 2, and Lemma 1 in Appendix A1.2.

$\Psi_{\mathtt{CG}}(\cdot)$ **for Querying Attributes in Large Discrete or Continuous Space.** All the above querying strategies are designed in the context that for each attribute, the range of its candidate values is a "small" discrete space, namely $|\mathcal{W}_x| \ll |\mathcal{V}_{k-1}|$ where $x \in \mathcal{X}_{k-1}$. When it comes to cases where $\mathcal{W}_x$ is a large discrete space or a continuous space, then either querying attribute $x$ or any attribute value $w_x \in \mathcal{W}_x$ would not be a good choice. For example, let $x$ be Hotel Price, then when querying $x$, the user would not respond with an accurate value, and querying $x$=one possible value could be ineffective. To address this issue, we propose to generate a new attribute value $w_x$ and query whether the user's preference is not smaller than it or not. Formally, we have:

$$\Psi_{\mathtt{CG}}(\mathtt{query}(x) \geq w_x) = \Psi_{\mathtt{CG}}(w_x \geq w_x^*) \cdot \mathtt{Pr}(w_x \geq w_x^*) + \Psi_{\mathtt{CG}}(w_x < w_x^*) \cdot \mathtt{Pr}(w_x < w_x^*), \quad (11)$$

where $x \in \mathcal{X}_{k-1}$ and $w_x$ can be either in or out of $\mathcal{W}_x$. Compared to querying attribute values (i.e., Eq. (9)), the new action space is $\mathcal{A} = \mathbb{R} \times \mathcal{X}_{k-1}$. Notice that Proposition 2 is also suitable for this case (see detailed description in Appendix A2.1), where the best partition is to divide the estimated relevance scores into two equal parts. Therefore, we produce $w_x$ by averaging all the candidate attribute values weighted by the corresponding relevance scores. Formally, for each $x \in \mathcal{X}_{k-1}$, we compute $w_x$ as:

$$w_x = \mathtt{AVERAGE}(\{\Psi_{\mathtt{RE}}(v_m) \cdot w_{v_m} | v_m \in \mathcal{V}_{k-1}\}), \quad (12)$$

where $w_{v_m}$ is the value of attribute $x$ in item $v_m$, e.g., in Figure 1(a), let $a$ be Hotel Level, and $v_m$ be Hotel A, then $w_{v_m} = 3$.

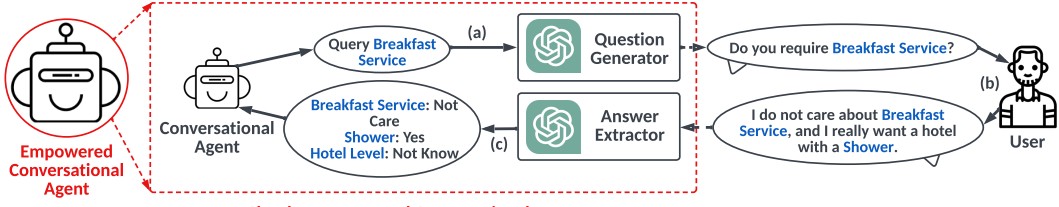

Figure 2: We illustrate the empowerment of our conversational agent through the utilization of a pre-trained chat-bot. In this context, the red box signifies the chat-bot-empowered conversational agent. To accomplish this, we input the output queries produced by the original conversational agent, such as `Breakfast Service` into the question generator, as depicted in (a). In (b), a user is expected to input the generated question in a free-text format and provide the corresponding answer in a free-text format. Subsequently, in (c), the answer extractor extracts key information from the user's response and provides it to the original conversational agent.

In this case, $\mathcal{X}_k = \mathcal{X}_{k-1}$, and $\mathcal{V}_k = \mathcal{V}_{k-1} \setminus \mathcal{V}_{x_{\texttt{value}} < w_x}$ if receiving `Yes` from the user when querying whether user preference is not smaller than $w_x$, $\mathcal{V}_k = \mathcal{V}_{k-1} \setminus \mathcal{V}_{x_{\texttt{value}} \geq w_x}$ otherwise. $\mathcal{V}_{x_{\texttt{value}} < w_x}$ is the set of all the items whose value of $x$ is smaller than $w_x$ and $\mathcal{V}_{x_{\texttt{value}} \geq w_x}$ is the set of the rest items.

### 3.3 Plugging the Conversational Agent into Recommender Systems

**Overall Algorithm.** We begin by summarizing `CORE` for querying items and attributes or querying items and attribute values in Algorithm 1. From the algorithm, we can clearly see that our $\Psi_{\texttt{CO}}(\cdot)$ puts no constraints on $\Psi_{\texttt{RE}}(\cdot)$ and only requires the estimated relevance scores from $\Psi_{\texttt{RE}}(\cdot)$, therefore, `CORE` can be seamlessly integrated into *any* recommendation platform. We note that *in a conversational agent, querying attributes and querying attribute values can be compatible, but can not simultaneously operate on the same attribute, due to Proposition 1.* See Appendix A2.3 for a detailed discussion.

**Making $\Psi_{\texttt{CG}}(\cdot)$ Consider Dependence among Attributes.** We notice that the above formulations of either querying attributes or querying attribute values, does not consider the dependence among attributes (e.g., as Figure 1(a) shows, attribute `Hotel Level` can largely determine attribute `Shower Service`). To address this issue, we take $\Psi_{\texttt{CG}}(\cdot)$ in Eq. (6) as an example (see detailed descriptions of the other $\Psi_{\texttt{CG}}(\cdot)$s in Appendix A2.2), and re-formulate it as:

$$\Psi_{\texttt{CG}}^{\texttt{D}}(\texttt{query}(a)) = \sum_{a' \in \mathcal{X}_{k-1}} \Big( \Psi_{\texttt{CG}}(\texttt{query}(a')) \cdot \texttt{Pr}(\texttt{query}(a')|\texttt{query}(a)) \Big), \qquad (13)$$

where $a \in \mathcal{X}_{k-1}$, and $\texttt{Pr}(\texttt{query}(a')|\texttt{query}(a))$ measures the probability of the user preference on $a$ determining the user preference on $a'$. Compared to $\Psi_{\texttt{CG}}(\texttt{query}(a))$, $\Psi_{\texttt{CG}}^{\texttt{D}}(\texttt{query}(a))$ further considers the impact of querying attribute $a$ on other attributes. To estimate $\texttt{Pr}(\texttt{query}(a')|\texttt{query}(a))$, we develop two solutions. We notice that many widely adopted recommendation approaches are developed on factorization machine (FM) [38], e.g., DeepFM [20]. Therefore, when applying these FM-based recommendation approaches, one approach is to directly adopt their learned weight for each pair of attributes $(a, a')$ as the estimation of $\texttt{Pr}(\texttt{query}(a')|\texttt{query}(a))$. When applying `CORE` to any other recommendation method (e.g., DIN [52]), we develop a statistics-based approach that does estimations by computing this conditional probability $\Psi_{\texttt{CG}}^{\texttt{D}}(\texttt{query}(a))$ based on the given candidate items. We leave the detailed computations of $\Psi_{\texttt{CG}}^{\texttt{D}}(\texttt{query}(a))$ in both ways in Appendix A2.2.

**Empowering $\Psi_{\texttt{CO}}(\cdot)$ to Communicate with Humans.** When applying `CORE` into real-world scenarios, users may provide a `Not Care` attitude regarding the queried attributes or queried attribute values. In these cases, we generate $\mathcal{V}_k$ and $\mathcal{X}_k$ by $\mathcal{V}_k = \mathcal{V}_{k-1}$ and $\mathcal{X}_k = \mathcal{X}_{k-1} \setminus \{a\}$, because querying $a$ is non-informative. To capture the user's different attitudes on queried items and attributes or attribute values, we can incorporate a pre-trained large language model (LLM) (e.g., `gpt-3.5-turbo`) in $\Psi_{\texttt{CO}}(\cdot)$. As our online-checking part does not require training, simply plugging an LLM would not cause the non-differentiable issue. In light of this, we exemplify some task-specific prompts to enable the conversational agent to (i) produce questions by prompting queried items and attributes, and (ii) extract the key messages from the user's answers. As shown in Figure 2, a conversational agent consisting of an LLM-chat-bot and our online decision tree algorithm would communicate like humans. We also provide some case studies of the conversational agent in the context of a question generator and an answer extractor. See Appendix A4.2 for a detailed description.

# 4 Experiments

## 4.1 Experimental Configurations

We summarize different experimental settings as follows. (i) We design two different quering strategies regarding attributes (shown in line 5 in Algorithm 1). One is querying attributes (i.e., attribute IDs); and the other is querying attribute values. (ii) We introduce two different recommender system settings. One is the hot-start setting (shown in line 1 in Algorithm 1) that initializes the estimated relevance scores of items by a given pre-trained recommender system; and the other is the cold-start setting where those estimated relevance scores are uniformly generated (corresponding to the case where the recommender system knows nothing about the given user). Because the conversational agent $\Psi_{\text{CO}}(\cdot)$ operates in a dynamic process, we de-

Table 1: Results comparison in the context of querying attributes. See Table A1 for the full version.

| $\Psi_{\text{RE}}(\cdot)$ | $\Psi_{\text{CO}}(\cdot)$ | Amazon | | | |
|---|---|---|---|---|---|
| | | T@3 | S@3 | T@5 | S@5 |
| COLD START | ME | 3.04 | 0.98 | 5.00 | 1.00 |
| | CORE | **2.88** | **1.00** | **2.87** | **1.00** |
| | CORE$_{\text{D}}^{+}$ | 2.84 | 1.00 | 2.86 | 1.00 |
| FM | AG | 2.76 | 0.74 | 2.97 | 0.83 |
| | CRM | 3.07 | 0.98 | 3.37 | 1.00 |
| | EAR | 2.98 | 0.99 | 3.13 | 1.00 |
| | CRIF | 2.84 | 1.00 | 2.64 | 1.00 |
| | UNICORN | 2.65 | 1.00 | 2.45 | 1.00 |
| | CORE | **2.17** | **1.00** | **2.16** | **1.00** |
| | CORE$_{\text{D}}^{+}$ | 2.14 | 1.00 | 2.14 | 1.00 |

velop a new simulator to simulate the Human-AI recommendation interactions, which consists of a conversational agent and a user agent. Specifically, for each user, we use her browsing log as session data, and treat all the items receiving positive feedback (e.g., chick) as target items. Then, for each $k$-th turn, when the conversational agent queries an attribute $x \in \mathcal{X}_{k-1}$, the user agent returns a specific attribute value if all the target items hold the same value for $x$; otherwise, the user agent returns Not Care. When the conversational agent queries an attribute value $w_x \in \mathcal{W}_x$, the user agent returns Yes if at least one target item holds $w_x$ as the value of attribute $x$; otherwise, returns No.

For each experimental setting, we first set $K_{\text{MAX}}$, and then evaluate the performance in terms of the average turns needed to end the sessions, denoted as T@$K_{\text{MAX}}$ (where for each session, if $\Psi_{\text{CO}}(\cdot)$ successfully queries the target item within $K_{\text{MAX}}$ turns, then return the success turn; otherwise, we enforce $\Psi_{\text{CO}}(\cdot)$ to query an item at $(K_{\text{MAX}} + 1)$-th turn, if succeeds, return $K_{\text{MAX}} + 1$, otherwise return $K_{\text{MAX}} + 3$); and the average success rate, denoted as S@$K_{\text{MAX}}$ (where for each session, if $\Psi_{\text{CO}}(\cdot)$ does not successfully query the target item within $K_{\text{MAX}}$ turns, then we enforce $\Psi_{\text{CO}}(\cdot)$ to query an item after $K_{\text{MAX}}$ turns, if succeeds, return 1, otherwise return 0).

To verify CORE can be applied to a variety of recommendation platforms, we conduct evaluations on three tubular datasets: Amazon [8, 33], LastFM [9] and Yelp [12], three sequential datasets: Taobao [10], Tmall [11] and Alipay [7], two graph-structured datasets: Douban Movie [35, 53] and Douban Book [35, 53]. The recommendation approaches used in this paper, i.e., $\Psi_{\text{RE}}(\cdot)$s, include FM [38], DEEP FM [20], PNN [37], DIN [52], GRU [23], LSTM [18], MMOE [32], GCN [27] and GAT [46]. We also use COLD START to denote the cold-start recommendation setting. The conversational methods used in this paper, i.e., $\Psi_{\text{CO}}(\cdot)$s, include (i) Abs Greedy (AG) always queries an item with the highest relevance score at each turn; (ii) Max Entropy (ME) always queries the attribute with the maximum entropy in the context of querying attributes, or queries the attribute value of the chosen attribute, with the highest frequency in the context of querying attribute values; (iii) CRM [44], (iv) EAR [30], (v) CRIF [25], (vi) UNICORN [13]. Here, AG can be regarded as a strategy of solely applying $\Psi_{\text{RE}}(\cdot)$. Both CRM and EAR are reinforcement learning based approaches, originally proposed on the basis of FM recommender system. Thus, we also evaluate their performance with hot-start FM-based recommendation methods, because when applying them to a cold-start recommendation platform, their strategies would reduce to a random strategy. Consider that ME is a $\Psi_{\text{CO}}(\cdot)$, independent of $\Psi_{\text{RE}}(\cdot)$ (namely, the performance of hot-start and cold-start recommendation settings are the same); and therefore, we only report their results in the cold-start recommendation setting. We further introduce a variant of CORE, denoted as CORE$_{\text{D}}^{+}$ where we compute and use $\Psi_{\text{CG}}^{\text{D}}(\cdot)$s instead of $\Psi_{\text{CG}}(\cdot)$s in line 5 in Algorithm 1.

We provide detailed descriptions of datasets and data pre-processing, simulation design, baselines, and implementations in Appendix A3.1, A3.2, A3.3, and A3.4.

Table 2: Results comparison of querying attribute values on tabular datasets. See Table A2 for the full version.

| $\Psi_{RE}(\cdot)$ | $\Psi_{CO}(\cdot)$ | Amazon | | | | LastFM | | | | Yelp | | | |
|---|---|---|---|---|---|---|---|---|---|---|---|---|---|
| | | T@3 | S@3 | T@5 | S@5 | T@3 | S@3 | T@5 | S@5 | T@3 | S@3 | T@5 | S@5 |
| COLD START | AG | 6.47 | 0.12 | 7.83 | 0.23 | 6.77 | 0.05 | 8.32 | 0.14 | 6.65 | 0.08 | 8.29 | 0.13 |
| | ME | 6.50 | 0.12 | 8.34 | 0.16 | 6.84 | 0.04 | 8.56 | 0.11 | 6.40 | 0.15 | 8.18 | 0.20 |
| | CORE | **6.02** | **0.25** | **6.18** | **0.65** | **5.84** | **0.29** | **5.72** | **0.74** | **5.25** | **0.19** | **6.23** | **0.65** |
| | $\text{CORE}_D^+$ | 6.00 | 0.26 | 6.01 | 0.67 | 5.79 | 0.30 | 5.70 | 0.75 | 5.02 | 0.21 | 6.12 | 0.68 |
| FM | AG | **2.76** | 0.74 | **2.97** | 0.83 | 4.14 | 0.52 | 4.67 | 0.64 | 3.29 | 0.70 | 3.39 | 0.81 |
| | CRM | 4.58 | 0.28 | 6.42 | 0.38 | 4.23 | 0.34 | 5.87 | 0.63 | 4.12 | 0.25 | 6.01 | 0.69 |
| | EAR | 4.13 | 0.32 | 6.32 | 0.42 | 4.02 | 0.38 | 5.45 | 0.67 | 4.10 | 0.28 | 5.95 | 0.72 |
| | CRIF | 4.34 | 0.28 | 6.24 | 0.45 | 3.98 | 0.58 | 4.11 | 0.76 | 4.07 | 0.31 | 6.02 | 0.70 |
| | UNICORN | 4.43 | 0.30 | 6.15 | 0.52 | 4.00 | 0.54 | 7.56 | 0.11 | 4.40 | 0.25 | 5.38 | 0.77 |
| | CORE | 3.26 | **0.83** | 3.19 | **0.99** | **3.79** | **0.72** | **3.50** | **0.99** | **3.14** | **0.84** | **3.20** | **0.99** |
| | $\text{CORE}_D^+$ | 3.16 | 0.85 | 3.22 | 1.00 | 3.75 | 0.74 | 3.53 | 1.00 | 3.10 | 0.85 | 3.23 | 1.00 |
| DEEP FM | AG | **3.07** | 0.71 | 3.27 | 0.82 | 3.50 | 0.68 | 3.84 | 0.79 | 3.09 | 0.74 | 3.11 | 0.88 |
| | CRM | 4.51 | 0.29 | 6.32 | 0.40 | 4.18 | 0.38 | 5.88 | 0.63 | 4.11 | 0.23 | 6.02 | 0.71 |
| | EAR | 4.47 | 0.30 | 6.35 | 0.43 | 4.01 | 0.37 | 5.43 | 0.69 | 4.01 | 0.32 | 5.74 | 0.75 |
| | CORE | 3.23 | **0.85** | **3.22** | **0.99** | **3.47** | **0.81** | **3.34** | **1.00** | **2.98** | **0.93** | **3.11** | **1.00** |
| PNN | AG | 3.02 | 0.74 | 3.10 | 0.87 | 3.44 | 0.67 | 3.53 | 0.84 | 2.83 | 0.77 | 2.82 | 0.91 |
| | CORE | **3.01** | **0.88** | **3.04** | **0.99** | **3.10** | **0.87** | **3.20** | **0.99** | **2.75** | **0.88** | **2.76** | **1.00** |

Table 3: Results comparison of querying attribute values on sequential datasets. See Table A3 for the full version.

| $\Psi_{RE}(\cdot)$ | $\Psi_{CO}(\cdot)$ | Taobao | | | | Tmall | | | | Alipay | | | |
|---|---|---|---|---|---|---|---|---|---|---|---|---|---|
| | | T@3 | S@3 | T@5 | S@5 | T@3 | S@3 | T@5 | S@5 | T@3 | S@3 | T@5 | S@5 |
| COLD START | AG | 6.30 | 0.15 | 7.55 | 0.27 | 6.80 | 0.04 | 8.54 | 0.09 | 6.47 | 0.11 | 7.95 | 0.19 |
| | ME | 6.43 | 0.14 | 7.82 | 0.29 | 6.76 | 0.05 | 8.50 | 0.12 | 6.71 | 0.07 | 8.46 | 0.11 |
| | CORE | **5.42** | **0.39** | **5.04** | **0.89** | **6.45** | **0.13** | **7.38** | **0.37** | **5.98** | **0.25** | **6.17** | **0.65** |
| DIN | AG | 2.71 | 0.85 | 2.83 | 0.95 | 4.14 | 0.51 | 4.81 | 0.59 | **3.10** | 0.82 | 3.35 | 0.85 |
| | CORE | **2.45** | **0.97** | **2.54** | **1.00** | **4.12** | **0.64** | **4.16** | **0.89** | 3.25 | **0.83** | **3.32** | **0.96** |
| GRU | AG | 2.80 | 0.80 | 2.64 | 0.97 | 3.82 | 0.56 | 4.40 | 0.64 | 3.17 | 0.83 | 3.29 | 0.87 |
| | CORE | **2.31** | **0.98** | **2.44** | **1.00** | **3.81** | **0.72** | **3.91** | **0.92** | **3.10** | **0.84** | **3.11** | **0.96** |
| LSTM | AG | 2.60 | 0.85 | 2.52 | 0.97 | 4.73 | 0.41 | 5.63 | 0.49 | 3.43 | 0.78 | 3.27 | 0.89 |
| | CORE | **2.37** | **0.97** | **2.49** | **1.00** | **4.58** | **0.55** | **4.36** | **0.90** | **3.03** | **0.84** | **3.16** | **0.97** |
| MMOE | AG | 3.04 | 0.75 | 2.98 | 0.92 | 4.10 | 0.54 | 4.56 | 0.62 | 3.58 | 0.83 | 3.90 | 0.92 |
| | CORE | **2.48** | **0.96** | **2.60** | **1.00** | **3.92** | **0.65** | **4.19** | **0.85** | **3.21** | **0.91** | **3.17** | **0.98** |

## 4.2 Experimental Results

We report our results of querying attributes and items in Table 1, and the results of querying attribute values and items in Tables 2, and 3, 4, and summarize our findings as follows.

**Reinforcement learning based methods work well in querying items and attributes but perform poorly in querying items and attribute values.** By comparing Table 1 to Table 2, we can see a huge performance reduction of CRM and EAR. One possible explanation is that compared to attribute IDs, the action space of querying attribute values is much larger. Thus, it usually requires much more collected data to train a well-performed policy.

**T@$K_{MAX}$ could not align with S@$K_{MAX}$.** A higher success rate might not lead to fewer turns, and ME gains a worse performance than AG in some cases in the cold-start recommendation setting. The main reason is that although querying an attribute value can obtain an equivalent or more certainty gain than querying an item at most times, however, only querying (a.k.a., recommending) an item could end a session. Therefore, sometimes, querying an attribute value is too conservative. It explains why CORE outperforms AG in terms of S@3 but gets a lower score of T@3 in the Amazon dataset and FM recommendation base.

**Our conversational agent can consistently improve the recommendation performance in terms of success rate.** CORE can consistently outperform AG, in terms of success rate, especially for the cold-start recommendation setting. As AG means solely using recommender systems, it indicates that $\Psi_{CO}(\cdot)$ can consistently help $\Psi_{RE}(\cdot)$. One possible reason is that our uncertainty minimization

Table 4: Results comparison of querying attribute values on graph datasets. See Table A4 for the full version.

| $\Psi_{\text{RE}}(\cdot)$ | $\Psi_{\text{CO}}(\cdot)$ | Douban Movie | | | | Douban Book | | | |
|---|---|---|---|---|---|---|---|---|---|
| | | T@3 | S@3 | T@5 | S@5 | T@3 | S@3 | T@5 | S@5 |
| COLD START | AG | 6.52 | 0.11 | 7.94 | 0.21 | 6.36 | 0.15 | 7.68 | 0.26 |
| | ME | 6.60 | 0.10 | 8.16 | 0.21 | 6.40 | 0.15 | 8.04 | 0.24 |
| | CORE | **5.48** | **0.38** | **4.84** | **0.94** | **5.96** | **0.26** | **5.08** | **0.92** |
| GAT | AG | 3.75 | 0.63 | 3.65 | 0.87 | 3.56 | 0.64 | 3.41 | 0.87 |
| | CORE | **2.89** | **0.91** | **2.97** | **1.00** | **2.80** | **0.92** | **2.91** | **1.00** |
| GCN | AG | 3.21 | 0.69 | 3.33 | 0.83 | 3.20 | 0.71 | 3.18 | 0.89 |
| | CORE | **2.76** | **0.92** | **2.81** | **1.00** | **2.85** | **0.91** | **2.85** | **1.00** |

Table 5: Result comparisons in the context of querying attribute values and items on tabular datasets, where we reduce the 50% training data of FM and DEEP FM.

| $\Psi_{\text{RE}}(\cdot)$ | $\Psi_{\text{CO}}(\cdot)$ | Amazon | | | | LastFM | | | |
|---|---|---|---|---|---|---|---|---|---|
| | | T@3 | S@3 | T@5 | S@5 | T@3 | S@3 | T@5 | S@5 |
| FM | CRIF | 5.14 | 0.18 | 6.78 | 0.41 | 4.98 | 0.33 | 6.34 | 0.56 |
| | UNICORN | 5.13 | 0.23 | 6.89 | 0.42 | 5.01 | 0.34 | 5.94 | 0.45 |
| | CORE | **3.86** | **0.73** | **3.98** | **0.80** | **3.99** | **0.65** | **4.04** | **0.86** |
| DEEP FM | CRIF | 5.42 | 0.39 | 5.98 | 0.57 | 4.61 | 0.53 | 4.46 | 0.67 |
| | UNICORN | 4.34 | 0.45 | 5.12 | 0.62 | 4.35 | 0.45 | 4.67 | 0.65 |
| | CORE | **3.96** | **0.78** | **4.23** | **0.85** | **3.75** | **0.72** | **3.83** | **0.86** |

framework unifies querying attribute values and items. In other words, AG is a special case of CORE, where only querying items is allowed.

**Considering Dependence among attributes is helpful.** Comparisons between CORE and $\text{CORE}_{\text{D}}^{+}$ reveal that considering the dependence among attributes could improve the performance of CORE in most cases.

We further investigate the impact of $K_{\text{MAX}}$ by assigning $K_{\text{MAX}} = 1, 3, 5, 7, 9$ and reporting the results of CORE and AG on Amazon dataset in the context of the cold-start and hot-start recommendation settings in Figure 3. The results further verify the superiority of CORE, especially with a cold-start $\Psi_{\text{RE}}(\cdot)$.

**Our conversational agent is more stable when the training data is limited.** For further assessment of CORE's stability in comparison to the baseline methods, we conducted an evaluation by randomly selecting a subset of the training set comprising only 50% of the samples. The results of this evaluation are presented in Table 5. The table clearly demonstrates that CORE exhibits a higher level of stability in its performance when compared to existing reinforcement learning-based frameworks.

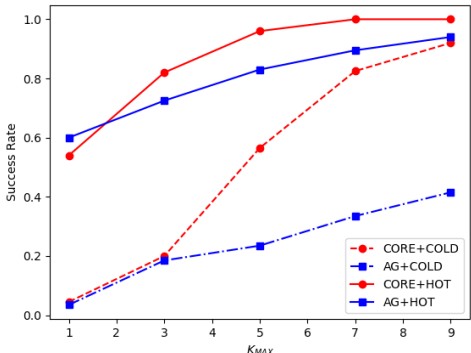

Figure 3: Comparisons of CORE and AG with different $K_{\text{MAX}}$ in both cold-start and hot-start settings.

We also provide a case study of incorporating an LLM into CORE to handle free-text inputs and output human language Appendix A4.2, where we provide detailed prompts. We further offer a visualization of an online decision tree in A4.3.

## 5 Conclusions and Future Work

In this paper, we propose CORE that can incorporate a conversational agent into any recommendation platform in a plug-and-play fashion. Empirical results verify that CORE outperforms existing reinforcement learning-based and statistics-based approaches in both the setting of querying items and attributes, and the setting of querying items and attribute values. In the future, it would be interesting to evaluate CORE in some online real-world recommendation platforms.

**Acknowledgement.** The Shanghai Jiao Tong University team is supported by National Key R&D Program of China (2022ZD0114804), Shanghai Municipal Science and Technology Major Project (2021SHZDZX0102) and National Natural Science Foundation of China (62076161, 62177033). Jiarui Jin would like to thank Wu Wen Jun Honorary Doctoral Scholarship from AI Institute, Shanghai Jiao Tong University.

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

# A1 Conversational Agent Design on Uncertainty Minimization Framework

## A1.1 Detailed Deviations

This paper introduces a conversational agent built upon the recommender system to interact with a human user. We begin by summarizing the interaction principles, taking Figure 1 as an example.

**Definition A1** (**Conversational Agent and Human User Interactions**). *Our conversational agent is designed to act in the following four ways.*

(i) *Query an item $v \in \mathcal{V}_{k-1}$, where $\mathcal{V}_{k-1}$ is the set of unchecked items after $k-1$ interactions (e.g., recommend* Hotel A *to the user).*

(ii) *Query an attribute $x \in \mathcal{X}_{k-1}$, where $\mathcal{X}_{k-1}$ is the set of unchecked attributes after $k-1$ interactions (e.g., query what* Hotel Level *does the user want).*

(iii) *Query whether the user's preference on an attribute $x \in \mathcal{X}_{k-1}$ is equal to a specific attribute value $w_x \in \mathcal{W}_x$ where $\mathcal{W}_x$ is the set of values of attribute $x$ (e.g., query whether the user likes a hotel with* Hotel Level=5*).*

(iv) *Query whether the user's preference on an attribute is not smaller than a specific value $w_x \in \mathbb{R}$ (e.g., query whether the user likes a hotel with* Hotel Level$\geq$3.5*).*

*The human user is supposed to respond in the following ways.*

(i) *For queried item $v$, the user should answer* Yes *(if $v$ satisfies the user) or* No *(otherwise) (e.g., answer* Yes*, if the user likes* Hotel A*).*

(ii) *For queried attribute $x$, the user should answer her preferred attribute value, denoted as $w_x^* \in \mathcal{W}_x$ (e.g., answer 3 for queried attribute* Hotel Level*), or answer* Not Care *to represent that any attribute value works.*

(iii) *For queried attribute value $w_x$, the user should answer* Yes *(if $w_x$ matches the user preference) or* No *(otherwise) (e.g., answer* Yes*, if the user wants a hotel with* Hotel Level=5*), or answer* Not Care *to represent that any attribute value works.*

(iv) *For queried attribute value $w_x$, the user should answer* Yes *(if the user wants an item whose value of attribute $x$ is not smaller than $w_x$) or* No *(otherwise) (e.g., answer* Yes*, if the user wants a hotel with* Hotel Level=5*), or answer* Not Care *to represent that any attribute value works.*

Then, we separately describe the key concepts, including uncertainty, certainty gain, and expected certainty gain, introduced in this paper.

**Definition A2** (**Uncertainty**). *For the $k$-th turn, we define uncertainty, denoted as $\mathtt{U}_k$, to measure how many estimated relevance scores are still unchecked, which can be formulated as:*

$$\mathtt{U}_k := \mathtt{SUM}(\{\Psi_{\mathtt{RE}}(v_m)|v_m \in \mathcal{V}_k\}), \tag{14}$$

*where $\Psi_{\mathtt{RE}}(v_m)$ outputs the estimated relevance score for item $v_m$. The above equation tells us that the uncertainty of each turn is decided by the unchecked items.*

It is straightforward to derive the certainty gain, as the uncertainty reduction at each turn.

**Definition A3** (**Certainty Gain**). *For the $k$-th turn, we define certainty gain of $k$-th interaction as:*

$$\Delta\mathtt{U}_k := \mathtt{U}_{k-1} - \mathtt{U}_k = \mathtt{SUM}(\{\Psi_{\mathtt{RE}}(v_m)|v_m \in \Delta\mathcal{V}_k\}), \tag{15}$$

*where $\Delta\mathcal{V}_k = \mathcal{V}_{k-1} \setminus \mathcal{V}_k$. For simplicity, we use $a$ to denote the $k$-th action of the conversational agent. According to the Human-AI interactions introduced in Definition A1, we can derive:*

$$\Delta\mathcal{V}_k = \begin{cases} \mathcal{V}_k, & a \in \mathcal{V}_{k-1} \text{ and the answer to querying (i) is } \mathtt{Yes}, \\ \{a\}, & a \in \mathcal{V}_{k-1} \text{ and the answer to querying (i) is } \mathtt{No}, \\ \mathcal{V}_{a_{\mathtt{value}} \neq w_a^*} \cap \mathcal{V}_{k-1}, & a \in \mathcal{X}_{k-1} \text{ and the answer to querying (ii) is } w_a^*, \\ \mathcal{V}_{x_{\mathtt{value}} \neq w_x} \cap \mathcal{V}_{k-1}, & a \in \mathcal{W}_x \text{ where } x \in \mathcal{X}_{k-1} \text{ and the answer to querying (iii) is } \mathtt{Yes}, \\ \mathcal{V}_{x_{\mathtt{value}} = w_x} \cap \mathcal{V}_{k-1}, & a \in \mathcal{W}_x \text{ where } x \in \mathcal{X}_{k-1} \text{ and the answer to querying (iii) is } \mathtt{No}, \\ \mathcal{V}_{x_{\mathtt{value}} < w_x} \cap \mathcal{V}_{k-1}, & a \in \mathbb{R}, x \in \mathcal{X}_{k-1} \text{ and the answer to querying (iv) is } \mathtt{Yes}, \\ \mathcal{V}_{x_{\mathtt{value}} \geq w_x} \cap \mathcal{V}_{k-1}, & a \in \mathbb{R}, x \in \mathcal{X}_{k-1} \text{ and the answer to querying (iv) is } \mathtt{No}. \\ \emptyset, & \text{the answer to querying either (ii), (iii) or (iv) is } \mathtt{Not\ Care}, \end{cases} \tag{16}$$

where $\mathcal{V}_{a_{\text{value}} \neq w_a^*}$ is the set of unchecked items whose value of attribute $a$ is not equal to the user answer $w_a^*$, $\mathcal{V}_{x_{\text{value}} \neq w_x}$ is the set of unchecked items whose value of attribute $x$ is not equal to the queried attribute value $w_x$, $\mathcal{V}_{x_{\text{value}} = w_x}$, a subset of $\mathcal{V}_{k-1}$, is the set of unchecked items whose value of attribute $x$ is equal to the queried attribute value $w_x$, $\mathcal{V}_{x_{\text{value}} < w_x}$ is the set of unchecked items whose value of attribute $x$ is smaller than the queried attribute value $w_x$, $\mathcal{V}_{x_{\text{value}} \geq w_x}$ is the set of unchecked items whose value of attribute $x$ is not smaller than the queried attribute value $w_x$.

To estimate the certainty gain from taking each possible action, we introduce the expected certainty gain as follows.

**Definition A4** (**Expected Certainty Gain**). *For the $k$-th turn, we define expected certainty gain to estimate $\Delta \mathrm{U}_k$ on $\Psi_{\text{CO}}(\cdot)$ taking a different action.*

$$\Psi_{\text{CG}}(\cdot) = \begin{cases} Eq.\ (4), & a \in \mathcal{V}_{k-1},\ i.e.,\ querying\ (i), \\ Eq.\ (6), & a \in \mathcal{X}_{k-1},\ i.e.,\ querying\ (ii), \\ Eq.\ (9), & a \in \mathcal{W}_x,\ i.e.,\ querying\ (iii), \\ Eq.\ (11), & a \in \mathbb{R}, x \in \mathcal{X}_{k-1},\ i.e.,\ querying\ (iv). \end{cases} \tag{17}$$

Then, at each turn, we can compute the candidate action, getting the maximum expected certainty gain, as the action to take, denoted as $a_{\text{query}}$. In practice, as shown in Proposition 1, for each attribute, querying attribute IDs, i.e., (ii), and querying attribute values, i.e., (iii), is not compatible. And, (iv) is particularly designed for a large discrete or continuous value space, which can be regarded as a specific attribute value generation engineering for (iii) (i.e., using Eq. (12) to directly compute the queried value for each attribute), and thus, we treat (iv) as a part of (iii). Therefore, we organize two querying strategies. One is querying (i) and (ii), whose objective can be formulated as Eq. (3). The other one is querying (i) and (iii), and the objective can be written as Eq. (10).

Besides $\mathcal{V}_k$, we further summarize the update of $\mathcal{X}_k$ as follows. Similarly, we can define $\Delta \mathcal{X}_k := \mathcal{X}_{k-1} \setminus \mathcal{X}_k$, then $\Delta \mathcal{X}_k$ can be written as:

$$\Delta \mathcal{X}_k = \begin{cases} \{a\}, & querying\ (ii), \\ \{x\}, & querying\ either\ (iii)\ or\ (iv),\ and\ there\ is\ no\ unchecked\ attribute\ value\ in\ x, \\ \emptyset, & querying\ either\ (i)\ or\ (iv). \end{cases} \tag{18}$$

Based on the above, `CORE` runs as Algorithm 1 shows.

**Remark.** One of the advantages of querying attribute values, compared to querying attributes, is that the user's answer to queried attribute would be out of the candidate attribute values (i.e., $\mathcal{W}_x$ for queried attribute $x$). We are also aware that one possible solution is that the conversational agent would list all the candidate attribute values in the query. However, we argue that this approach would work only when the number of candidate values is small (namely, $|\mathcal{W}_x|$ is small) such as attributes `Color` and `Hotel Level`, but can not work when there are many candidate values, e.g., attribute `Brand`, since listing all of them would significantly reduce the user satisfaction.

### A1.2 Proofs

**Proposition A1.** *For any attribute $x \in \mathcal{X}_{k-1}$, $\Psi_{\text{CG}}(\text{query}(x)) \geq \Psi_{\text{CG}}(\text{query}(x) = w_x)$ holds for all the possible value $w_x \in \mathcal{W}_x$.*

*Proof.* For consistency, we re-formulate Eq. (6) as:

$$\Psi_{\text{CG}}(\text{query}(x)) = \sum_{w_x \in \mathcal{W}_x} \Big( \Psi_{\text{CG}}(w_x = w_x^*) \cdot \text{Pr}(w_x = w_x^*) \Big), \tag{19}$$

where $x$ is the queried attribute, and $w_x^*$ represents the user preference on $x$ (corresponding to the notations $a$ and $w_a^*$ respectively). We can also re-write $\Psi_{\text{CG}}(w_x = w_x^*)$ as:

$$\begin{aligned} \Psi_{\text{CG}}(w_x' = w_x^*) &= \text{SUM}(\{\Psi_{\text{RE}}(v_m) | v_m \in \mathcal{V}_{k-1} \cap \mathcal{V}_{x_{\text{value}} \neq w_x'}\}) \\ &= \sum_{w_x'' \in \mathcal{W}_x \setminus \{w_x'\}} \Big( \text{SUM}(\{\Psi_{\text{RE}}(v_m) | v_m \in \mathcal{V}_{k-1} \cap \mathcal{V}_{x_{\text{value}} = w_x''}\}) \Big) \\ &= \sum_{w_x'' \in \mathcal{W}_x \setminus \{w_x'\}} \Psi_{\text{CG}}(w_x'' \neq w_x^*) \geq \Psi_{\text{CG}}(w_x \neq w_x^*), \end{aligned} \tag{20}$$

where $w_x$ is an arbitrary attribute value in $\mathcal{W}_x \backslash \{w'_x\}$. The above equation is built upon the simple fact that after an attribute $x$, the answer of the user preferring $w'_x$ is equivalent to the answer of the user not preferring all the other $w''_x$s, which can remove all the unchecked items whose value is equal to any $w''_x$. Thus, the expected certainty gain of knowing the user preferring $w'_x$ is not smaller than knowing the user not preferring any one $w_x \in \mathcal{W}_x \backslash \{w'_x\}$, and the equality holds only in the case where $\mathcal{W}_x = \{w_x, w'_x\}$, namely there are only two candidate attribute values.

Based on the above equations, we can derive:

$$
\begin{aligned}
\Psi_{\mathrm{CG}}(\mathtt{query}(x)) &= \sum_{w_x \in \mathcal{W}_x} \Big( \Psi_{\mathrm{CG}}(w_x = w_x^*) \cdot \mathtt{Pr}(w'_x = w_x^*) \Big) \\
&= \Psi_{\mathrm{CG}}(w_x = w_x^*) \cdot \mathtt{Pr}(w_x = w_x^*) + \sum_{w'_x \in \mathcal{W}_x \backslash \{w_x\}} \Big( \Psi_{\mathrm{CG}}(w'_x = w_x^*) \cdot \mathtt{Pr}(w'_x = w_x^*) \Big) \\
&\geq \Psi_{\mathrm{CG}}(w_x = w_x^*) \cdot \mathtt{Pr}(w_x = w_x^*) + \sum_{w'_x \in \mathcal{W}_x \backslash \{w_x\}} \Big( \Psi_{\mathrm{CG}}(w_x \neq w_x^*) \cdot \mathtt{Pr}(w'_x = w_x^*) \Big) \\
&\geq \Psi_{\mathrm{CG}}(w_x = w_x^*) \cdot \mathtt{Pr}(w_x = w_x^*) + \Psi_{\mathrm{CG}}(w_x \neq w_x^*) \cdot \sum_{w'_x \in \mathcal{W}_x \backslash \{w_x\}} \Big( \mathtt{Pr}(w'_x = w_x^*) \Big) \\
&\geq \Psi_{\mathrm{CG}}(w_x = w_x^*) \cdot \mathtt{Pr}(w_x = w_x^*) + \Psi_{\mathrm{CG}}(w_x \neq w_x^*) \cdot \mathtt{Pr}(w_x \neq w_x^*) \\
&\geq \Psi_{\mathrm{CG}}(\mathtt{query}(x) = w_x).
\end{aligned}
\tag{21}
$$

Since we put no constraint on $x \in \mathcal{X}_{k-1}$, thus it proves the proposition. $\qquad\square$

**Proposition A2.** *In the context of querying attribute values, an ideal choice is always the one that can partition all the unchecked relevance scores into two equal parts (i.e., the ideal $w_x \in \mathcal{W}_x, x \in \mathcal{X}_{k-1}$ is the one that makes $\Psi_{\mathrm{CG}}(w_x = w_x^*) = \mathtt{SUM}(\{\Psi_{\mathrm{RE}}(v_m)|v_m \in \mathcal{V}_{k-1}\})/2$ hold), if it is achievable. And the certainty gain in this case is $\Psi_{\mathrm{CG}}(\mathtt{query}(x) = w_x) = \mathtt{SUM}(\{\Psi_{\mathrm{RE}}(v_m)|v_m \in \mathcal{V}_{k-1}\})/2$.*

*Proof.* Without loss of generalizability, in the context of querying attribute values, we recap the formulation of $\Psi_{\mathrm{CG}}(\mathtt{query}(x) = w_x)$, shown in Eq. (9) as:

$$
\begin{aligned}
\Psi_{\mathrm{CG}}(\mathtt{query}(x) = w_x) &= \Psi_{\mathrm{CG}}(w_x = w_x^*) \cdot \mathtt{Pr}(w_x = w_x^*) + \Psi_{\mathrm{CG}}(w_x \neq w_x^*) \cdot \mathtt{Pr}(w_x \neq w_x^*) \\
&= \mathtt{SUM}(\{\Psi_{\mathrm{RE}}(v_m)|v_m \in \mathcal{V}_{k-1} \cap \mathcal{V}_{a_{\mathrm{value}} \neq w_a}\}) \cdot \frac{\mathtt{SUM}(\{\Psi_{\mathrm{RE}}(v_m)|v_m \in \mathcal{V}_{k-1} \cap \mathcal{V}_{a_{\mathrm{value}} = w_a}\})}{\mathtt{SUM}(\{\Psi_{\mathrm{RE}}(v_m)|v_m \in \mathcal{V}_{k-1}\})} \\
&\quad + \mathtt{SUM}(\{\Psi_{\mathrm{RE}}(v_m)|v_m \in \mathcal{V}_{k-1} \cap \mathcal{V}_{a_{\mathrm{value}} = w_a}\}) \cdot \frac{\mathtt{SUM}(\{\Psi_{\mathrm{RE}}(v_m)|v_m \in \mathcal{V}_{k-1} \cap \mathcal{V}_{a_{\mathrm{value}} \neq w_a}\})}{\mathtt{SUM}(\{\Psi_{\mathrm{RE}}(v_m)|v_m \in \mathcal{V}_{k-1}\})} \\
&= R_{\mathtt{YES}} \cdot \frac{R - R_{\mathtt{YES}}}{R} + (R - R_{\mathtt{YES}}) \cdot \frac{R_{\mathtt{YES}}}{R},
\end{aligned}
\tag{22}
$$

where we use $R_{\mathtt{YES}}$ to denote $\mathtt{SUM}(\{\Psi_{\mathrm{RE}}(v_m)|v_m \in \mathcal{V}_{k-1} \cap \mathcal{V}_{a_{\mathrm{value}} \neq w_a}\})$, the expected certainty gain of the event $w_x = w_x^*$ happening (i.e., the user answers $\mathtt{Yes}$ to querying $w_x$), and use $R$ to denote the summation of relevance scores of all the unchecked items, i.e., $\mathtt{SUM}(\{\Psi_{\mathrm{RE}}(v_m)|v_m \in \mathcal{V}_{k-1}\})$. For convenience, we use $\Psi$ to denote $\Psi_{\mathrm{CG}}(\mathtt{query}(x) = w_x)$. Then, $\Psi$ can be regarded as a function of $R_{\mathtt{YES}}$. Namely, $R_{\mathtt{YES}}$ is the independent variable and $\Psi$ is the dependent variable.

To maximize $\Psi$, we have:

$$
\frac{\partial \Psi}{\partial R_{\mathtt{YES}}} = \frac{2}{R} \cdot (R - 2 \cdot R_{\mathtt{YES}}) = 0.
\tag{23}
$$

Therefore, we have $R_{\mathtt{Yes}} = R/2$, and in this case, $\Psi = R/2$. Then, we can reach the conclusion that the ideal partition is the one dividing all the unchecked relevance scores, i.e., $R$, into two equal parts; and in this case, $\Psi_{\mathrm{CG}}(\mathtt{query}(x) = w_x) = R/2 = \mathtt{SUM}(\{\Psi_{\mathrm{RE}}(v_m)|v_m \in \mathcal{V}_{k-1}\})/2$, which indicates that querying $w_x$ can check half of the relevance scores in expectation. $\qquad\square$

**Lemma A1.** *In the context of querying attribute values, suppose that $\Psi_{\mathrm{RE}}(v_m) = 1/M$ holds for any $v_m \in \mathcal{V}$, then the expected number of turns (denoted as $\widehat{K}$) is bounded by $\log_2^{M+1} \leq \widehat{K} \leq (M+1)/2$.*

*Proof.* We begin by considering the best case. According to Proposition 2, if we can find an attribute value $w_x$, where querying $w_x$ can partition the unchecked relevance scores into two equal parts, then

we can build a binary tree, where we can check $M/2^k$ at the $k$-th turn. Therefore, we have:

$$1 + 2 + \cdots + 2^{\widehat{K}-1} = M, \tag{24}$$

which derives $\widehat{K} = \log_2^{M+1}$. In the worst case, we only can query one item during one query. Then, the expected number of turns is:

$$\widehat{K} = 1 \cdot \frac{1}{M} + 2 \cdot (1 - \frac{1}{M}) \cdot \frac{1}{M-1} + \cdots + M \cdot \prod_{i=0}^{M-1}(1 - \frac{1}{M-i}) \cdot 1 = \frac{M+1}{2}. \tag{25}$$

Combining Eqs. (24) and (25) together, we can draw $\log_2^{M+1} \leq \widehat{K} \leq (M+1)/2$. $\qquad\square$

## A2 Plugging the Conversational Agent into Recommender Systems

### A2.1 $\Psi_{\texttt{CG}}(\cdot)$ for Querying Attributes in Large Discrete or Continuous Space

The main idea of our conversational agent is to recursively query the user to reduce uncertainty. The core challenge is that there exist some cases where querying any attribute values or items can not effectively reduce uncertainty. Most of these cases occur when some key attributes have a large discrete space or a continuous space, leading to a broad decision tree. Formally, for a key attribute $x \in \mathcal{X}_{k-1}$, a "small" discrete space usually means $|\mathcal{W}_x| \ll |\mathcal{V}_{k-1}|$. For example, for attribute Hotel Price, then querying $x$, the user would not respond with an accurate value, and querying $x$=one possible value could be ineffective.

To address this issue, we propose to find a $w_x \in \mathbb{R}$ instead of $w_x \in \mathcal{W}_x$, and then we can query whether the user's preference is not smaller than it or not, i.e., $\texttt{query}(x) \geq w_x$ instead of whether the user's preference is equal to $w_x$ or not, i.e., $\texttt{query}(x) = w_x$. Then, the expected certainty gain, in this case, can be written as:

$$\Psi_{\texttt{CG}}(\texttt{query}(x) \geq w_x) = \Psi_{\texttt{CG}}(w_x \geq w_x^*) \cdot \texttt{Pr}(w_x \geq w_x^*) + \Psi_{\texttt{CG}}(w_x < w_x^*) \cdot \texttt{Pr}(w_x < w_x^*), \tag{26}$$

where

$$\begin{aligned} \Psi_{\texttt{CG}}(w_x \geq w_x^*) &= \texttt{SUM}(\{\Psi_{\texttt{RE}}(v_m)|v_m \in \mathcal{V}_{x<w_x} \cap \mathcal{V}_{k-1}\}), \\ \Psi_{\texttt{CG}}(w_x < w_x^*) &= \texttt{SUM}(\{\Psi_{\texttt{RE}}(v_m)|v_m \in \mathcal{V}_{x\geq w_x} \cap \mathcal{V}_{k-1}\}), \end{aligned} \tag{27}$$

where $\mathcal{V}_{x\geq w_x}$ is the set of items whose value of attribute $x$ is not smaller than $w_x$, and $\mathcal{V}_{x<w_x}$ is the set of the rest items, namely $\mathcal{V}_{x\geq w_x} \cup \mathcal{V}_{x<w_x} = \mathcal{V}_{k-1}$; and

$$\begin{aligned} \texttt{Pr}(w_x \geq w_x^*) &= \frac{\texttt{SUM}(\{\Psi_{\texttt{RE}}(v_m)|v_m \in \mathcal{V}_{x\geq w_x} \cap \mathcal{V}_{k-1}\})}{\texttt{SUM}(\{\Psi_{\texttt{RE}}(v_{m'})|v_{m'} \in \mathcal{V}_{k-1}\})}, \\ \texttt{Pr}(w_x < w_x^*) &= \frac{\texttt{SUM}(\{\Psi_{\texttt{RE}}(v_m)|v_m \in \mathcal{V}_{x<w_x} \cap \mathcal{V}_{k-1}\})}{\texttt{SUM}(\{\Psi_{\texttt{RE}}(v_{m'})|v_{m'} \in \mathcal{V}_{k-1}\})}. \end{aligned} \tag{28}$$

Therefore, the same as $\texttt{query}(x) = w_x$, $\texttt{query}(x) \geq w_x$ also divides the unchecked items into two parts, and the user is supposed to answer Yes or No, corresponding to either one of the two parts. Then, Proposition 2 also works here. Namely, for each attribute $x \in \mathcal{X}_{k-1}$, the oracle $w_x$, denoted as $w_x^0$, is the one that can partition the relevance scores into two equal parts. Formally, we have:

$$w_x^0 = \underset{w_x \in \mathbb{R}}{\arg\min} \left\| \texttt{SUM}(\{\Psi_{\texttt{RE}}(v_m)|v_m \in \mathcal{V}_{x\geq w_x} \cap \mathcal{V}_{k-1}\}) - \frac{\texttt{SUM}(\{\Psi_{\texttt{RE}}(v_{m'})|v_{m'} \in \mathcal{V}_{k-1}\})}{2} \right\|. \tag{29}$$

Since it is infeasible to find an exact oracle one, we approximate $w_x^0$ as:

$$w_x = \texttt{AVERAGE}(\{\Psi_{\texttt{RE}}(v_m) \cdot w_{v_m}|v_m \in \mathcal{V}_{k-1}\}), \tag{30}$$

where $w_{v_m}$ is the value of attribute $x$ in item $v_m$. It indicates that our estimation is the average of the attribute values for the items in $\mathcal{V}_{k-1}$ weighted by their relevance scores.

## A2.2 Making $\Psi_{\mathtt{CG}}(\cdot)$ Consider Dependence among Attributes

The following techniques allow CORE to take the dependence among attributes into account. We provide two ways, where one requires an FM-based recommender system, while the other one poses no constraint.

Taking $\Psi_{\mathtt{CG}}(\cdot)$ in Eq. (6) as an example, we re-formulate Eq. (6) as, when $a \in \mathcal{X}_{k-1}$, and then we can compute $\Psi_{\mathtt{CG}}(\mathtt{query}(a))$ as:

$$\Psi_{\mathtt{CG}}^{\mathtt{D}}(\mathtt{query}(a)) = \sum_{a' \in \mathcal{A}} \Big( \Psi_{\mathtt{CG}}(\mathtt{query}(a')) \cdot \mathtt{Pr}(\mathtt{query}(a')|\mathtt{query}(a)) \Big), \qquad (31)$$

where we use $\Psi_{\mathtt{CG}}^{\mathtt{D}}(\cdot)$ to denote this variant of $\Psi_{\mathtt{CG}}(\cdot)$.

**Estimation from a Pre-trained FM-based Recommender System.** If our recommender system applies a factorization machine (FM) based recommendation approach, then we can directly adopt the learned weights as the estimation of $\mathtt{Pr}(\mathtt{query}(a')|\mathtt{query}(a))$ in Eq. (31). Taking DeepFM [20] as an example, we begin by recapping its FM component:

$$y_{\mathtt{FM}} = w_0 + \sum_{n=1}^{N} w_n x_n + \sum_{n=1}^{N} \sum_{n'=n+1}^{N} \langle \mathbf{v}_n, \mathbf{v}_{n'} \rangle x_n x_{n'}, \qquad (32)$$

where the model parameters should be estimated in the recommender system (in line 1 in Algorithm 1), including $w_0 \in \mathbb{R}$, $\mathbf{w} \in \mathbb{R}^N$, $\mathbf{V} \in \mathbb{R}^{N \times D}$ and $D$ is the dimension of embedding. And, $\langle \cdot, \cdot \rangle$ is the dot product of two vectors of size $D$, defined as $\langle \mathbf{v}_i, \mathbf{v}_j \rangle = \sum_{d=1}^{D} v_{id} \cdot v_{jd}$. In this regard, for each pair of attributes (e.g., $(a, a')$ in Eq. (31)), we can find the corresponding normalized $\langle \mathbf{v}_n, \mathbf{v}_{n'} \rangle / |\mathbf{v}_n| \cdot |\mathbf{v}_{n'}|$ as the estimation of $\mathtt{Pr}(\mathtt{query}(a')|\mathtt{query}(a))$.

**Estimation in a Statistical Way.** If applying any other recommendation approach to the recommender system, we design a statistical way. We first decompose $\Psi_{\mathtt{CG}}^{\mathtt{D}}(\mathtt{query}(a))$ according to Eq. (6):

$$\Psi_{\mathtt{CG}}^{\mathtt{D}}(\mathtt{query}(a)) = \sum_{w_a \in \mathcal{W}_a} \Big( \Psi_{\mathtt{CG}}^{\mathtt{D}}(w_a = w_a^*) \cdot \mathtt{Pr}(w_a = w_a^*) \Big), \qquad (33)$$

where we define $\Psi_{\mathtt{CG}}^{\mathtt{D}}(w_a = w_a^*)$ as:

$$\Psi_{\mathtt{CG}}^{\mathtt{D}}(w_a = w_a^*) = \sum_{a' \in \mathcal{A}} \sum_{w_{a'} \in \mathcal{W}_{a'}} \Big( \Psi_{\mathtt{CG}}(w_{a'} = w_{a'}^*) \cdot \mathtt{Pr}(w_{a'} = w_{a'}^*|w_a = w_a^*) \Big), \qquad (34)$$

where $\mathtt{Pr}(w_{a'} = w_{a'}^*|w_a = w_a^*)$ measures the probability of how likely getting the user's preference on attribute $a$ (i.e., $w_a = w_a^*$) determinates the user's preference on other attributes (i.e., $w_{a'} = w_{a'}^*$). For example, in Figure 1, if the user's preference on attribute `Hotel Level` is 5 (i.e., $a$ is `Hotel Level`, $w_a$ is 5 and the user's answer is `Yes`), then we could be confident to say that the user preference on attribute `Shower Service` is `Yes` (i.e., $a'$ is `Shower Service`, $w_{a'}$ is `Yes`, and the user's answer is `Yes`), i.e., $\mathtt{Pr}(w_{a'} = w_{a'}^*|w_a = w_a^*)$ is close to 1.

We estimate $\mathtt{Pr}(w_{a'} = w_{a'}^*|w_a = w_a^*)$ by using the definition of the conditional probability:

$$\mathtt{Pr}(w_{a'} = w_{a'}^*|w_a = w_a^*) = \frac{|\mathcal{V}_{(a_{\mathtt{value}}=w_a) \wedge (a'_{\mathtt{value}}=w_{a'})} \cap \mathcal{V}_{k-1}|}{|\mathcal{V}_{a_{\mathtt{value}}=w_a} \cap \mathcal{V}_{k-1}|}, \qquad (35)$$

where $\mathcal{V}_{a_{\mathtt{value}}=w_a}$ is the set of items whose value of $a$ equals $w_a$, and $\mathcal{V}_{(a_{\mathtt{value}}=w_a) \wedge (a'_{\mathtt{value}}=w_{a'})}$ is the set of items whose value of $a$ equals $w_a$ and value of $a'$ equals $w_{a'}$. By incorporating Eqs. (34) and (35) into Eq. (33), we can compute $\Psi_{\mathtt{CG}}^{\mathtt{D}}(\mathtt{query}(a))$ for any $a \in \mathcal{X}_{k-1}$.

**Extensions to Other Cases.** Besides querying attributes, we also introduce another querying strategy to query attribute values. Formally, we can have:

$$\Psi_{\mathtt{CG}}^{\mathtt{D}}(\mathtt{query}(x) = w_a) = \Psi_{\mathtt{CG}}^{\mathtt{D}}(w_x = w_x^*) \cdot \mathtt{Pr}(w_x = w_x^*) + \Psi_{\mathtt{CG}}^{\mathtt{D}}(w_x \neq w_x^*) \cdot \mathtt{Pr}(w_x \neq w_x^*), \quad (36)$$

where $\Psi_{\mathtt{CG}}^{\mathtt{D}}(w_x = w_x^*)$ can be computed by Eq. (34), and the formulation of $\Psi_{\mathtt{CG}}^{\mathtt{D}}(w_x \neq w_x^*)$ could be directly extended from Eq. (34) by replacing $\Psi_{\mathtt{CG}}(w_x = w_x^*)$ with $\Psi_{\mathtt{CG}}(w_x \neq w_x^*)$, and replacing $\mathtt{Pr}(w_{a'} = w_{a'}^*|w_a = w_a^*)$ with $\mathtt{Pr}(w_{a'} \neq w_{a'}^*|w_a \neq w_a^*)$. $\mathtt{Pr}(w_{a'} \neq w_{a'}^*|w_a \neq w_a^*)$ could be computed by replacing $\mathcal{V}_{a_{\mathtt{value}}=w_a}$ with $\mathcal{V}_{a_{\mathtt{value}} \neq w_a}$, and replacing $\mathcal{V}_{(a_{\mathtt{value}}=w_a) \wedge (a'_{\mathtt{value}}=w_{a'})}$ with

$\mathcal{V}_{(a_{\text{value}} \neq w_a) \wedge (a'_{\text{value}} \neq w_{a'})}$. $\mathcal{V}_{a_{\text{value}} \neq w_a}$ is the set of items whose value of $a$ does not equal $w_a$, and $\mathcal{V}_{(a_{\text{value}} \neq w_a) \wedge (a'_{\text{value}} \neq w_{a'})}$ is the set of items whose value of $a$ does not equal $w_a$ and value of $a'$ does not equal $w_{a'}$.

Then, we have made our conversational agent consider the dependence among attributes for cases (ii) and (iii), summarized in Definition A1. There is no need to consider the dependence in case (i), and, as concluded in Appendix A1.1, (iv) can be regarded as a special engineering technique in (iii), and thus, one just needs to follow the same way to handle case (iv).

### A2.3 Overall Algorithm

We summarize the overall algorithm in Algorithm 1. CORE follows an offline-training-and-online-checking paradigm, where offline-training is represented in lines 1 and 12, and online-checking is represented in lines 5, 6 and 7.

As shown in line 5, there are two querying settings, i.e., querying items and attributes, and querying items and attribute values. We note that querying attributes and querying attribute values can be compatible, but can not simultaneously operate on the same attribute. We recap that Proposition 1 says that for each attribute, assuming users could give a clear answer showing their preference, querying an attribute can always obtain certainty gain not smaller than querying any attribute value of the attribute.

Therefore, in practice, we would select those attributes that are likely to receive a clear preference from users (e.g., attributes Category, Brand) in the setting of querying items and attributes, and use the rest of attributes (e.g., attribute Price) in the setting of querying items and attribute values. Also, as stated at the end of Appendix A1.1, we can further select several attributes with a small space of attribute values, use them in the setting of querying items and attributes, and list all the candidate attribute values in the queries. In this regard, for any attribute, since the space of attribute values is changing in the context of querying attribute values, then we may transfer from the setting of querying attribute values to querying attributes, when there are few unchecked candidate attribute values.

All the above operations need careful feature engineering, which should be task-specific and dataset-specific. We argue that this is out of the scope of this paper, and we leave it for future work.

## A3 Experimental Configuration

### A3.1 Dataset Descriptions and Data Pre-processing

We summarize the datasets used in this paper as follows.

- **Amazon** dataset [8, 33] is a dataset collected by Amazon from May 1996 to July 2014. There are 1,114,563 reviews of 133,960 users and 431,827 items and 6 attributes.
- **LastFM** dataset [9] is a dataset collected from Lastfm, a music artist recommendation platform. There are 76,693 interactions of 1,801 users and 7,432 items and 33 attributes.
- **Yelp** dataset [12] is a dataset collected from Yelp, a business recommendation platform. There are 1,368,606 interactions of 27,675 users and 70,311 items and 590 attributes. We follow [30] to create 29 (parents) attributes upon 590 original attributes, and we use the newly created ones in our experiments.
- **Taobao** dataset [10] is a dataset collected by Taobao from November 2007 to December 2007. It consists of 100,150,807 interactions of 987,994 users and 4,162,024 items with an average sequence length of 101 and 4 attributes.
- **Tmall** dataset [11] is a dataset collected by Tmall from May 2015 to November 2015. It consists of 54,925,331 interactions of 424,170 users and 1,090,390 items with an average length of 129 and 9 attributes.
- **Alipay** dataset [7] is a dataset collected by Alipay, from July 2015 to November 2015. There are 35,179,371 interactions of 498,308 users and 2,200,191 items with an average sequence length of 70 and 6 attributes.
- **Douban Movie** dataset [35, 53] is a dataset collected from Douban Movie, a movie recommendation platform. There are 1,278,401 interactions of 2,712 users and 34,893 items with 4 attributes.

Table A1: Results comparison in the context of querying attributes and items on tabular datasets.

| $\Psi_{RE}(\cdot)$ | $\Psi_{CO}(\cdot)$ | Amazon | | | | LastFM | | | | Yelp | | | |
|---|---|---|---|---|---|---|---|---|---|---|---|---|---|
| | | T@3 | S@3 | T@5 | S@5 | T@3 | S@3 | T@5 | S@5 | T@3 | S@3 | T@5 | S@5 |
| COLD START | AG | 6.47 | 0.12 | 7.83 | 0.23 | 6.77 | 0.05 | 8.32 | 0.14 | 6.65 | 0.08 | 8.29 | 0.13 |
| | ME | 3.04 | 0.98 | 5.00 | 1.00 | 3.00 | 1.00 | 5.00 | 1.00 | 3.00 | 1.00 | 5.00 | 1.00 |
| | CORE | **2.88** | **1.00** | **2.87** | **1.00** | **2.73** | **1.00** | **2.75** | **1.00** | **2.92** | **1.00** | **2.94** | **1.00** |
| | CORE$_D^+$ | 2.84 | 1.00 | 2.86 | 1.00 | 2.74 | 1.00 | 2.73 | 1.00 | 2.90 | 1.00 | 2.91 | 1.00 |
| FM | AG | 2.76 | 0.74 | 2.97 | 0.83 | 4.14 | 0.52 | 4.67 | 0.64 | 3.29 | 0.70 | 3.39 | 0.81 |
| | CRM | 3.07 | 0.98 | 3.37 | 0.83 | 2.98 | 0.99 | 3.43 | 1.00 | 3.08 | 0.98 | 3.12 | 0.96 |
| | EAR | 2.98 | 0.99 | 3.13 | 1.00 | 3.02 | 1.00 | 3.51 | 1.00 | 2.94 | 1.00 | 3.02 | 0.99 |
| | CORE | **2.17** | **1.00** | **2.16** | **1.00** | **2.06** | **1.00** | **2.07** | **1.00** | **2.09** | **1.00** | **2.10** | **1.00** |
| | CORE$_D^+$ | 2.14 | 1.00 | 2.14 | 1.00 | 2.05 | 1.00 | 2.05 | 1.00 | 2.10 | 1.00 | 2.08 | 1.00 |
| DEEP FM | AG | 3.07 | 0.71 | 3.27 | 0.82 | 3.50 | 0.68 | 3.84 | 0.79 | 3.09 | 0.74 | 3.11 | 0.88 |
| | CRM | 2.68 | 0.99 | 2.99 | 0.99 | 2.94 | 0.99 | 3.05 | 0.99 | 2.92 | 1.00 | 2.99 | 1.00 |
| | EAR | 2.70 | 1.00 | 2.88 | 1.00 | 2.95 | 1.00 | 3.21 | 0.98 | 2.87 | 1.00 | 2.97 | 1.00 |
| | CRIF | 2.84 | 1.00 | 2.64 | 1.00 | 2.78 | 0.99 | 2.81 | 1.00 | 2.93 | 0.99 | 2.01 | 0.99 |
| | UNICORN | 2.65 | 1.00 | 2.45 | 1.00 | 2.87 | 1.00 | 2.90 | 1.00 | 2.88 | 1.00 | 2.92 | 1.00 |
| | CORE | **2.07** | **1.00** | **2.06** | **1.00** | **2.07** | **1.00** | **2.08** | **1.00** | **2.06** | **1.00** | **2.07** | **1.00** |
| | CORE$_D^+$ | 2.08 | 1.00 | 2.02 | 1.00 | 2.05 | 1.00 | 2.03 | 1.00 | 2.03 | 1.00 | 2.06 | 1.00 |
| PNN | AG | 3.02 | 0.74 | 3.10 | 0.87 | 3.44 | 0.67 | 3.53 | 0.84 | 2.83 | 0.77 | 2.82 | 0.91 |
| | CORE | **2.71** | **1.00** | **3.00** | **1.00** | **2.05** | **1.00** | **2.06** | **1.00** | **2.15** | **1.00** | **2.16** | **1.00** |
| | CORE$_D^+$ | 2.68 | 1.00 | 2.98 | 1.00 | 2.07 | 1.00 | 2.02 | 1.00 | 2.08 | 1.00 | 2.11 | 1.00 |

- **Douban Book** dataset [35, 53] is a dataset collected from Douban Book, a book recommendation platform. There are 96,041 interactions of 2,110 users and 6,777 items with 5 attributes.

In summary, our paper includes three tubular datasets (i.e., Amazon, LastFM, Yelp), three sequential datasets (i.e., Taobao, Tmall, Alipay), and two graph-structured datasets (i.e., Douban Book, Douban Movie). First, we follow the common setting of recommendation evaluation [22, 39] that reduces the data sparsity by pruning the users that have less than 10 historical interactions and the users that have at least 1 positive feedback (e.g., clicks in Taobao). We construct each session by sampling one user and 30 items from her browsing log (if less than 30 items, we randomly sample some items that are not browsed, as the items receive negative feedback, into the session). During sampling, we manage the ratio of the number of items receiving positive feedback and the number of negative feedback falls into the range from 1:10 to 1:30. We use a one-to-one mapping function to map all the attribute values into a discrete space to operate. From those attributes with continuous spaces, we directly apply our proposed method introduced in Section 3.2.

### A3.2 Simulator Design

As summarized in Definition A1, there are two main agents in our simulator, namely a conversational agent and a user agent. The conversational agent is given the set of candidate items (i.e., $\mathcal{V}$), and the set of candidate attributes (i.e., $\mathcal{X}$) (together with their candidate values, i.e., $\mathcal{W}_x$ for every $x \in \mathcal{X}$). Then, at $k$-th turn, the conversational agent is supposed to provide an action of querying, either one from (i), (ii), (iii) and (iv) shown in Definition A1, and the user agent is supposed to generate the corresponding answer and derive the set of unchecked items (i.e., $\mathcal{V}_k$), and the set of unchecked attributes (i.e., $\mathcal{X}_k$) (together with the unchecked values of each attribute $x$). Let $\mathcal{W}_x^k$ be the set of the unchecked values of $x$, then its update function is simple. Firstly, we assign $\mathcal{W}_x^0 = \mathcal{W}_x$, and we can further define $\Delta\mathcal{W}_x^k = \mathcal{W}_x^{k-1} \setminus \mathcal{W}_x^k$, then $\Delta\mathcal{W}_x^k$ can be written as:

$$\Delta\mathcal{W}_x^k = \begin{cases} \{w_x\}, & \text{querying (iii), and selecting an attribute value in } x, \\ \emptyset, & \text{otherwise.} \end{cases} \quad (37)$$

For simplicity, we omit the above update in the main text.

From the above description, we know that the conversational agent and the user agent are communicating through exchanging the set of unchecked items and unchecked attributes (and unchecked attribute values). We also develop a port function in the conversational agent that leverages a pre-trained large language model to generate the human text for each action. See Appendix A4.2 for detailed descriptions and examples.

Table A2: Results comparison of querying attribute values and items on tabular datasets.

| $\Psi_{\mathtt{RE}}(\cdot)$ | $\Psi_{\mathtt{CO}}(\cdot)$ | Amazon | | | | LastFM | | | | Yelp | | | |
|---|---|---|---|---|---|---|---|---|---|---|---|---|---|
| | | T@3 | S@3 | T@5 | S@5 | T@3 | S@3 | T@5 | S@5 | T@3 | S@3 | T@5 | S@5 |
| COLD START | AG | 6.47 | 0.12 | 7.83 | 0.23 | 6.77 | 0.05 | 8.32 | 0.14 | 6.65 | 0.08 | 8.29 | 0.13 |
| | ME | 6.50 | 0.12 | 8.34 | 0.16 | 6.84 | 0.04 | 8.56 | 0.11 | 6.40 | 0.15 | 8.18 | 0.20 |
| | CORE | **6.02** | **0.25** | **6.18** | **0.65** | **5.84** | **0.29** | **5.72** | **0.74** | **5.25** | **0.19** | **6.23** | **0.65** |
| | $\text{CORE}_{\mathtt{D}}^{+}$ | 6.00 | 0.26 | 6.01 | 0.67 | 5.79 | 0.30 | 5.70 | 0.75 | 5.02 | 0.21 | 6.12 | 0.68 |
| FM | AG | **2.76** | 0.74 | **2.97** | 0.83 | 4.14 | 0.52 | 4.67 | 0.64 | 3.29 | 0.70 | 3.39 | 0.81 |
| | CRM | 4.58 | 0.28 | 6.42 | 0.38 | 4.23 | 0.34 | 5.87 | 0.63 | 4.12 | 0.25 | 6.01 | 0.69 |
| | EAR | 4.13 | 0.32 | 6.32 | 0.42 | 4.02 | 0.38 | 5.45 | 0.67 | 4.10 | 0.28 | 5.95 | 0.72 |
| | CRIF | 4.34 | 0.28 | 6.24 | 0.45 | 3.98 | 0.58 | 4.11 | 0.76 | 4.07 | 0.31 | 6.02 | 0.70 |
| | UNICORN | 4.43 | 0.30 | 6.15 | 0.52 | 4.00 | 0.54 | 7.56 | 0.11 | 4.40 | 0.25 | 5.38 | 0.77 |
| | CORE | 3.26 | **0.83** | 3.19 | **0.99** | **3.79** | **0.72** | **3.50** | **0.99** | **3.14** | **0.84** | **3.20** | **0.99** |
| | $\text{CORE}_{\mathtt{D}}^{+}$ | 3.16 | 0.85 | 3.22 | 1.00 | 3.75 | 0.74 | 3.53 | 1.00 | 3.10 | 0.85 | 3.23 | 1.00 |
| DEEP FM | AG | **3.07** | 0.71 | 3.27 | 0.82 | 3.50 | 0.68 | 3.84 | 0.79 | 3.09 | 0.74 | 3.11 | 0.88 |
| | CRM | 4.51 | 0.29 | 6.32 | 0.40 | 4.18 | 0.38 | 5.88 | 0.63 | 4.11 | 0.23 | 6.02 | 0.71 |
| | EAR | 4.47 | 0.30 | 6.35 | 0.43 | 4.01 | 0.37 | 5.43 | 0.69 | 4.01 | 0.32 | 5.74 | 0.75 |
| | CORE | 3.23 | **0.85** | **3.22** | **0.99** | **3.47** | **0.81** | **3.34** | **1.00** | **2.98** | **0.93** | **3.11** | **1.00** |
| | $\text{CORE}_{\mathtt{D}}^{+}$ | 3.16 | 0.87 | 3.21 | 1.00 | 3.45 | 0.83 | 3.30 | 1.00 | 2.97 | 0.94 | 3.10 | 1.00 |
| PNN | AG | 3.02 | 0.74 | 3.10 | 0.87 | 3.44 | 0.67 | 3.53 | 0.84 | 2.83 | 0.77 | 2.82 | 0.91 |
| | CORE | **3.01** | **0.88** | **3.04** | **0.99** | **3.10** | **0.87** | **3.20** | **0.99** | **2.75** | **0.88** | **2.76** | **1.00** |
| | $\text{CORE}_{\mathtt{D}}^{+}$ | 3.00 | 0.92 | 3.04 | 1.00 | 3.05 | 0.88 | 3.12 | 1.00 | 2.74 | 0.88 | 2.76 | 1.00 |

### A3.3 Baseline Descriptions

We first summarize the recommendation approaches, denoted as $\Psi_{\mathtt{RE}}(\cdot)$, used in this paper as follows.

- **COLD START** denotes the cold-start setting, where all the relevance scores of items are uniformly generated. In other words, for the item set $\mathcal{V} = \{v_m\}_{m=1}^{M}$, we set the relevance score for each item $v_m \in \mathcal{V}$ by $\Psi_{\mathtt{RE}}(v_m) = 1/M$.
- **FM** [38] is a factorization machine-based recommendation method working on tabular data, which considers the second-order interactions among attributes (i.e., feature fields).
- **DEEP FM** [20] combines an FM component and a neural network component together to produce the final prediction.
- **PNN** [37] includes an embedding layer to learn a representation of the categorical data and a product layer to capture interactive patterns among categories.
- **DIN** [52] designs a deep interest network that uses a local activation unit to adaptively learn the representation of user interests from historical behaviors.
- **GRU** [23] applies a gated recurrent unit (GRU) to encode the long browsing histories of users.
- **LSTM** [18] applies a long short term memory unit (LSTM) to encode the historical browsing logs of users.
- **MMOE** [32] develops a multi-gate mixture-of-experts that can model the user's multiple behaviors by sharing the expert sub-models across all the behaviors.
- **GCN** [27] designs a graph convolutional network that learns representations of nodes (either users or items) by passing and aggregating their neighborhood information.
- **GAT** [46] designs a graph attention network that adopts an attention mechanism to consider the different contributions from the neighbor nodes in representing the central nodes (either users or items).

We then summarize the conversational techniques, denoted as $\Psi_{\mathtt{CO}}(\cdot)$, used in this paper as follows.

- **AG** (Abs Greedy) always queries an item with the highest relevance score at each turn, which is equivalent to solely using the recommender system as a conversational agent.
- **ME** (Max Entropy) always generates a query in the attribute level. In the setting of querying items and attributes, it queries the attribute with the maximum entropy, which can be formulated as:

$$a_{\mathtt{query}} = \arg\max_{x \in \mathcal{X}_{k-1}} \sum_{w_x \in \mathcal{W}_x} \left( \frac{|\mathcal{V}_{x_{\mathtt{value}}=w_x} \cap \mathcal{V}_{k-1}|}{|\mathcal{V}_{k-1}|} \log \frac{|\mathcal{V}_{x_{\mathtt{value}}=w_x} \cap \mathcal{V}_{k-1}|}{|\mathcal{V}_{k-1}|} \right). \qquad (38)$$

In the setting of querying items and attribute values, we first apply Eq. (38) to obtain the chosen attribute and then we select the attribute value with the highest frequency of the chose attribute as:

$$a_{\texttt{query}} = \underset{w_x \in \mathcal{W}_x}{\arg\max} |\mathcal{V}_{x_{\texttt{value}}=w_x} \cap \mathcal{V}_{k-1}|, \tag{39}$$

where xx is computed following Eq. (38). To evaluate the success rate, during the evaluation turn, we apply AG after employing ME.

- **CRM** [44] integrates the conversational component and the recommender component by feeding the belief tracker results to an FM-based recommendation method. It is originally designed for the single-round setting, and we follow [30] to extend it to the multiple-round setting.

- **EAR** [30] consists of three stages, i.e., the estimation stage to build predictive models to estimate user preference on both items and attributes based on an FM-based recommendation approach, the action stage to determine whether to query attributes or recommend items, the reflection stage to update the recommendation method.

- **CRIF** [25] formulates the conversational recommendation scheme as a four-phase process consisting of offline representation learning, tracking, decision, and inference. In the inference module, by fully utilizing the relation between users' attribute-level and item-level feedback, CRIF can explicitly deduce users' implicit preferences.

- **UNICORN** [13] formulate the conversational recommendation problem as a unified policy learning task. UNICORN uses a dynamic weighted graph-based enforcement learning method to learn a policy to select the action at each conversation turn, either asking an attribute or recommending items.

The proposed methods are listed as follows.

- `CORE` is our proposed method calculating $\Psi_{\texttt{CG}}(\cdot)$s in line 5 in Algorithm 1.

- $\texttt{CORE}_{\texttt{D}}^{+}$ is a variant of `CORE` that computes $\Psi_{\texttt{CG}}^{\texttt{D}}(\cdot)$s instead of $\Psi_{\texttt{CG}}(\cdot)$s, making $\Psi_{\texttt{CG}}^{\texttt{D}}(\cdot)$s consider the dependence among attributes.

### A3.4 Implementation Details

For each recommendation approach, we directly follow their official implementations with the following hyper-parameter settings. The learning rate is decreased from the initial value $1 \times 10^{-2}$ to $1 \times 10^{-6}$ during the training process. The batch size is set as 100. The weight for the L2 regularization term is $4 \times 10^{-5}$. The dropout rate is set as 0.5. The dimension of embedding vectors is set as 64. For those FM-based methods (i.e., FM, DEEP FM), we build a representation vector for each attribute. We treat it as the static part of each attribute embedding, while the dynamic part is the representation of attribute values stored in the recommendation parameters. In practice, we feed the static and dynamic parts together as a whole into the model. After the training process, we store the static part and use it to estimate the dependence among attributes, as introduced in Appendix A2.2. All the models are trained under the same hardware settings with 16-Core AMD Ryzen 9 5950X (2.194GHZ), 62.78GB RAM, NVIDIA GeForce RTX 3080 cards.

## A4 Additional Experimental Results

### A4.1 Performance Comparisons

We conduct the experiment in two different experimental settings. One is the setting of querying items and attributes, and the other is the setting of querying items and attribute values. We report the results of the former setting on tabular datasets (i.e., Amazon, LastFM, Yelp) in Table A1, and also report the results of the latter setting on these tabular datasets in Table A2. We also evaluate the performance of `CORE` in sequential datasets and graph-structured datasets, and report their results in Table A3 and Table A4 respectively.

By combining these tables, our major findings are consistent with those shown in Section 4.2. Moreover, we also note that the performance of `CORE` in querying items and attributes is close to the oracle, and thus considering the dependence among attributes in $\texttt{CORE}_{\texttt{D}}^{+}$ does not bring much improvement.

Table A3: Results comparison of querying attribute values and items on sequential datasets.

| $\Psi_{RE}(\cdot)$ | $\Psi_{CO}(\cdot)$ | Taobao | | | | Tmall | | | | Alipay | | | |
|---|---|---|---|---|---|---|---|---|---|---|---|---|---|
| | | T@3 | S@3 | T@5 | S@5 | T@3 | S@3 | T@5 | S@5 | T@3 | S@3 | T@5 | S@5 |
| COLD START | AG | 6.30 | 0.15 | 7.55 | 0.27 | 6.80 | 0.04 | 8.54 | 0.09 | 6.47 | 0.11 | 7.95 | 0.19 |
| | ME | 6.43 | 0.14 | 7.82 | 0.29 | 6.76 | 0.05 | 8.50 | 0.12 | 6.71 | 0.07 | 8.46 | 0.11 |
| | CORE | **5.42** | **0.39** | **5.04** | **0.89** | **6.45** | **0.13** | **7.38** | **0.37** | **5.98** | **0.25** | **6.17** | **0.65** |
| | CORE$_D^+$ | 5.41 | 0.40 | 5.05 | 0.90 | 6.34 | 0.17 | 7.14 | 0.40 | 5.91 | 0.28 | 6.12 | 0.68 |
| FM | AG | 3.03 | 0.70 | 3.17 | 0.81 | 3.57 | 0.58 | 4.32 | 0.61 | **2.99** | 0.84 | **3.20** | 0.87 |
| | CORE | **3.01** | **0.87** | 2.95 | **1.00** | 3.53 | 0.69 | 4.14 | 0.86 | 3.37 | **0.90** | 3.29 | **0.97** |
| | CORE$_D^+$ | 3.02 | 0.88 | 2.91 | 1.00 | 3.50 | 0.71 | 4.11 | 0.87 | 3.32 | 0.91 | 3.14 | 0.97 |
| DEEP FM | AG | 2.99 | 0.72 | 2.93 | 0.89 | 4.38 | 0.46 | 5.23 | 0.52 | **3.03** | 0.83 | 3.22 | 0.87 |
| | CORE | **2.73** | 0.92 | 2.78 | 0.99 | 4.31 | 0.62 | 4.43 | 0.84 | 3.17 | **0.87** | 3.18 | **0.97** |
| | CORE$_D^+$ | 2.68 | **0.94** | 2.80 | **1.00** | 4.13 | 0.65 | 4.42 | 0.85 | 3.12 | 0.87 | 3.17 | 0.97 |
| PNN | AG | 2.93 | 0.76 | 2.87 | 0.92 | 3.98 | 0.52 | 4.60 | 0.61 | **3.18** | 0.88 | **2.94** | 0.91 |
| | CORE | **2.51** | **0.98** | **2.64** | **1.00** | **3.20** | 0.64 | **4.11** | **0.90** | 3.19 | 0.88 | 3.15 | **0.98** |
| | CORE$_D^+$ | 2.48 | 0.98 | 2.61 | 1.00 | 3.20 | 0.65 | 4.02 | 0.94 | 3.18 | 0.88 | 3.11 | 0.98 |
| DIN | AG | 2.71 | 0.85 | 2.83 | 0.95 | 4.14 | 0.51 | 4.81 | 0.59 | 3.10 | 0.82 | 3.35 | 0.85 |
| | CORE | **2.45** | **0.97** | **2.54** | **1.00** | **4.12** | 0.64 | **4.16** | 0.89 | 3.25 | **0.83** | 3.32 | 0.96 |
| | CORE$_D^+$ | 2.44 | 0.97 | 2.50 | 1.00 | 4.10 | 0.66 | 4.12 | 0.91 | 3.22 | 0.85 | 3.30 | 0.97 |
| GRU | AG | 2.80 | 0.80 | 2.64 | 0.97 | 3.82 | 0.56 | 4.40 | 0.64 | 3.17 | 0.83 | 3.29 | 0.87 |
| | CORE | **2.31** | 0.98 | **2.44** | **1.00** | **3.81** | 0.72 | **3.91** | 0.92 | **3.10** | 0.84 | 3.11 | 0.96 |
| | CORE$_D^+$ | 2.96 | 0.99 | 2.40 | 1.00 | 3.78 | 0.74 | 3.90 | 0.93 | 3.10 | 0.84 | 3.12 | 0.95 |
| LSTM | AG | 2.60 | 0.85 | 2.52 | 0.97 | 4.73 | 0.41 | 5.63 | 0.49 | 3.43 | 0.78 | 3.27 | 0.89 |
| | CORE | **2.37** | 0.97 | **2.49** | **1.00** | **4.58** | 0.55 | **4.36** | 0.90 | 3.03 | 0.84 | 3.16 | 0.97 |
| | CORE$_D^+$ | 2.30 | 0.98 | 2.49 | 1.00 | 4.56 | 0.57 | 4.34 | 0.91 | 3.05 | 0.85 | 3.18 | 0.97 |
| MMOE | AG | 3.04 | 0.75 | 2.98 | 0.92 | 4.10 | 0.54 | 4.56 | 0.62 | 3.58 | 0.83 | 3.90 | 0.92 |
| | CORE | **2.48** | **0.96** | **2.60** | **1.00** | **3.92** | 0.65 | **4.19** | 0.85 | 3.21 | **0.91** | 3.17 | **0.98** |
| | CORE$_D^+$ | 2.46 | 0.97 | 2.61 | 1.00 | 3.90 | 0.66 | 4.20 | 0.84 | 3.19 | 0.89 | 3.12 | 0.99 |

Table A4: Results comparison of querying attribute values and items on graph datasets.

| $\Psi_{RE}(\cdot)$ | $\Psi_{CO}(\cdot)$ | Douban Movie | | | | Douban Book | | | |
|---|---|---|---|---|---|---|---|---|---|
| | | T@3 | S@3 | T@5 | S@5 | T@3 | S@3 | T@5 | S@5 |
| COLD START | AG | 6.52 | 0.11 | 7.94 | 0.21 | 6.36 | 0.15 | 7.68 | 0.26 |
| | ME | 6.60 | 0.10 | 8.16 | 0.21 | 6.40 | 0.15 | 8.04 | 0.24 |
| | CORE | **5.48** | **0.38** | **4.84** | **0.94** | **5.96** | **0.26** | **5.08** | **0.92** |
| | CORE$_D^+$ | 5.45 | 0.40 | 4.81 | 0.94 | 5.91 | 0.28 | 4.98 | 0.94 |
| GAT | AG | 3.75 | 0.63 | 3.65 | 0.87 | 3.56 | 0.64 | 3.41 | 0.87 |
| | CORE | **2.89** | 0.91 | **2.97** | **1.00** | 2.80 | 0.92 | **2.91** | **1.00** |
| | CORE$_D^+$ | 2.87 | 0.92 | 2.96 | 1.00 | 2.81 | 0.93 | 2.90 | 1.00 |
| GCN | AG | 3.21 | 0.69 | 3.33 | 0.83 | 3.20 | 0.71 | 3.18 | 0.89 |
| | CORE | **2.76** | **0.92** | **2.81** | **1.00** | **2.85** | 0.91 | **2.85** | **1.00** |
| | CORE$_D^+$ | 2.74 | 0.93 | 2.80 | 1.00 | 2.83 | 0.93 | 2.78 | 1.00 |

### A4.2 Incorporating Our Conversational Agent with a Frozen Chat-bot

With the development of pre-trained large language models (LLMs), chat-bots built based on these LLMs are capable of communicating like humans, which is a powerful tool to allow our conversational agent to extract the key information from the user's free text feedback and generate free text for querying attributes and items. Concretely, chat-bot can act as either a *question generator* or a *answer extractor*. As shown in Figure 2, if our conversational agent decides to query attribute `breakfast service`, then the command passes to the question generator to generate a free text question "Do you require breakfast service?" The user answers the question by free text "I do not care about breakfast service, and I really want a hotel with shower", and then the answer extractor extracts the user preference on the given answer, namely the user does not care about attribute `breakfast service` and gives positive feedback on attribute `shower`.

For this purpose, we follow a short OpenAI tutorial[2] for prompt engineering to design the following prompts based on `gpt-3.5-turbo` model.

```
# Load the API key and relevant Python libaries.
import openai
import os

def get_completion(prompt, model="gpt-3.5-turbo"):
    messages = [{"role": "user", "content": prompt}]
    response = openai.ChatCompletion.create(
        model=model,
        messages=messages,
        temperature=0, # this is the degree of randomness of the model's output
    )
    return response.choices[0].message["content"]
```

We first evaluate using an LLM as a question generator by an example of generating a question to query an attribute, e.g., `breakfast service`.

```
# Large language model as question generator.
text = f"""
Attribute, Breakfast Service, Hotel
"""
prompt = f"""
You will be provided with text delimited by triple quotes.
If it starts with the word "item", it denotes an item,
then you should generate a text to recommend the item to the user.
Otherwise, it denotes an attribute,
then you should generate a text to query the user's preference on the attribute.
You should be gentle.
```{text}```
"""
response = get_completion(prompt)
print(response)
```

The following is the corresponding output provided by the LLM.

```
Good day! May I ask for your preference regarding breakfast service in a hotel?
Would you like to have a complimentary breakfast or do you prefer to have the
option to purchase breakfast at the hotel restaurant?
```

We then evaluate using an LLM as a question generator by an example of generating a question to query (i.e., recommend) an item, e.g., `hotel A`.

```
# Large language model as question generator.
text = f"""
Item, Hotel A
"""
prompt = f"""
You will be provided with text delimited by triple quotes.
If it starts with the word "item", it denotes an item,
then you should generate a text to recommend the item to the user.
Otherwise, it denotes an attribute,
then you should generate a text to query the user's preference on the attribute.
You should be gentle.
```{text}```
"""
response = get_completion(prompt)
print(response)
```

[2]https://www.deeplearning.ai/short-courses/chatgpt-prompt-engineering-for-developers/

The following is the corresponding output provided by the LLM.

```
Great choice! Hotel A is a wonderful option. Would you like me to provide more
information about the hotel or help you book a room?
```

Also, we evaluate using an LLM as an answer extractor by an example of extracting the user preference on attributes, e.g., `breakfast service`, `hotel level`, and `shower`.

```
text = f"""
I do not care about breakfast service, and I really want a hotel with a shower.
"""
prompt = f"""
You will be provided with text delimited by triple quotes.
If you can infer the user preference on attributes,
then re-write the text in the following format:
[attribute name]: [user perference]
Attribute names include Breakfast Service, Hotel Level, and Shower.
User preference includes Yes to denote the positive preference, No to denote the
negative preference, and Not Care to denote the user does not care.
If you can not infer the user preference on attributes,
then re-write the text in the following format:
[attribute name]: Not Know
'''{text}'''
"""
response = get_completion(prompt)
print(response)
```

The following is the corresponding output by the LLM.

```
Breakfast Service: Not Care
Hotel Level: Not Know
Shower: Yes
```

Similarly, we also can evaluate using an LLM as a question generator by an example of generating a question to query an attribute value, e.g., `Hotel Level=5`.

```
# Large language model as question generator.
text = f"""
Attribute, Hotel Level is 5, Hotel
"""
prompt = f"""
You will be provided with text delimited by triple quotes.
If it starts with the word "item", it denotes an item \
then you should generate a text to recommend the item to the user.
Otherwise, it denotes an attribute \
then you should generate a text to query the user's preference on the attribute.
You should be gentle.
'''{text}'''
"""
response = get_completion(prompt)
print(response)
```

The following is the corresponding output by the LLM.

```
Excuse me, may I ask for your preference on hotel level? Would you prefer a
5-star hotel or are you open to other options?
```

According to Definition A1, we have exemplified querying (i), (ii), and (iii) in the above examples. We further evaluate querying (iv). Namely, we evaluate using an LLM as a question generator by an example of generating a question to query whether the user preference is not smaller than an attribute value, e.g., `Hotel Level` not smaller than 3.

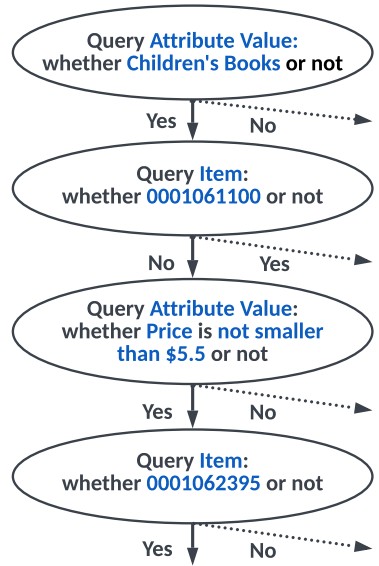

Figure A1: An illustrated example of an online decision tree in the setting of querying items and attribute values, where the target item is 0001062395.

```
# Large language model as question generator.
text = f"""
Attribute, Hotel Level is not smaller than 3, Hotel
"""
prompt = f"""
You will be provided with text delimited by triple quotes.
If it starts with the word "item", it denotes an item \
then you should generate a text to recommend the item to the user.
Otherwise, it denotes an attribute \
then you should generate a text to query the user's preference on the attribute.
You should be gentle.
```{text}```
"""
response = get_completion(prompt)
print(response)
```

The following is the corresponding output by the LLM.

```
Excuse me, may I ask for your preference on hotel level? Would you prefer a
hotel with a level of 3 or higher?
```

We note that if there are too many attribute IDs (or too many attribute values) in the use case, then it might need to further incorporate some hierarchical designs [3] and ambiguous matching [45] into the above system, which is out of the scope of this paper, and we leave it for future work.

### A4.3   Visualization and Case Study

We investigate a case in Amazon dataset in the setting of querying items and attribute values, where the target item is 0001062395. We depict the online decision tree in Figure A1. From the figure, we can see that the conversational agent first queries a value of attribute `Category`, then queries (i.e., recommends) an item 000161100; and after that, it queries another attribute, i.e., `Price`, and finally queries an item 0001062395.

Compared to Figure 1(b), this example seems like a chain. The main reason is that in practice, the user would give the corresponding answer to the query at each turn. Therefore, the binary tree (in the setting of querying items and attribute values) would reduce to a chain.

From this case, we also can observe that our conversational agent is capable of jointly considering the items and attributes to search for the target items in the session.

## A5  Connections to Existing Approaches

### A5.1  Connections to Conversational Recommender Systems

Bridging recently emerged conversational techniques and recommender systems becomes an appealing solution to model the dynamic preference and weak explainability problems in recommendation task [15, 26], where the core sub-task is to dynamically select attributes to query and make recommendations upon the corresponding answers. Along this line, one popular direction is to build a conversational recommender system, which combines the conversational models and the recommendation models from a systematic perspective. In other words, these models are treated and learned as two individual modules [30, 2, 44, 48]. For example, compared to previous literature [5, 6, 47], recent work [49] builds the systems upon the multiple turn scenarios; unfortunately, it does not investigate when to query attributes and when to make recommendations (i.e., query items). To solve this issue, prior works [30, 31] develop reinforcement learning-based solutions. However, all these previous methods based on a reinforcement learning framework are innately suffering from insufficient usage of labeled data and high complexity costs of deployment.

Instead, CORE can be seamlessly adopted to any recommendation method (especially those widely adopted supervised learning-based recommendation methods), and is easy to implement due to our conversational strategy based on the uncertainty minimization theory.

### A5.2  Connections to (Offline) Decision Tree Algorithms

Decision tree algorithms [24] such as ID3 and C4.5 were proposed based on information theory, which measures the *uncertainty* in each status by calculating its entropy. If we want to directly adopt the entropy measurement for the conversational agent, then one possible definition of entropy is

$$\mathtt{H}_k = -\sum_{y \in \mathcal{Y}} \Big( \frac{|\mathcal{V}_{y_{\mathtt{value}}=y} \cap \mathcal{V}_k|}{|\mathcal{V}_k|} \log \frac{|\mathcal{V}_{y_{\mathtt{value}}=y} \cap \mathcal{V}_k|}{|\mathcal{V}_k|} \Big), \tag{40}$$

where $\mathtt{H}_k$ is the empirical entropy for $k$-th turn, $\mathcal{Y}$ is the set of all the labels, and $\mathcal{V}_{y_{\mathtt{value}}=y}$ is the set of items whose label is $y$. For convenience, we call this traditional decision tree as *offline decision tree*.

The main difference between the previous offline decision tree and our online decision tree lies in that *our online decision tree algorithm does not have labels to measure the "uncertainty", instead, we have access to the estimated relevance scores given by recommender systems.* We also note that directly using the user's previous behaviors as the labels would lead to a sub-optimal solution, because (i) offline labels in collected data are often biased and can only cover a small number of candidate items, and (ii) offline labels only can reflect the user's previous interests, but the user's preferences are always shifting.

To this end, we measure the *uncertainty* in terms of the summation of the estimated relevance scores of all the unchecked items after previous $(k-1)$ interactions. Formally, we define our uncertainty as:

$$\mathtt{U}_k = \mathtt{SUM}(\{\Psi_{\mathtt{RE}}(v_m)|v_m \in \mathcal{V}_k\}), \tag{41}$$

where $\Psi_{\mathtt{RE}}(\cdot)$ denotes the recommender system. Similar to the *information gain* in the offline decision tree, we then derive the definition of *certainty gain* (as described in Definition 1), and formulate the conversational agent into an uncertainty minimization framework.

### A5.3  Connections to Recommendation Approaches to Addressing Cold-start Issue

Cold-start issues are situations where no previous events, e.g., ratings, are known for certain users or items [29, 43]. Commonly, previous investigations have revealed that the more (side) information, the better the recommendation results. In light of this, roughly speaking, there are two main branches to address the cold-start problem. One direction is to combine the content information into collaborative filtering to perform a hybrid recommendation [1, 19, 36, 42], and a recent advance [34, 50] proposes to further combine the cross-domain information to the recommender system. The other direction

is to incorporate an active learning strategy into the recommender system [14, 40], whose target is to select items for the newly-signed users to rate. For this purpose, representative methods include the popularity strategy [17], the coverage strategy [16], and the uncertainty reduction strategy [41], where the first one selects items that have been frequently rated by users, the second one selects items that have been highly co-rated with other items, and the third one selects items that can help the recommender system to better learn the user preference.

Integrating the conversational agent with the recommender system offers a promising solution to address the cold start issue by effectively leveraging user interactions. This approach allows for querying additional information from users, thereby enhancing the system's understanding of their preferences and requirements.

A distinctive aspect of our conversational agent's strategy compared to existing active learning strategies, lies in its primary objective. While active learning strategies aim to improve the recommender system's performance, our conversational agent prioritizes meeting the user's immediate needs during the ongoing session. Consequently, the offline-training and online-checking paradigm employed in our approach assigns higher relevance scores to items that are considered "uncertain." This diverges from conventional settings where uncertainty estimation is typically conducted independently of relevance estimation.

By incorporating the conversational agent into recommender system, we leverage user interactions to gather more comprehensive information, which can effectively address the challenges associated with cold-start scenarios. Our primary focus is to provide an optimal user experience in real-time rather than solely optimizing the recommender system's accuracy. This novel paradigm emphasizes the importance of uncertainty estimation in relevance estimation, setting our approach apart from traditional methods.

