}}(a \in \mathcal{V}^*) \cdot \texttt{Pr}(a \in \mathcal{V}^*) + \Psi_{\texttt{CG}}(a \notin \mathcal{V}^*) \cdot \texttt{Pr}(a \notin \mathcal{V}^*)$$
$$= \Big( \sum_{v_m \in \mathcal{V}_{k-1}} \Psi_{\texttt{RE}}(v_m) \Big) \cdot \texttt{Pr}(a \in \mathcal{V}^*) + \Psi_{\texttt{RE}}(a) \cdot \texttt{Pr}(a \notin \mathcal{V}^*), \tag{4}$$

where $a \in \mathcal{V}^*$ and $a \notin \mathcal{V}^*$ denote that queried $a$ is a target item and not. If $a \in \mathcal{V}^*$, the session is done, and therefore, the certainty gain (i.e., $\Psi_{\texttt{CG}}(a \in \mathcal{V}^*)$) is the summation of all the relevance scores in $\mathcal{V}_{k-1}$. Otherwise, only $a$ is checked, and the certainty gain (i.e., $\Psi_{\texttt{CG}}(a \notin \mathcal{V}^*)$) is the relevance score of $a$, and we have $\mathcal{V}_k = \mathcal{V}_{k-1} \backslash \{a\}$ and $\mathcal{X}_k = \mathcal{X}_{k-1}$.

Considering that $a$ being a target item means $a$ being a relevant item, we leverage the user's previous behaviors to estimate the user's current preference. With relevance scores estimated by $\Psi_{\texttt{RE}}(\cdot)$, we estimate $\texttt{Pr}(a \in \mathcal{V}^*)$ as:

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

_{\text{YES}} \cdot \frac{R - R_{\text{YES}}}{R} + (R - R_{\text{YES}}) \cdot \frac{R_{\text{YES}}}{R},
\end{aligned}
\tag{22}
$$

where we use $R_{\text{YES}}$ to denote $\text{SUM}(\{\Psi_{\text{RE}}(v_m)|v_m \in \mathcal{V}_{k-1} \cap \mathcal{V}_{a_{\text{value}} \neq w_a}\})$, the expected certainty gain of the event $w_x = w^*_x$ happening (i.e., the user answering Yes to querying $w_x$), and use $R$ to denote the summation of relevance scores of all the unchecked items, i.e., $\text{SUM}(\{\Psi_{\text{RE}}(v_m)|v_m \in \mathcal{V}_{k-1}\})$. For convenience, we use $\Psi$ to denote $\Psi_{\text{CG}}(\text{query}(x) = w_x)$. Then, $\Psi$ can be regarded as a function of $R_{\text{YES}}$, where $R_{\text{YES}}$ is the independent variable and $\Psi$ is the dependent variable.

To maximize $\Psi$, we have:
$$
\frac{\partial \Psi}{\partial R_{\text{YES}}} = \frac{2}{R} \cdot (R - 2 \cdot R_{\text{YES}}) = 0.
\tag{23}
$$

Therefore, we have $R_{\text{Yes}} = R/2$, and in this case, $\Psi = R/2$. Then, we can reach the conclusion that the ideal partition is the one dividing all the unchecked relevance scores, i.e., $R$, into two equal parts; and in this case, $\Psi_{\text{CG}}(\text{query}(x) = w_x) = R/2 = \text{SUM}(\{\Psi_{\text{RE}}(v_m)|v_m \in \mathcal{V}_{k-1}\})/2$, which indicates that querying $w_x$ can check half of the relevance scores in expectation. $\qquad\square$

**Lemma A1.** *In the context of querying attribute values, suppose that $\Psi_{\text{RE}}(v_m) = 1/M$ holds for any $v_m \in \mathcal{V}$, then the expected number of turns (denoted as $\widehat{K}$) is bounded by $\log_2^{M+1} \leq \widehat{K} \leq (M+1)/2$.*

*Proof.* We begin by considering the best case. According to Proposition 2, if we can find an attribute value $w_x$, where querying $w_x$ can partition the unchecked relevance scores into two equal parts, then

we can build a binary tree, where we can check $M/2^k$ at the $k$-th turn. Therefore, we have:

$$1 + 2 + \cdots + 2^{\widehat{K}-1} = M, \tag{24}$$

which derives $\widehat{K} = \log_2^{M+1}$. In the worst case, we only can query one item during one query. Then, the expected number of turns is:

$$\widehat{K} = 1 \cdot \frac{1}{M} + 2 \cdot (1 - \frac{1}{M}) \cdot \frac{1}{M-1} + \cdots + M \cdot \prod_{i=0}^{M-1} (1 - \frac{1}{M-i}) \cdot 1 = \frac{M+1}{2}. \tag{25}$$

Combining Eqs. (24) and (25) together, we can draw $\log_2^{M+1} \le \widehat{K} \le (M+1)/2$. $\qquad\square$

## A2 Plugging the Conversational Agent in Recommender Systems

### A2.1 $\Psi_{\texttt{CG}}(\cdot)$ for Querying Attributes in Large Discrete or Continuous Space

The main idea of our conversational agent is to recursively query the user to reduce uncertainty. The core challenge is that there exist some cases where querying any attribute values or items can not effectively reduce uncertainty. Most of these cases occur when some key attributes have a large discrete space or a continuous space, leading to a broad decision tree. Formally, for a key attribute $x \in \mathcal{X}_{k-1}$, a "small" discrete space usually means $|\mathcal{W}_x| \ll |\mathcal{V}_{k-1}|$. For example, for attribute $\texttt{Hotel}$ $\texttt{Price}$, then querying $x$, the user would not respond with an accurate value, and querying $x$=one possible value could be ineffective.

To address this issue, we propose a find a $w_x \in \mathbb{R}$ instead of $w_x \in \mathcal{W}_x$, and then we can query whether the user's preference is not smaller than it or not, i.e., $\texttt{query}(x) \ge w_x$ instead of whether the user's preference is equal to $w_x$ or not, i.e., $\texttt{query}(x) = w_x$. Then, the expected certainty gain in this case can be written as:

$$\Psi_{\texttt{CG}}(\texttt{query}(x) \ge w_x) = \Psi_{\texttt{CG}}(w_x \ge w_x^*) \cdot \texttt{Pr}(w_x \ge

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

$$\Psi_{\texttt{CG}}^{\texttt{D}}(w_a = w_a^*) = \sum_{a' \in \mathcal{A}} \sum_{w_{a'} \in \mathcal{W}_{a'}} \Big( \Psi_{\texttt{CG}}(w_{a'} = w_{a'}^*) \cdot \texttt{Pr}(w_{a'} = w_{a'}^*|w_a = w_a^*) \Big), \tag{34}$$

where $\texttt{Pr}(w_{a'} = w_{a'}^*|w_a = w_a^*)$ measures the probability of how likely getting the user's preference on attribute $a$ (i.e., $w_a = w_a^*$) determinate the user's preference on other attributes (i.e., $w_{a'} = w_{a'}^*$). For example, in Figure 1, if the user's preference on attribute Hotel Level and the answer is 5 (i.e., $a$ is Hotel Level, $w_a$ is 5 and the user's answer is Yes), then we could be confident to say that the user preference on attribute Shower Service is Yes (i.e., $a'$ is Shower Service, $w_{a'}$ is Yes, and the user's answer is Yes), i.e., $\texttt{Pr}(w_{a'} = w_{a'}^*|w_a = w_a^*)$ is close to 1.

We estimate $\texttt{Pr}(w_{a'} = w_{a'}^*|w_a = w_a^*)$ by using the definition of the conditional probability:

$$\texttt{Pr}(w_{a'} = w_{a'}^*|w_a = w_a^*) = \frac{|\mathcal{V}_{(a_{\texttt{value}}=w_a) \wedge (a'_{\texttt{value}}=w_{a'})} \cap \mathcal{V}_{k-1}|}{|\mathcal{V}_{a_{\texttt{value}}=w_a} \cap \mathcal{V}_{k-1}|}, \tag{35}$$

where $\mathcal{V}_{a_{\texttt{value}}=w_a}$ is the set of items whose value of $a$ equals $w_a$, and $\mathcal{V}_{(a_{\texttt{value}}=w_a) \wedge (a'_{\texttt{value}}=w_{a'})}$ is the set of items whose value of $a$ equals $w_a$ and value of $a'$ equals $w_{a'}$. By incorporating Eqs. (34) and (35) into Eq. (33), we can compute $\Psi_{\texttt{CG}}^{\texttt{D}}(\texttt{query}(a))$ for any $a \in \mathcal{X}_{k-1}$.

**Extensions to Other Cases.** Besides querying attributes, we also introduce another querying strategy to query attribute values. Formally, we can have:

$$\Psi_{\texttt{CG}}^{\texttt{D}}(\texttt{query}(x) = w_a) = \Psi_{\texttt{CG}}^{\texttt{D}}(w_x = w_x^*) \cdot \texttt{Pr}(w_x = w_x^*) + \Psi_{\texttt{CG}}^{\texttt{D}}(w_x \neq w_x^*) \cdot \texttt{Pr}(w_x \neq w_x^*), \tag{36}$$

where $\Psi_{\texttt{CG}}^{\texttt{D}}(w_x = w_x^*)$ can computed by Eq. (34), and the formulation of $\Psi_{\texttt{CG}}^{\texttt{D}}(w_x \neq w_x^*)$ could be directly extended from Eq. (34) by replacing $\Psi_{\texttt{CG}}(w_x = w_x^*)$ with $\Psi_{\texttt{CG}}(w_x \neq w_x^*)$, and replacing $\texttt{Pr}(w_{a'} = w_{a'}^*|w_a = w_a^*)$ with $\texttt{Pr}(w_{a'} \neq w_{a'}^*|w_a \neq w_a^*)$. $\texttt{Pr}(w_{a'} \neq w_{a'}^*|w_a \neq w_a^*)$ could be computed by replacing $\mathcal{V}_{a_{\texttt{value}}=w_a}$ with $\mathcal{V}_{a_{\texttt{value}} \neq w_a}$, and replacing $\mathcal{V}_{(a_{\texttt{value}}=w_a) \wedge (a'_{\texttt{value}}=w_{a'})}$ with

$\mathcal{V}_{(a_{\texttt{value}} \neq w_a) \wedge (a'_{\texttt{value}} \neq w_{a'})}$. $\mathcal{V}_{a_{\texttt{value}} \neq w_a}$ is the set of items whose value of $a$ does not equal $w_a$, and $\mathcal{V}_{(a_{\texttt{value}} \neq w_a) \wedge (a'_{\texttt{value}} \neq w_{a'})}$ is the set of items whose value of $a$ does not equal $w_a$ and value of $a'$ does not equal $w_{a'}$.

Then, we have made our conversational agent consider the dependence among attributes for cases (ii) and (iii), summarized in Definition A1. There is no need to consider the dependence in case (i), and, as concluded in Appendix A1.1, (iv) can be regarded as a special engineering technique in (iii), and thus, one just follow the same way to handle case (iv).

### A2.3 Overall Algorithm

We summarize the overall algorithm in Algorithm 1. CORE follows an offline-training-and-online-checking paradigm, where offline-training represents in lines 1 and 12, and online-checking represents in lines 5, 6 and 7.

As shown in line 5, there are two querying settings, i.e., querying items and attributes, and querying items and attribute values. We note that querying attributes and querying attribute values can be compatible, but can not simultaneously operate on the same attribute. We recap that Proposition 1 says that for each attribute, assuming users could give a clear answer showing their preference, querying an attribute can always obtain certainty gain not smaller than querying any attribute value of the attribute.

Therefore, in practice, we would select those attributes that are likely to receive a clear preference from users (e.g., attributes Category, Brand) in the setting of querying items and attributes, and use the rest of attributes (e.g., attributes Price) in the setting of querying items and attribute values. Also, as stated at the end of Appendix A1.1, we can further select several attributes with a small space of attribute values, use them in the setting of querying items and attributes, and list all the candidate attribute values in the queries. In this regard, for any attribute, since the space of attribute values is changing in the context of querying attribute values, then we may transfer from the setting of querying attribute values to querying attributes, when there are few unchecked candidate attribute values.

All the above operations need careful feature engineering, which should be task-specific and dataset-specific. We argue that this is out of the scope of this paper, and we leave it for future work.

## A3 Experimental Configuration

### A3.1 Dataset Descriptions and Data Pre-processing

We summarize the datasets used in this paper as follows.

- **Amazon** dataset [8, 32] is a dataset collected by Amazon from May 1996 to July 2014. There are 1,114,563 reviews of 133,960 users and 431,827 items and 6 attributes.
- **LastFM** dataset [9] is a dataset collected from Lastfm, a music artist recommendation platform. There are 76,693 interactions of 1,801 users and 7,432 items and 33 attributes.
- **Yelp** dataset [12] is a dataset collected from Yelp, a business recommendation platform. There are 1,368,606 interactions of 27,675 users and 70,311 items and 590 attributes. We follow [28] to create 29 (parents) attributes upon 590 original attributes, and we use the newly created ones in our experiments.
- **Taobao** dataset [10] is a dataset collected by Taobao from November 2007 to December 2007. It consists of 100,150,807 interactions of 987,994 users and 4,162,024 items with an average sequence length of 101 and 4 attributes.
- **Tmall** dataset [11] is a dataset collected by Tmall from May 2015 to November 2015. It consists of 54,925,331 interactions of 424,170 users and 1,090,390 items with an average length of 129 and 9 attributes.
- **Alipay** dataset [7] is a dataset collected by Alipay, from July 2015 to November 2015. There are 35,179,371 interactions of 498,308 users and 2,200,191 items with an average sequence length of 70 and 6 attributes.
- **Douban Movie** dataset [34, 52] is a dataset collected from Douban Movie, a movie recommendation platform. There are 1,278,401 interactions of 2,712 users and 34,893 items with 4 attributes.

Table A1: Result comparisons in the context of querying attributes and items on tabular datasets. $^*$ indicates that the average value of CORE, when subtracted by the deviation, still outperforms the best baseline.

| $\Psi_{\text{RE}}(\cdot)$ | $\Psi_{\text{CO}}(\cdot)$ | Amazon | | | | LastFM | | | | Yelp | | | |
|---|---|---|---|---|---|---|---|---|---|---|---|---|---|
| | | T@3 | S@3 | T@5 | S@5 | T@3 | S@3 | T@5 | S@5 | T@3 | S@3 | T@5 | S@5 |
| COLD START | AG | 6.47 | 0.12 | 7.83 | 0.23 | 6.77 | 0.05 | 8.32 | 0.14 | 6.65 | 0.08 | 8.29 | 0.13 |
| | ME | 3.04 | 0.98 | 5.00 | 1.00 | 3.00 | 1.00 | 5.00 | 1.00 | 3.00 | 1.00 | 5.00 | 1.00 |
| | CORE | **2.88**$^*$ | **1.00** | **2.87**$^*$ | **1.00** | **2.73**$^*$ | **1.00** | **2.75**$^*$ | **1.00** | **2.92** | **1.00** | **2.94**$^*$ | **1.00** |
| | CORE$_{\text{D}}^{+}$ | 2.84 | 1.00 | 2.86 | 1.00 | 2.74 | 1.00 | 2.73 | 1.00 | 2.90 | 1.00 | 2.91 | 1.00 |
| FM | AG | 2.76 | 0.74 | 2.97 | 0.83 | 4.14 | 0.52 | 4.67 | 0.64 | 3.29 | 0.70 | 3.39 | 0.81 |
| | CRM | 3.07 | 0.98 | 3.37 | 0.83 | 2.98 | 0.99 | 3.43 | 1.00 | 3.08 | 0.98 | 3.12 | 0.96 |
| | EAR | 2.98 | 0.99 | 3.13 | 1.00 | 3.02 | 1.00 | 3.51 | 1.00 | 2.94 | 1.00 | 3.02 | 0.99 |
| | CORE | **2.17**$^*$ | **1.00** | **2.16**$^*$ | **1.00** | **2.06**$^*$ | **1.00** | **2.07**$^*$ | **1.00** | **2.09**$^*$ | **1.00** | **2.10**$^*$ | **1.00** |
| | CORE$_{\text{D}}^{+}$ | 2.14 | 1.00 | 2.14 | 1.00 | 2.05 | 1.00 | 2.05 | 1.00 | 2.10 | 1.00 | 2.08 | 1.00 |
| DEEP FM | AG | 3.07 | 0.71 | 3.27 | 0.82 | 3.50 | 0.68 | 3.84 | 0.79 | 3.09 | 0.74 | 3.11 | 0.88 |
| | CRM | 2.68 | 0.99 | 2.99 | 0.99 | 2.94 | 0.99 | 3.05 | 0.99 | 2.92 | 1.00 | 2.99 | 1.00 |
| | EAR | 2.70 | 1.00 | 2.88 | 1.00 | 2.95 | 1.00 | 3.21 | 0.98 | 2.87 | 1.00 | 2.97 | 1.00 |
| | CORE | **2.07**$^*$ | **1.00** | **2.06**$^*$ | **1.00** | **2.07**$^*$ | **1.00** | **2.08**$^*$ | **1.00** | **2.06**$^*$ | **1.00** | **2.07**$^*$ | **1.00** |
| | CORE$_{\text{D}}^{+}$ | 2.08 | 1.00 | 2.02 | 1.00 | 2.05 | 1.00 | 2.03 | 1.00 | 2.03 | 1.00 | 2.06 | 1.00 |
| PNN | AG | 3.02 | 0.74 | 3.10 | 0.87 | 3.44 | 0.67 | 3.53 | 0.84 | 2.83 | 0.77 | 2.82 | 0.91 |
| | CORE | **2.71**$^*$ | **1.00**$^*$ | **3.00** | **1.00**$^*$ | **2.05**$^*$ | **1.00**$^*$ | **2.06**$^*$ | **1.00**$^*$ | **2.15**$^*$ | **1.00**$^*$ | **2.16**$^*$ | **1.00**$^*$ |
| | CORE$_{\text{D}}^{+}$ | 2.68 | 1.00 | 2.98 | 1.00 | 2.07 | 1.00 | 2.02 | 1.00 | 2.08 | 1.00 | 2.11 | 1.00 |

- **Douban Book** dataset [34, 52] is a dataset collected from Douban Book, a book recommendation platform. There are 96,041 interactions of 2,110 users and 6,777 items with 5 attributes.

In summary, our paper includes three tubular datasets (i.e., Amazon, LastFM, Yelp), three sequential datasets (i.e., Taobao, Tmall, Alipay), and two graph-structured datasets (i.e., Douban Book, Douban Movie). First, we follow the common setting of recommendation evaluation [21, 38] that reduces the data sparsity by pruning the users that have less than 10 historical interactions and the users that have at least 1 positive feedback (e.g., clicks in Taobao). We construct each session by sampling one user and 30 items from her browsing log (if less than 30 items, we randomly sample some items that are not browsed, as the items receive negative feedback, into the session). During sampling, we manage the ratio of the number of items receiving positive feedback and the number of negative feedback fails into the range from 1:10 to 1:30. We use a one-to-one mapping function to map all the attribute values into a discrete space to operate. From those attributes with continuous spaces, we directly apply our proposed method introduced in Section 3.2.

## A3.2 Simulator Design

As summarized in Definition A1, there are two main agents in our simulator, namely a conversational agent and a user agent. The conversational agent is given the set of candidate items (i.e., $\mathcal{V}$), and the set of candidate attributes (i.e., $\mathcal{X}$) (together with their candidate values, i.e., $\mathcal{W}_x$ for every $x \in \mathcal{X}$). Then, at $k$-th turn, the conversational agent is supposed to provide an action of querying, either one from (i), (ii), (iii) and (iv) shown in Definition A1, and the user agent is supposed to generate the corresponding answer and derive the set of unchecked items (i.e., $\mathcal{V}_k$), and the set of unchecked attributes (i.e., $\mathcal{X}_k$) (together with the unchecked values of each attribute $x$). Let $\mathcal{W}_x^k$ be the set of the unchecked values of $x$, then its update function is simple. Firstly, we assign $\mathcal{W}_x^0 = \mathcal{W}_x$, and we can further define $\Delta \mathcal{W}_x^k = \mathcal{W}_x^{k-1} - \mathcal{W}_x^k$, then $\Delta \mathcal{W}_x^k$ can be written as:

$$\Delta \mathcal{W}_x^k = \begin{cases} \{w_x\}, & \text{querying (iii), and selecting an attribute value in } x, \\ \emptyset, & \text{otherwise.} \end{cases} \tag{37}$$

For simplicity, we omit the above update in the main text.

From the above description, we know that the conversational agent and the user agent are communicating through exchanging the set of unchecked items and unchecked attributes (and unchecked attribute values). We also develop a port function in the conversational agent that leverages a pre-trained large language model to generate the human text for each action. See Appendix A4.2 for detailed description and examples.

Table A2: Result comparisons of querying attribute values and items on tabular datasets. $^*$ indicates that the average value of CORE, when subtracted by the deviation, still outperforms the best baseline.

| $\Psi_{\texttt{RE}}(\cdot)$ | $\Psi_{\texttt{CO}}(\cdot)$ | Amazon | | | | LastFM | | | | Yelp | | | |
|---|---|---|---|---|---|---|---|---|---|---|---|---|---|
| | | T@3 | S@3 | T@5 | S@5 | T@3 | S@3 | T@5 | S@5 | T@3 | S@3 | T@5 | S@5 |
| COLD START | AG | 6.47 | 0.12 | 7.83 | 0.23 | 6.77 | 0.05 | 8.32 | 0.14 | 6.65 | 0.08 | 8.29 | 0.13 |
| | ME | 6.50 | 0.12 | 8.34 | 0.16 | 6.84 | 0.04 | 8.56 | 0.11 | 6.40 | 0.15 | 8.18 | 0.20 |
| | CORE | **6.02**$^*$ | **0.25**$^*$ | **6.18**$^*$ | **0.65**$^*$ | **5.84**$^*$ | **0.29**$^*$ | **5.72**$^*$ | **0.74**$^*$ | **5.25**$^*$ | 0.19 | **6.23**$^*$ | **0.65**$^*$ |
| | CORE$_{\texttt{D}}^+$ | 6.00 | 0.26 | 6.01 | 0.67 | 5.79 | 0.30 | 5.70 | 0.75 | 5.02 | 0.21 | 6.12 | 0.68 |
| FM | AG | **2.76** | 0.74 | **2.97** | 0.83 | 4.14 | 0.52 | 4.67 | 0.64 | 3.29 | 0.70 | 3.39 | 0.81 |
| | CRM | 4.58 | 0.28 | 6.42 | 0.38 | 4.23 | 0.34 | 5.87 | 0.63 | 4.12 | 0.25 | 6.01 | 0.69 |
| | EAR | 4.13 | 0.32 | 6.32 | 0.42 | 4.02 | 0.38 | 5.45 | 0.67 | 4.10 | 0.28 | 5.95 | 0.72 |
| | CORE | 3.26 | **0.83**$^*$ | 3.19 | **0.99**$^*$ | **3.79**$^*$ | **0.72**$^*$ | **3.50**$^*$ | **0.99**$^*$ | **3.14**$^*$ | **0.84**$^*$ | **3.20**$^*$ | **0.99**$^*$ |
| | CORE$_{\texttt{D}}^+$ | 3.16 | 0.85 | 3.22 | 1.00 | 3.75 | 0.74 | 3.53 | 1.00 | 3.10 | 0.85 | 3.23 | 1.00 |
| DEEP FM | AG | **3.07** | 0.71 | 3.27 | 0.82 | 3.50 | 0.68 | 3.84 | 0.79 | 3.09 | 0.74 | 3.11 | 0.88 |
| | CRM | 4.51 | 0.29 | 6.32 | 0.40 | 4.18 | 0.38 | 5.88 | 0.63 | 4.11 | 0.23 | 6.02 | 0.71 |
| | EAR | 4.47 | 0.30 | 6.35 | 0.43 | 4.01 | 0.37 | 5.43 | 0.69 | 4.01 | 0.32 | 5.74 | 0.75 |
| | CORE | 3.23 | **0.85**$^*$ | **3.22** | **0.99**$^*$ | **3.47** | **0.81**$^*$ | **3.34**$^*$ | **1.00**$^*$ | **2.98** | **0.93**$^*$ | **3.11** | **1.00**$^*$ |
| | CORE$_{\texttt{D}}^+$ | 3.16 | 0.87 | 3.21 | 1.00 | 3.45 | 0.83 | 3.30 | 1.00 | 2.97 | 0.94 | 3.10 | 1.00 |
| PNN | AG | 3.02 | 0.74 | 3.10 | 0.87 | 3.44 | 0.67 | 3.53 | 0.84 | 2.83 | 0.77 | 2.82 | 0.91 |
| | CORE | **3.01** | **0.88**$^*$ | **3.04** | **0.99**$^*$ | **3.10**$^*$ | **0.87**$^*$ | **3.20**$^*$ | **0.99**$^*$ | **2.75**$^*$ | **0.88**$^*$ | **2.76**$^*$ | **1.00**$^*$ |

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

| $\Psi_{RE}(\cdot)$ | $\Psi_{CO}(\cdot)$ | Taobao | | | | Tmall | | | | Alipay | | | |
|---|---|---|---|---|---|---|---|---|---|---|---|---|---|
| | | T@3 | S@3 | T@5 | S@5 | T@3 | S@3 | T@5 | S@5 | T@3 | S@3 | T@5 | S@5 |
| COLD START | AG | 6.30 | 0.15 | 7.55 | 0.27 | 6.80 | 0.04 | 8.54 | 0.09 | 6.47 | 0.11 | 7.95 | 0.19 |
| | ME | 6.43 | 0.14 | 7.82 | 0.29 | 6.76 | 0.05 | 8.50 | 0.12 | 6.71 | 0.07 | 8.46 | 0.11 |
| | CORE | **5.42**$^*$ | **0.39**$^*$ | **5.04**$^*$ | **0.89**$^*$ | **6.45**$^*$ | **0.13**$^*$ | **7.38**$^*$ | **0.37**$^*$ | **5.98**$^*$ | **0.25**$^*$ | **6.17**$^*$ | **0.65**$^*$ |
| | CORE$_D^+$ | 5.41 | 0.40 | 5.05 | 0.90 | 6.34 | 0.17 | 7.14 | 0.40 | 5.91 | 0.28 | 6.12 | 0.68 |
| FM | AG | 3.03 | 0.70 | 3.17 | 0.81 | 3.57 | 0.58 | 4.32 | 0.61 | **2.99** | 0.84 | **3.20** | 0.87 |
| | CORE | **3.01** | **0.87**$^*$ | **2.95**$^*$ | **1.00**$^*$ | **3.53** | **0.69**$^*$ | **4.14**$^*$ | **0.86**$^*$ | 3.37 | **0.90**$^*$ | 3.29 | **0.97**$^*$ |
| | CORE$_D^+$ | 3.02 | 0.88 | 2.91 | 1.00 | 3.50 | 0.71 | 4.11 | 0.87 | 3.32 | 0.91 | 3.14 | 0.97 |
| DEEP FM | AG | 2.99 | 0.72 | 2.93 | 0.89 | 4.38 | 0.46 | 5.23 | 0.52 | **3.03** | 0.83 | 3.22 | 0.87 |
| | CORE | **2.73**$^*$ | **0.92**$^*$ | **2.78**$^*$ | **0.99**$^*$ | **4.31** | **0.62**$^*$ | **4.43**$^*$ | **0.84**$^*$ | 3.17 | **0.87** | 3.18 | **0.97**$^*$ |
| | CORE$_D^+$ | 2.68 | 0.94 | 2.80 | 1.00 | 4.13 | 0.65 | 4.42 | 0.85 | 3.12 | 0.87 | 3.17 | 0.97 |
| PNN | AG | 2.93 | 0.76 | 2.87 | 0.92 | 3.98 | 0.52 | 4.60 | 0.61 | **3.18** | 0.88 | **2.94** | 0.91 |
| | CORE | **2.51**$^*$ | **0.98**$^*$ | **2.64**$^*$ | **1.00**$^*$ | **3.20**$^*$ | **0.64**$^*$ | **4.11**$^*$ | **0.90**$^*$ | 3.19 | 0.88 | 3.15 | **0.98**$^*$ |
| | CORE$_D^+$ | 2.48 | 0.98 | 2.61 | 1.00 | 3.20 | 0.65 | 4.02 | 0.94 | 3.18 | 0.88 | 3.11 | 0.98 |
| DIN | AG | 2.71 | 0.85 | 2.83 | 0.95 | 4.14 | 0.51 | 4.81 | 0.59 | **3.10** | 0.82 | 3.35 | 0.85 |
| | CORE | **2.45**$^*$ | **0.97**$^*$ | **2.54**$^*$ | **1.00**$^*$ | **4.12** | **0.64**$^*$ | **4.16**$^*$ | **0.89**$^*$ | 3.25 | **0.83** | 3.32 | **0.96**$^*$ |
| | CORE$_D^+$ | 2.44 | 0.97 | 2.50 | 1.00 | 4.10 | 0.66 | 4.12 | 0.91 | 3.22 | 0.85 | 3.30 | 0.97 |
| GRU | AG | 2.80 | 0.80 | 2.64 | 0.97 | 3.82 | 0.56 | 4.40 | 0.64 | 3.17 | 0.83 | 3.29 | 0.87 |
| | CORE | **2.31**$^*$ | **0.98**$^*$ | **2.44**$^*$ | **1.00**$^*$ | **3.81** | **0.72**$^*$ | **3.91**$^*$ | **0.92**$^*$ | **3.10** | **0.84** | **3.11**$^*$ | **0.96**$^*$ |
| | CORE$_D^+$ | 2.96 | 0.99 | 2.40 | 1.00 | 3.78 | 0.74 | 3.90 | 0.93 | 3.10 | 0.84 | 3.12 | 0.95 |
| LSTM | AG | 2.60 | 0.85 | 2.52 | 0.97 | 4.73 | 0.41 | 5.63 | 0.49 | 3.43 | 0.78 | 3.27 | 0.89 |
| | CORE | **2.37**$^*$ | **0.97**$^*$ | **2.49** | **1.00**$^*$ | **4.58**$^*$ | **0.55**$^*$ | **4.36**$^*$ | **0.90**$^*$ | **3.03**$^*$ | **0.84**$^*$ | **3.16**$^*$ | **0.97**$^*$ |
| | CORE$_D^+$ | 2.30 | 0.98 | 2.49 | 1.00 | 4.56 | 0.57 | 4.34 | 0.91 | 3.05 | 0.85 | 3.18 | 0.97 |
| MMOE | AG | 3.04 | 0.75 | 2.98 | 0.92 | 4.10 | 0.54 | 4.56 | 0.62 | 3.58 | 0.83 | 3.90 | 0.92 |
| | CORE | **2.48**$^*$ | **0.96**$^*$ | **2.60**$^*$ | **1.00**$^*$ | **3.92**$^*$ | **0.65**$^*$ | **4.19**$^*$ | **0.85**$^*$ | **3.21**$^*$ | **0.91**$^*$ | **3.17**$^*$ | **0.98**$^*$ |
| | CORE$_D^+$ | 2.46 | 0.97 | 2.61 | 1.00 | 3.90 | 0.66 | 4.20 | 0.84 | 3.19 | 0.89 | 3.12 | 0.99 |

Table A4: Result comparisons of querying attribute values and items on graph datasets. $^*$ indicates that the average value of CORE, when subtracted by the deviation, still outperforms the best baseline.

| $\Psi_{RE}(\cdot)$ | $\Psi_{CO}(\cdot)$ | Douban Movie | | | | Douban Book | | | |
|---|---|---|---|---|---|---|---|---|---|
| | | T@3 | S@3 | T@5 | S@5 | T@3 | S@3 | T@5 | S@5 |
| COLD START | AG | 6.52 | 0.11 | 7.94 | 0.21 | 6.36 | 0.15 | 7.68 | 0.26 |
| | ME | 6.60 | 0.10 | 8.16 | 0.21 | 6.40 | 0.15 | 8.04 | 0.24 |
| | CORE | **5.48**$^*$ | **0.38**$^*$ | **4.84**$^*$ | **0.94**$^*$ | **5.96**$^*$ | **0.26**$^*$ | **5.08**$^*$ | **0.92**$^*$ |
| | CORE$_D^+$ | 5.45 | 0.40 | 4.81 | 0.94 | 5.91 | 0.28 | 4.98 | 0.94 |
| GAT | AG | 3.75 | 0.63 | 3.65 | 0.87 | 3.56 | 0.64 | 3.41 | 0.87 |
| | CORE | **2.89**$^*$ | **0.91**$^*$ | **2.97**$^*$ | **1.00**$^*$ | **2.80**$^*$ | **0.92**$^*$ | **2.91**$^*$ | **1.00**$^*$ |
| | CORE$_D^+$ | 2.87 | 0.92 | 2.96 | 1.00 | 2.81 | 0.93 | 2.90 | 1.00 |
| GCN | AG | 3.21 | 0.69 | 3.33 | 0.83 | 3.20 | 0.71 | 3.18 | 0.89 |
| | CORE | **2.76**$^*$ | **0.92**$^*$ | **2.81**$^*$ | **1.00**$^*$ | **2.85**$^*$ | **0.91**$^*$ | **2.85**$^*$ | **1.00**$^*$ |
| | CORE$_D^+$ | 2.74 | 0.93 | 2.80 | 1.00 | 2.83 | 0.93 | 2.78 | 1.00 |

For this purpose, we follow a short OpenAI tutorial[2] for prompt engineering to design the following prompts based on `gpt-3.5-turbo` model.

```
# Load the API key and relevant Python libaries.
import openai
import os

def get_completion(prompt, model="gpt-3.5-turbo"):
    messages = [{"role": "user", "content": prompt}]
    response = openai.ChatCompletion.create(
        model=model,
        messages=messages,
```

---

[2] https://www.deeplearning.ai/short-courses/chatgpt-prompt-engineering-for-developers/

```
814        temperature=0, # this is the degree of randomness of the model's output
815    )
816    return response.choices[0].message["content"]
```

817  We first evaluate using an LLM as a question generator by an example of generating a question to
818  query an attribute, e.g., `breakfast service`.

```
819  # Large language model as question generator.
820  text = f"""
821  Attribute, Breakfast Service, Hotel
822  """
823  prompt = f"""
824  You will be provided with text delimited by triple quotes.
825  If it starts with the word "item", it denotes an item,
826  then you should generate a text to recommend the item to the user.
827  Otherwise, it denotes an attribute,
828  then you should generate a text to query the user's preference on the attribute.
829  You should be gentle.
830  ```{text}```
831  """
832  response = get_completion(prompt)
833  print(response)
```

834  The following is the corresponding output provided by the LLM.

```
835  Good day! May I ask for your preference regarding breakfast service in a hotel?
836  Would you like to have a complimentary breakfast or do you prefer to have the
837  option to purchase breakfast at the hotel restaurant?
```

838  We then evaluate using an LLM as a question generator by an example of generating a question to
839  query (i.e., recommend) an item, e.g., `hotel A`.

```
840  # Large language model as question generator.
841  text = f"""
842  Item, Hotel A
843  """
844  prompt = f"""
845  You will be provided with text delimited by triple quotes.
846  If it starts with the word "item", it denotes an item,
847  then you should generate a text to recommend the item to the user.
848  Otherwise, it denotes an attribute,
849  then you should generate a text to query the user's preference on the attribute.
850  You should be gentle.
851  ```{text}```
852  """
853  response = get_completion(prompt)
854  print(response)
```

855  The following is the corresponding output provided by the LLM.

```
856  Great choice! Hotel A is a wonderful option. Would you like me to provide more
857  information about the hotel or help you book a room?
```

858  Also, we evaluate using an LLM as an answer extractor by an example of extracting the user
859  preference on attributes, e.g., `breakfast service`, `hotel level`, and `shower`.

```
860  text = f"""
861  I do not care about breakfast service, and I really want a hotel with a shower.
862  """
863  prompt = f"""
864  You will be provided with text delimited by triple quotes.
```

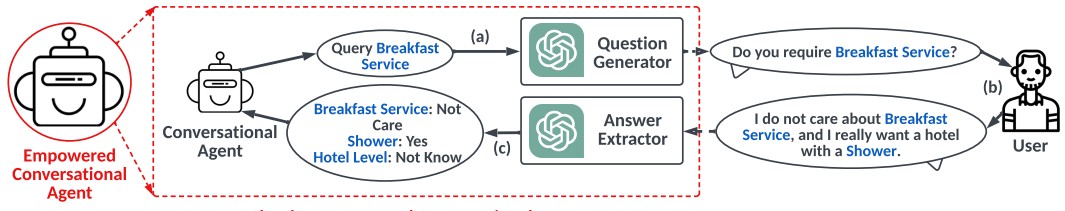

Figure A1: An illustrated example of empowering our conversational agent by a pre-trained chat-bot, where the red box denotes the chat-bot empowered conversational agent. For this purpose, we feed the output queries generated by the original conversational agent, e.g., `Breakfast Service` into the question generator, as shown in (a). The user should input the generated question in a free-text format and provide the corresponding answer also in a free-text format, as shown in (b). The answer extractor would extract the key information from the user response and give them to the original conversational agent, as shown in (c).

```

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

The main reason that our plugging the conversational agent into the recommender system could address the cold-start issue, also can be explained as querying more information from users. The major difference between our conversational agent's strategy and the above active learning strategies is that our goal is not to gain a better recommender system but to meet the user's needs in the current session. Therefore, in our offline-training and online-checking paradigm, the items receiving the high estimated relevance scores are "uncertain" ones, which is pretty different from previous settings (where the uncertainty is usually estimated independently of the relevance estimation).