# OpenReview forum: "Lending Interaction Wings to Recommender Systems with Conversational Agents"
_NeurIPS.cc/2023/Conference — NeurIPS 2023 poster_

### Official Review · Reviewer_SPx5 · 2023-06-25

**Soundness:** 3 good
**Presentation:** 2 fair
**Contribution:** 2 fair
**Rating:** 6
**Confidence:** 3

**Summary:**

This paper proposes a method that combines offline recommendation learning with online decision tree learning to enable recommendation in a conversational format with a few rounds of interaction. The proposed approach is evaluated on multiple datasets and in various validation settings to assess its effectiveness.

**Strengths:**

This paper proposes a novel framework called CORE for uncertainty reduction. CORE improves the performance of recommender systems by revealing user preferences through querying attributes and attribute values.

The paper evaluates the performance of CORE using multiple datasets and different recommendation methods. The experimental results demonstrate that CORE improves the success rate of recommendations.

**Weaknesses:**

(1) This paper heavily relies on decision trees, and there is a significant design cost involved in creating those decision trees. Decision trees are domain-specific and lack clear design guidelines, and the paper does not discuss this aspect or provide any discussion on it in relation to each experimental dataset. Therefore, the weakness lies in the inability to evaluate this aspect.


(2) The reliance on metadata such as price, location, and hotel rank in decision trees makes the approach simplistic. In conversational recommender systems, it is important to consider finer item characteristics that can be obtained from diverse evaluations in user reviews while engaging in conversational interactions with AI. Furthermore, in conversational recommender systems, it is crucial to incorporate language models (LM) to capture linguistic variations and users' intuitive expressions in order to reflect them in the recommendations. However, this aspect is not well addressed in the paper (though it states a little in Section 4).


(3) The evaluation setup of this paper raises some concerns. While the main evaluation metric is the number of turns, the criteria used to determine correctness seem to be based solely on whether an item was checked or not. In the case of hotels, for example, it is important to consider whether a reservation was actually made, and relying solely on item checking may lead to an inflated number of correct predictions.
And, while evaluations are performed on different datasets, there is a lack of detailed explanations about the specifics of the conversations and the actual recommendations made in each dataset. Without this information, it becomes difficult to fully understand the effectiveness and characteristics of the proposed method.
Furthermore, the experimental results of this paper do not include comparisons with other methods or existing approaches. It is important to compare the proposed method with other recommendation techniques in order to clearly demonstrate its advantages and improvements.
Moreover, it appears that there is a lack of qualitative evaluation. While the paper utilizes multiple experimental datasets for validation, it is difficult to grasp the nature of the interactions and how satisfactory item recommendations are achieved in each dataset. Given that the conversation is a crucial aspect, the absence of information on the nature of the conversations makes it challenging to form a clear picture of the system's performance.

**Questions:**

It seems that the paper does not address the importance of considering finer item characteristics derived from diverse evaluations in user reviews during conversational interactions with AI. Could you please provide some insights into this aspect and its potential impact on the proposed approach?

Are there any existing conversational recommender systems that could serve as comparative methods to the proposed approach in this paper?

**Limitations:**

The paper lacks consideration for finer item characteristics obtained from user reviews with language models (LM) to capture linguistic variations and users' intuitive expressions. This limitation restricts the ability to provide more nuanced and personalized recommendations in conversational recommender systems.

---

> ### Author Rebuttal · Authors · 2023-08-10
>
> Thanks for your suggestions. Please also see the main response above.
>
> > CORE heavily relies on decision trees.
>
> We want to emphasize that the core idea of our online decision tree is how to compute the certainty gain, as shown in Section 3.1. As the estimated score for each item is calculated offline, there are no extra computation costs to get these scores online. Therefore, CORE also requires the computation to form a chain (as exemplified in Figure A2) instead of an entire tree.
>
> Furthermore, since the computation of certainty gain is an non-parametric equation, therefore, the computationally costly part would lie in addressing continuous features. But in the context of RS, most of features (e.g., 88% features in Tmall) are discrete features that is efficient to compute.
>
> > It is needed to consider finer characters that can be obtained from diverse evaluations in user reviews.
>
> We apologize that we might not fully understand the review’s meaning in terms of “finer characters”. But as a brief background context, CORE could get these “finer characters” (some complicated user perference hidden in textual responses) by designing specific prompts. For example, a teenager would say that her budget is limited as she is a student in conversations. Our Chatbot APIs would encode these information, and if we ask Chatbot APIs by what her preference on attribute “Price”. Chatbot APIs would return “Price: $1-100”. In this case, we can remove attribute Price from our unchecked attributes. We will show more case studies in our revisions to show that CORE would encode these characters by prompt designs.
>
> > The evaluation setup of this paper raises some concerns.
>
> Firstly, these evaluation metrics are widely used in previous conversational RS papers. Secondly, we agree that these metrics lack of qualitative evaluation. Therefore, we will list some examples of generated converations by CORE and compare them with other conversational RS algorithms by human evaluations. In this regards, each conversation would be evaluated both from how likely the queried items reach user needs and to what extent the texts are friendly to user. Due to time limitation of rebuttal period, we plan to add them in our revision.
>
>
> > Are there any existing conversational recommender systems that could serve as comparative methods to the proposed approach in this paper?
>
> Our main topic focus on how to bridge RS to conversational agent, and therefore, our baselines lie in existing conversational RS methods including CRIF [1], UNICORN [2], EAR [3], and CRM [4]. And, the ability of analyzing user textual response largely depends on what kind of LMs (or conversational components) are used in the method. As CORE can utilize Chatbot APIs, therefore, our ability of analyzing textual responses totally depend on what LLMs are used.
>
> [1] Learning to Infer User Implicit Preference in Conversational Recommendation. 2022.
>
> [2] Unified Conversational Recommendation Policy Learning via Graph-based Reinforcement Learning. 2021.
>
> [3] Estimation-action-reflection: Towards deep interaction between conversational and recommender systems. 2020.
>
> [4] Conversational recommender system. 2018.

---

> > ### Comment · Reviewer_SPx5 · 2023-08-13
> >
> > (1) Thank you for the clarification regarding the computation speed. However, my concern still remains with the design and application of decision trees, which are domain-specific and often lack clear design guidelines. The paper does not appear to address this aspect or discuss its relevance to each experimental dataset. I believe that considering this aspect could provide valuable insights into the applicability and potential limitations of the proposed method.
> >
> > (2) Thank you for providing further insight into the application of CORE in conversational recommender systems. I appreciate the clarification regarding the utilization of specific prompts to capture user preferences and characteristics during conversations.
> >
> > However, my concern still revolves around the consideration of more intricate item characteristics that can be derived from diverse user evaluations present in reviews. For instance, aspects such as musical genre preferences, specific artists, historical periods, and other nuanced interests may require a deeper level of representation than simple attributes like price. Incorporating such detailed characteristics might result in decision trees becoming more complex.

---

> > > ### Author Response · Authors · 2023-08-13
> > > **Response to Reviewer SPx5**
> > >
> > > Thanks for your reply. We are very happy to see some of our clarifications help. We hope the following can further address your concern.
> > >
> > > > Design and application of decision trees are domain-specific and often lack clear design guidelines.
> > >
> > > We apologize that we might not fully understand the review’s meaning of the terms “design” and “application” (if the reviewer could elaborate further, we would be happy to respond more precisely). As a brief background text, we think the concerns may lie in two aspects: (i) How to decide what item (from candidate items) and attributes (or attribute values) (from candidate attributes) to query online? As described in line 5 in Algorithm 1, we compute the expected certainty gain of querying items and attributes (or attribute values) and choose one with the largest expected certainty gain to query. As a result, after multiple runs, CORE can form an online decision tree as illustrated in Figure A2. We want to note that all the above computations only depend on the matrix of candidate items, as illustrated in Figure 1(a), and do not require any handcrafted design. (ii) How to decide candidate items and candidate attributes (i.e., matrix of candidate items). In our experiments, we use all the raw features as the attributes, since CORE, as introduced in Section 3.2, can deal with both discrete and continuous features. In other words, in our experiments, there are no handcrafted designs in both our online and offline components.
> > >
> > > We also note that, while practical, indeed there are some cases where CORE is required to address large feature space and large item space. In these cases, one simple yet effective solution could be doing an (offline) re-selection for both items and attributes. For attributes, we can compute AUC of RS with solely using each attribute as the input, and select those with high AUC as candidates; while for items, we can rank the items according to the estimated scores (given by RS) and select the top as candidates. We would like to note that feature selection is one of the key topics in data mining field investigating how to extract key features from all the raw features, another distinct topic from conversational RS.
> > >
> > > > Discussion on its relevance to each experimental dataset.
> > >
> > > As clarified above, in our results, we use all the raw features as the candidate attributes (to verify that CORE can address both discrete and continuous features) and compute what item and attribute (or attribute value) to query at each turn directly following Algorithm 1. In our experiments, we find that in most cases, discrete features, such as category, play more important roles than continuous space, such as date. This observation would hold for the e-commerce domain as conversational RS is proposed to work on the e-commerce platform. We will summarize the results supporting this observation in our revision. It would be interesting to further explore more observations (i.e., feature selection heuristics) in other domains, which we leave as future work.
> > >
> > > > More intricate item characteristics that can be derived from diverse user evaluations present in reviews, such as musical genre preferences, specific artists, and historical periods.
> > >
> > > Thanks for your question. We want to note that our CORE (i.e., online decision tree algorithm) is not a feature modeling algorithm (a.k.a., is not a feature representation learning algorithm). We model user preference from two components: (i) Offline component is RS which is a feature modeling algorithm modeling (offline) features to get user recent (or previous) preferences including user historical data and other side information (e.g., social networks). (ii) Online component is our conversational agent which directly obtains user online preference by querying the user; and our conversational agent empowered by LLM is expected to model user online conversation (a.k.a., online features) into user preference through LLM, e.g., a user is a student (which can be regarded as an online feature) -> user preference on low-price items due to the limited budget.
> > >
> > > Our CORE is a play-and-play bridge to combine offline and online components where offline features are compressed by RS into estimated scores and online features are summarized by LLM by prompts.
> > >
> > > If you have any remaining or further concerns, we are very glad to discuss them further.

---

> > > > ### Comment · Reviewer_SPx5 · 2023-08-14
> > > >
> > > > Thank you for your prompt and detailed response. I appreciate your clarifications.

---

### Official Review · Reviewer_HpBp · 2023-07-06

**Soundness:** 2 fair
**Presentation:** 1 poor
**Contribution:** 2 fair
**Rating:** 4
**Confidence:** 4

**Summary:**

The paper proposes a novel framework called CORE that bridges conversational agents and recommender systems via an uncertainty minimization principle. The framework treats a recommender system as an offline relevance score estimator and a conversational agent as an online relevance score checker. The conversational agent can query either items or attributes (or attribute values) to reduce the uncertainty of the user’s preference and find a target item. The paper shows that CORE can be applied to various recommendation platforms and datasets, and can outperform existing reinforcement learning-based or statistical methods in both hot-start and cold-start settings. The paper also demonstrates how to empower the conversational agent with a pre-trained language model to communicate more naturally with the user.

**Strengths:**

1.	The paper presents a comprehensive study of the core task of conversational recommendation by considering the complex situations that arise in conversational contexts.
2.	The authors conduct an extensive experimental analysis and exploration.
3.	The proposed method achieves lower complexity and latency compared to existing state-of-the-art approaches.


**Weaknesses:**

1.	The innovations and core contributions of the paper are questionable. Most conversational recommendation papers consist of offline and online components, but CORE does not appear to have any notable novel features despite considering more complex interaction settings. The improvements in these scenarios do not seem to be sufficient to warrant acceptance of this paper.
2.	The formatting and organization of the paper could be improved to facilitate readability, with inconsistencies in notation and an excessive number of annotations. The description in Section 3 is too verbose, with the core method not being highlighted adequately. A concise explanation of Algorithm 1 would suffice to clarify the scenario.
3.	The core of CORE still relies on an offline recommendation system, and even the online decision making depends on scores from the offline recommendation system. The key challenge of conversational recommendation is determining how to make dynamic decisions based on user interests and conversation context. Existing reinforcement learning-based methods evidently consider more factors, whereas the proposed heuristic method in this paper depends on recommendation scores that themselves have uncertainty in dynamic interactions. The methodological foundations of CORE appear weak and unconvincing.


**Questions:**

1.	The modeling approach preferred by CORE users appears to be more of a heuristic estimate compared to some dynamic decision-making methods in reinforcement learning. How does your method ensure that decision-making performance is better than RL-based methods, given that your method seems more like an improved version of Max-entropy?
2.	As mentioned in the Weaknesses Section, all of your decisions rely on the estimated scores from the offline recommendation model. Using this score as a benchmark for uncertainty measurement is dubious. How can you ensure that the recommendation model's estimates are accurate and reliable?


**Limitations:**

1.	Please refer to the Weaknesses Section.
2.	The lack of comparison with the latest CRS methods, such as CPR[1] and UNICORN[2]. These new methods share some similarities with the core settings of the paper, such as UNICORN's decision space, which also includes item space and attribute space.

[1] Wenqiang Lei, Gangyi Zhang, Xiangnan He, Yisong Miao, Xiang Wang, Liang Chen, and Tat-Seng Chua. 2020. Interactive Path Reasoning on Graph for Conversational Recommendation. In Proceedings of the 26th ACM SIGKDD International Conference on Knowledge Discovery & Data Mining (KDD ’20). 2073–2083.
[2] Yang Deng, Yaliang Li, Fei Sun, Bolin Ding, and Wai Lam. 2021. Unified conversational recommendation policy learning via graph-based reinforcement learning. In Proceedings of the 44th International ACM SIGIR Conference on Research and Development in Information Retrieval. 1431–1441.

---

> ### Author Rebuttal · Authors · 2023-08-10
>
> Thanks for your suggestions. Please also see the main response above.
>
> > The innovation and motivation of the paper.
>
> We emphasize that the core idea of CORE is not to introduce an online component but to propose a plug-and-play framework to enable any offline RS to online query user preferences without the need for a heavy change on industrial supervised learning based recommendation platform. Moreover, CORE can use powerful Chatbot APIs while RL methods with joint optimizations over RS components and conversational components can not, since there is no gradient through APIs. Please refer to the main response above for detailed comparison between CORE and RL methods. We are very happy to further discuss if you have any further concern.
>
> > Heavy notations.
>
> Thanks for your suggestion. We will clarify them in the revision.
>
>
> > How do you ensure the CORE outperforms RL?
>
> Indeed, RL can consider complicated states and encode more factors. However, RL often requires large numbers of training samples (known as data insufficiency) and relatively small action space (corresponding to querying attributes in our paper). Also, without sufficient training data, RL could not well generalize to open-world cases. Moreover, RL’s performance largely relies on careful reward function design, while CORE could be easily deployed in various use cases.
>
> Therefore, in many real-world RS cases, the above requirements of RL could not be fully reached. As a result, RL might not get its best performance. In contrast, CORE, a learning-free method (our online decision tree does not have any parameters to tune online), could achieve stable performance and could achieve better performance as few training samples and a huge action space (i.e., querying attribute values).
>
> Please refer to the main response for detailed comparison and a list of experiments for comparison.
>
> > The methodological foundations of CORE are weak and unconvincing.
>
> An intuitive theoretical explanation of CORE is that RS function can be regarded as a prior from offline training, which is unlikely to perform well in the online setting, so we propose CORE to refine the prior online to better capture user needs. We propose Proposition 2 and Lemma 1 to analyze CORE’s performance on an ideal RS case and a bad RS case respectively.
>
> We are very glad to further discuss if you could provide some specific examples of unconvincing foundations.
>
> > Lack of comparison between recent RL methods.
>
> Thanks for your suggestions. We have included CRIF [1] and UNICORN [2] as two new baselines. Results (Tables R1 and R2) verify that CORE can perform well for querying attribute values, while RL methods excel at querying attributes because querying attribute values holds a huge action space which is not friendly for RL.
>
> We will the completed version of the results in our revision.
>
> [1] Learning to Infer User Implicit Preference in Conversational Recommendation. 2022.
>
> [2] Unified Conversational Recommendation Policy Learning via Graph-based Reinforcement Learning. 2021.

---

> > ### Comment · Reviewer_HpBp · 2023-08-22
> >
> > This is still a borderline paper to me. I would like to leave the decision to ASs.

---

### Official Review · Reviewer_8sN6 · 2023-07-10

**Soundness:** 3 good
**Presentation:** 3 good
**Contribution:** 3 good
**Rating:** 7
**Confidence:** 3

**Summary:**

The paper is about conversational recommender systems (CRS), which are systems that can interact with users through natural language and provide personalized recommendations. The paper addresses the challenge of incorporating a conversational agent into any existing recommender system in a plug-and-play fashion, without requiring reinforcement learning or data collection. The paper proposes CORE, a novel offline-training and online-checking framework that bridges a conversational agent and a recommender system via a unified uncertainty minimization objective. The paper claims that CORE can benefit any recommendation platform and can handle different types of data, attributes, and queries. The paper reports that CORE outperforms existing methods in both hot-start and cold-start recommendation settings, and can communicate as a human if empowered by a pre-trained language model.

**Strengths:**

1) Originality: The paper proposes a novel offline-training and online-checking framework, CORE, that bridges a conversational agent and a recommender system via a unified uncertainty minimization objective. This approach is original in its ability to incorporate a conversational agent into any existing recommender system in a plug-and-play fashion, without requiring reinforcement learning or data collection.
2) Quality: The paper develops a new human-AI recommendation simulator and conducts extensive experiments on eight industrial datasets with nine popular recommendation approaches. The results show that CORE outperforms existing methods in both hot-start and cold-start recommendation settings.
3) Significance: The proposed CORE framework has the potential to benefit any recommendation platform and can handle different types of data, attributes, and queries. This makes it a significant contribution to the field of conversational recommender systems.

**Weaknesses:**

1) The introduction does not provide enough background and motivation for the problem of conversational recommender systems. It should explain why this problem is important and challenging, and what are the existing gaps and limitations in the literature. A possible way to improve it is to cite more relevant works and compare them with the proposed approach.
2) The proposed approach in Section 3 is not clearly explained and justified. It does not provide enough details and intuition for how the uncertainty minimization framework works, how the expected certainty gain is derived, how the online decision tree algorithm is implemented, and how the dependence among attributes is considered. It also does not discuss the advantages and disadvantages of the proposed approach compared to other methods. A possible way to improve it is to provide more examples, figures, pseudocode, and analysis to illustrate the proposed approach.
3) The experimental setup in Section 4 is not comprehensive and fair. It does not describe how the datasets are preprocessed, how the hyperparameters are tuned, how the baselines are implemented, and how the evaluation metrics are calculated.
4) The experimental results in Section 4 are not convincing and insightful. They do not show the statistical significance of the performance differences, the impact of different factors/hyperparameters, or the qualitative analysis of the generated conversations. They also do not discuss the limitations and challenges of the proposed approach, such as scalability, robustness, diversity, etc.
5) The references in Section 6 are incomplete and inconsistent. Some references are missing important information such as authors, titles, venues, pages, etc., or have different formats or styles.

**Questions:**

1) How does CORE handle the situation when the user’s answer is not clear or consistent with their previous answers?
2) How does CORE compare with other conversational recommender systems that use natural language generation or understanding techniques?
3) How does CORE deal with the trade-off between exploration and exploitation in querying items or attributes?
4) How does CORE adapt to different domains or scenarios of recommendation, such as books, movies, etc.?
5) How does CORE cope with the noise or bias in the offline estimated relevance scores or the online user responses?
6) How does CORE handle the scalability and efficiency issues when dealing with large-scale datasets or action spaces?
7) How does CORE incorporate user preferences on multiple attributes or items simultaneously?

---

> ### Author Rebuttal · Authors · 2023-08-10
>
> Thanks for your suggestions. Please also see the main responses above.
>
> > Cite more relevant work.
>
> We will include more relevant literature in our revision.
>
> > More examples, figures, pseudocode, and analysis.
>
> We provide detailed derivations of certainty gain and expected certainty gain in Appendix 1.1. We provide a figure to illustrate an example of the online decision tree in Figure A2. The pseudocode of CORE is available in Algorithm 1. The dependence among attributes is encoded by introducing the dependence into the formulation of certainty gain as shown in Eq. (13). Please check the main response above to see the detailed advantages and disadvantages between CORE and other RL methods.
>
> > The experimental setup in Section 4 is not comprehensive and fair.
>
> All the raw data (a.k.a., all the features) in these datasets are used to train our RS, and serve as the candidate attributes (and attribute values). CORE has two stages: (i) One is the offline stage where RS is tuned following the classical supervised learning paradigm and hyper-parameters are listed in Appendix A3.4. (ii) The other one is the online stage where there is no tuned needed and there is no hyper-parameter, as summarized in Algorithm 1. All the baselines are described in Appendix 3.3 and we directly follow their official implementations. The computations of evaluation metrics are described in lines 281 to 287.
>
> > The experimental results in Section 4 are not convincing and insightful.
>
> We will add the t-test evaluation metrics of each table in our revision. As mentioned above, there are no hyperparameters in our online component, we do not discuss the impact of hyperparameters. We have discussed the effect of the offline component: RS from the following three perspectives as shown in the main response above.
>
> > They also do not discuss the limitations and challenges.
>
> We summarize the comparisons between CORE and RL methods in the main response.
>
> > The references in Section 6 are incomplete and inconsistent
>
> We will fix it in our revision.
>
> > How does CORE handle the situation when the user’s answer is not clear or consistent with their previous answers?
>
> Firstly, we allow the user to answer Not Care when the user is not clear about her preference in conversations, as introduced in lines 250-258. Secondly, the inconsistency with their previous answers could be handled by the LLMs (i.e., Chatbots). As stated in Appendix A 4.2, we can use some prompts to ask LLMs about the user preference for specific attributes, and in this case, LLMs are expected to encode the contextual conversations to provide the answers.
>
> > How does CORE compare with other conversational RS that use natural language generation or understanding techniques?
>
> There are two aspects of evaluating the conversational RS. One is to evaluate whether the system can find an item satisfying the user within the minimum number of turns, which corresponds to the evaluation metrics such as average turns and success rate in the paper. The other one is to what extent the generated texts are friendly to users. From this perspective, we plan to do some case studies on the generated texts by CORE. We will list some examples and compare them with the texts generated by other conversational RS algorithms in terms of human evaluations. Due to time limitation of rebuttal period, we plan to add them to our revision. Also, we want to emphasize that CORE does not focus on the quality of generated language, since we can borrow power from pre-trained Chatbots, as described in Section 3.3 and Appendix A4.2.
>
> > How does CORE deal with the trade-off between exploration and exploitation?
>
> We introduce Eq. (10) to tradeoff the exploration by querying attributes and exploitation by recommending items. We also note that Eq. (10) is extendable because one can add some weights or regularization terms to fit some particular use cases (i.e., if the use cases prefer exploration, then one can add a penalty on querying item v; otherwise, one can add a penalty on querying attribute w_x). We will discuss this further in our revision.
>
> > How does CORE adapt to different domains?
>
> From Algorithm 1, one can see CORE only requires the estimated scores of candidate items from RS. Therefore, CORE can be directly adopted in different domains of recommendations once an RS is available. As for the conversational agent, our conversational agent using pre-trained Chatbots could easily adapt to different domains since these Chatbots such as ChatGPT-3.5 carry rich domain knowledge.
>
> >  How does CORE cope with the noise or bias?
>
> Unbiased RS is actually another big topic in the recommendation field, which discussed how to debias from offline data; also addressing ambiguous texts is another big topic in natural language models. Therefore, we argue that these problems are out of the scope of this paper. Moreover, CORE acts as a bridge connecting offline RS and online Chatbots, and therefore, CORE is unbiased and denoised if offline RS and online Chatbots are unbiased and denoised.
>
> > How does CORE handle the scalability and efficiency issues?
>
> We believe CORE is efficient when scaling up to large-scale datasets, due to the following three reasons: (i) Our RS is offline-trained, therefore there are no training costs online. (ii) In practice, there are multiple stages in recommendation platform including matching, pre-ranking, ranking, re-ranking. CORE only needs to perform on re-ranking stage where there are only tens of items needed to be computed. (iii) In real-world cases, all the RS need to assign scores to each re-ranked item, CORE only brings extra computations on the expectation of certainty gain for each recommendation.
>
> > How does CORE incorporate user preferences?
>
> As introduced in Algorithm 1, CORE holds sets of unchecked items and unchecked attributes. Then, once the user shows her preferences, CORE can update the sets correspondingly. We summarize how CORE updates in Appendix A 1.1.

---

> > ### Comment · Reviewer_8sN6 · 2023-08-21
> > **Response to rebuttal**
> >
> > Thank you for the detailed response. I am satisfied with authors' response and improving the score.

---

### Official Review · Reviewer_7XeM · 2023-07-21

**Soundness:** 2 fair
**Presentation:** 2 fair
**Contribution:** 1 poor
**Rating:** 4
**Confidence:** 3

**Summary:**

In this paper, a conversational part of a recommender system is proposed. It is assumed that a recommendation model is available that assigns scores to user-item pairs (which estimate the probabilities of acceptance of the corresponding recommendations), items have a number of important attributes (numerical and categorical), and each user has preferences over the values of these attributes. A conversational model should sequentially decide (based on the recommendation scores) whether to ask the current user about the preferenced values of one attribute, or to try recommending an item, which can be accepted or rejected by the user. The goals are to maximize the rate of successful dialogs which accepted recommendations and to minimize the average number of rounds until the user accepts a recommendation.

In the current paper, authors propose a simple greedy algorithm that chooses an action that minimizes the expected sum of estimated recommendation scores of the unchecked items, that is, items that are still can be chosen by the user in the light of the obtained information so far.

Authors compare their algorithm with two previous baselines based on RL (CRM and EAR) and conclude that the proposed method outperforms RL-based approaches.

**Strengths:**

-	Paper is mostly well-written, the algorithmic ideas and propositions are clear.
-	The idea of expected uncertainty minimization is reasonable

**Weaknesses:**

1.	The main questionable point is the contribution of the paper. I have the following doubts:

-	Papers on conversational RS referenced in related work are very old, I do not see any works dated by 21-23 years. Expected gain maximization resembles expected improvement criterion, one of the most widely-used Bayesian optimization algorithms (see "Efficient global optimization of expensive black-box functions." by Jones et al.) Both algorithms are based on the estimation of the expected gain using the posterior distribution. I think, corresponding references are needed here.

-	Motivation behind the proposed method in comparison with RL-based approaches is not convincing for me. Problem setting does inherently lead to RL approaches: it includes actions, states, rewards, and requires a policy for an optimal trajectory. Why a simple definite empirical greedy algorithm should outperform RL-based models?

2.	Theoretical part is not perfect.

In problem setting, it would be useful to state explicitly what assumptions underly the proposed method. For example, it in implicitly assumed that there is a unique item the user needs (therefore the sum of probabilities over items equals 1, see Eq. 5). Another example is that information on user preferences on an attribute carries binary information on user preferences on items, what underlies their division on “cheched” and “unchecked”.  The motivation behind the approach is based on these assumptions, which are rather controversial in practical cases considered in experiments.

Theoretical results are very simple. Claimed propositions are self-evident, but do not carry much understanding on the specific properties, novelty, and contribution of the proposed EG maximization technique.

There are some inaccuracies in equations. For example:

-	Equation for the action space at line 175 looks formally incorrect. As $W_x$ is different for different $x$, we cannot obtain $A$ as a set product.
-	Eq. 10 Is not formally accurate: the argmax operator should be applied to a function.


3.	Experiments and practice.

The baselines are rather old (CRM and EAR). What about CRIF from “Learning to Infer User Implicit Preference in Conversational Recommendation”?

There is no discussion on the possibility of efficient implementation of the proposed algorithm. How the calculation of uncertainty gains can be implemented in practice? Naïve summation over all items with the given attribute (see, e.g., eqs. 7-8) for each possible attribute values (see eq. 7) looks impractical in real-time conversational recommender systems.



**Questions:**

-	Could you summaries the proposed learning method?
-	What is the theoretical foundation behind the proposed greedy algorithm of expected gain maximization? Is it suboptimal in some assumptions?

**Limitations:**

I do not see any limitations of the proposed work

---

> ### Author Rebuttal · Authors · 2023-08-10
>
> Thanks for your suggestions. Please also see the main response above.
>
> > The main questionable point is the contribution of the paper.
>
> Please see the summary of our contributions in the main response above.
>
> > Reference is old.
>
> Thanks for your suggestion. We will add more recent literature in our revision.
>
> > Why the proposed method outperforms RL based methods?
>
> Indeed, RL can consider complicated states and encode more information. However, RL often requires large numbers of training samples (known as data insufficiency) and relatively small action space (corresponding to querying attributes in our paper). Also, without sufficient training data, RL could not well generalize to open-world cases. Moreover, RL’s performance largely relies on careful reward function design, while CORE could be easily deployed in various use cases.
>
> Therefore, in many real-world RS cases, the above requirements of RL could not be fully reached. As a result, RL might not get its best performance. In contrast, CORE, a learning-free method (our online decision tree does not have any parameters to tune online), could achieve stable performance and could achieve better performance as few training samples and a huge action space (i.e., querying attribute values).
>
> > Implicit assumption on a unique item of a user and binary information on user preferences.
>
> We apologize that we might not fully understand the review’s meaning here (if the reviewer could elaborate further, we would be happy to respond more precisely). But as a brief background context, we do believe CORE does not need these assumptions: (i) Target items do not need to be unique, since we can support a set of target items, as stated as a \in \mathcal{V}^* in lines 131-132, and Eq. 5 is a normalization trick to normalizes the score of each item over all the unchecked items. (ii) We consider the case where users may not care about specific attributes (or attribute values), as stated in line 251. Our experiments are also conducted in this setting, as stated in lines 278.
>
> We will further clarify this in our reversion.
>
> > Theoretical part is not perfect.
>
> Thanks for your suggestion. We will correct them and carefully check our analysis in our revision.
>
> > Baseline is old.
>
> Comparison with [1] (CRIF) and [2] (UNICORN). Results (Tables R1 and R2) verify that CORE can perform well for querying attribute values, while RL methods excel at querying attributes because querying attribute values holds a huge action space which is not friendly for RL.
>
> We will the completed version of the results in our revision.
>
> [1] Learning to Infer User Implicit Preference in Conversational Recommendation. 2022.
>
> [2] Unified Conversational Recommendation Policy Learning via Graph-based Reinforcement Learning. 2021.
>
> > How to implement it online?
>
> We believe CORE can be easily implemented online, due to the following three reasons: (i) Our RS is offline-trained, therefore there are no training costs online. (ii) In practice, there are multiple stages in recommendation platform including matching, pre-ranking, ranking, re-ranking. CORE only needs to perform on re-ranking stage where there are only tens of items needed to be computed. (iii) In real-world cases, all the RS need to assign scores to each re-ranked item, CORE only brings extra computations on the expectation of certainty gain for each recommendation. We will clarify this in our revision.
>
> > Could you summarize the proposed learning method?
>
> We have summarized the main contributions of CORE in the main response above. We are very glad to further discuss if you have any confusion.
>
> > What are the theoretical foundations behind it?
>
> We apologize that we might not fully understand the review’s meaning in terms of “theoretical foundations” (if the reviewer could elaborate further, we would be happy to respond more precisely). An intuitive theoretical explanation of CORE is that RS function can be regarded as a prior from offline training, which is unlikely to perform well in the online setting, so we propose CORE to refine the prior online to better capture user needs. We propose Proposition 2 and Lemma 1 to analyze CORE’s performance on an ideal RS case and a bad RS case respectively.

---

> > ### Comment · Reviewer_7XeM · 2023-08-15
> >
> > “Implicit assumption on a unique item of a user and binary information on user preferences.“
> >
> > I mean that the proposed algorithm is not supported by any theoretical guarantees, and it is needed at least some logical motivation. As such, you implicitly assume that each time we seek for an item to recommend for a user, there is only one relevant, target item for that user. Otherwise, the sum of probabilities over items could not sum to 1. You consider prior, probabilities, information, entropy, etc. in your text, not just normalization. Anyways, theoretical assumptions underlying the motivation of the algorithm should be decoupled from details of practical application.

---

> > > ### Author Response · Authors · 2023-08-16
> > > **Response to Reviewer 7XeM**
> > >
> > > Thanks for your further clarifications. We have carefully checked Eq. 5, and we notice that the misleading part is the definition of $V^*$. We apologize for the misleading. We hope the following clarifications could address your concern.
> > >
> > > > The proposed algorithm is not supported by any theoretical guarantees, and it is needed at least some logical motivation. As such, you implicitly assume that each time we seek for an item to recommend for a user, there is only one relevant, target item for that user.
> > >
> > > Here, we provide two aspects to explain our Eq. 5.
> > >
> > > (i) Definition of $V^*$ should be the first item that the user clicks instead of the set of items matching the user's need. For example, assuming that both item A and item B can meet the user’s need; however, the user would firstly only click one item (either item A or B) and then jump to the page showing the detailed information of the clicked item (just like you are browsing the Amazon book store, there are multiple books you are interested in; however, you would firstly click one book you are most interested). In our paper, we define $V^*$ as the set of items matching user needs (e.g., items A and B in the above example), which is indeed not accurate. $V^*$ in this paper should be defined as the item first clicked by the user, because when a user clicks the item and then a user would be posted with another page, and in this case, we consider the session (i.e., the conversation) is finished (as introduced in Definition 1, lines 94-96). In other words, **CORE allows the user to favor multiple items at the same time, but we only focus on the very first clicked item**. In this regard, we can introduce Eq. 5 as
> > >
> > > $$
> > > Pr(a \text{ is } V^* ) = Pr(a \text{ is } V^* | a \in V_{k-1}) = \Psi_{RE}(a) / \text{SUM}(\Psi_{RE}(v)|v\in V_{k-1}),
> > > $$
> > >
> > > where we use the estimated scores of RS as the prior to estimate the probability of the user first clicking item $a$.
> > >
> > > (ii) If we treat each item equally (as there is no prior for all the candidate items), we can define $Pr(a \in V^*)$ as
> > >
> > > $$
> > > Pr(a \in  V^*) = Pr(a \in V^* | a \in V_{k-1}) = \text{SUM}(\Psi_{RE}(v)|v\in V^*) / \text{SUM}(\Psi_{RE}(v)|v\in V_{k-1}).
> > > $$
> > >
> > > If one compares the above equation with Eq. 5, one could conclude that we are using $\Psi_{RE}(a)$ to estimate $\text{SUM}(\Psi_{RE}(v)|v\in V^*)$, which means that we are implicitly assuming that there is only one item in $V^*$ . However, one assumption that the above equation holds is that all the items are treated equally. Namely, if the above equation holds, we can derive that $Pr(a \in  V^*) = Pr(b \in  V^*)$ holds for every possible pair of items $a$ and $b$, meaning that the probability of the user favoring each item is the same. This assumption only holds when there is no prior for all the candidate items. Here, we are using the estimated score as the prior to establish a weight factor in the above equation:
> > >
> > > $$
> > > Pr(a \in V^*) = Pr(a \in V^* | a \in V_{k-1}) = \text{SUM}(\Psi_{RE}(v)|v\in V^*) / \text{SUM}(\Psi_{RE}(v)|v\in V_{k-1}) \times  \Psi_{RE}(a) / \text{SUM}(\Psi_{RE}(v)|v\in V^*)  =  \Psi_{RE}(a) / \text{SUM}(\Psi_{RE}(v)|v\in V_{k-1}),
> > > $$
> > >
> > > which is our Eq. 5.
> > >
> > > We will make it clear in our revision.
> > >
> > > We appreciate the reviewer’s further clarification of the question and we hope the above explanations could address your concern. If there is any remaining concern or further questions, we are always willing to answer.
> > >
> > >
> > > > Anyways, theoretical assumptions underlying the motivation of the algorithm should be decoupled from details of practical application.
> > >
> > > In our experiments, there can be multiple target items (when they are posted to the user, the user would click) in one session, and we set the session as finished (a.k.a., is succeeded) if there is one target item posted to the user. This empirical setting is consistent with our theoretical definition in Definition 1, and the above explanation about Eq. 5.
> > >
> > > There is indeed one assumption about the use cases: there is at least one item matching the user's needs. We consider this assumption as a general one of RS, because even if there is no item that can exactly match the user’s need, any RS will still try to recommend the one closest to the user’s need. We will make this clear in our revision.

---

### Official Review · Reviewer_48VR · 2023-07-22

**Soundness:** 3 good
**Presentation:** 4 excellent
**Contribution:** 3 good
**Rating:** 6
**Confidence:** 4

**Summary:**

In this paper,  the authors propose a learning framework called CORE that can incorporate a conversational agent into any recommendation platform, to complementarily check estimated offline relevance scores in each online user session. Experiments results on comprehensive benchmark datasets show that CORE outperforms existing reinforcement learning-based and statistics-based approaches in querying items and attributes/attribute values.



**Strengths:**

1. The proposed offline-training and online-checking paradigm bridges a conversational agent and recommender systems via a unified uncertainty minimization framework. It's effective and efficient, and flexible to handle recommendations in both cold-start and hot-start settings.

2. Various large-scale benchmark datasets are used in experiments, and the results are convincing.




**Weaknesses:**


1. Are the comparison results in Tables 1, 2, 3 and 4 statistically significant? It'd be great if t-test results could be provided.

2. The central idea in this paper is to bring a conversational agent for recommendation systems. If possible, it'd be great to conduct an experiment on a real online platform to have user sessions to show the superiority of the proposed algorithm.

**Questions:**

1. What kind of data/features in each dataset are used to estimate the offline relevance scores in the first place?  Are the accuracy of the offline relevance scores critical to the performance of recommendations after online-checking framework via conversational agent?

---

> ### Author Rebuttal · Authors · 2023-08-10
>
> Thanks for your suggestions. Please also see the main response above.
>
> 1. It is better to provide t-test results for tables 1, 2, 3, and 4.
>
> Thanks for your suggestion. We will provide t-test results in our revision.
>
> 2. It is great to conduct CORE on online platforms.
>
> Thanks for your advice. We are thrilled about bringing CORE online to further verify its performance and use cases. However, as it requires deep incorporation with the industry, therefore, we are still looking for potential incorporation.
>
>
> 3. What kind of feature/data is used?
>
> We use all the features provided in the public datasets including both discrete and continuous features, e.g., seller ID, item ID, category ID, action ID (showing the type of user behaviors), and date (showing the time of user behaviors) in Taobao for training RS offline and generating estimated scores online. We will add these statistics in our revision.
>
> 4. Are the accuracy of offline recommendations critical to CORE?
>
> We evaluate the effect of RS performance on CORE from the following three perspectives:
> 1. Tables 1 to 4 (A1 to A4) provide the results of cold-start setting (on user side). In these cases, RS knows nothing about the user, and our results show that CORE also could outperform other baselines. Besides, Lemma 1 also provides a bound for the expected number of turns in the context of cold-start setting. These empirical results and theoretical analysis guarantee our performance when RS performs poorly (i.e., assigning the same scores to all the items).
> 2. Tables 1 to 4 (A1 to A4) also include different RS methods (with different RS performance). Results show that CORE can consistently benefit their RS compared to other baselines.
> 3. Table R4 shows the result of ablation studies. Different amounts of training data often lead to different RS performances, and CORE can stably outperform other baselines.
>
> We will make it clear in our revision.

---

> > ### Comment · Reviewer_48VR · 2023-08-12
> > **Thank the authors for the response**
> >
> > Thanks a lot for the detailed answers to my questions.

---

### Official Review · Reviewer_URPy · 2023-07-28

**Soundness:** 2 fair
**Presentation:** 3 good
**Contribution:** 3 good
**Rating:** 6
**Confidence:** 4

**Summary:**

The authors assert that there is a gap between traditional recommender systems trained on offline (historical) preference data and conversational assistants that aim to elicit user feedback about their *current* preferences. The authors assert that conversational assistant training relies on reinforcement learning-based frameworks and suffer from data insufficiency. They propose the CORE method for learning a conversational recommender system based on the idea of maximizing "certainty gain" by using a pre-trained recommender system as a scoring model for an online decision tree process. Experiments across a wide range of datasets demonstrates consistent improvements over reinforcement learning methods for learning conversational recommenders.

**Strengths:**

- The problem formulation makes sense and the authors organize the methodology in 3.1 and 3.2 in an intuitive manner. However, as mentioned in the weaknesses section there could be improvements in clarity.
- It is important that the authors extend attribute certainty gain computation to continuous attributes (Lemma 1) as this is a realistic setting for many attributes of interest in real-world situations (see prices, interest rates, or other continuous values).
- As a general extension of the above, the authors carefully consider multiple cases that bring the problem setting closer to real-world user interactions and preference considerations (neutral responses, attribute dependence). They extend the derivation of certainty gain across these situations.
- Experiments across a variety of datasets and base recommender systems demonstrate that CORE performs better in most cases compared to CRM and EAR baselines.

**Weaknesses:**

- The authors should clarify the reasoning for the statement on lines (135-136) that "if [the queried item is in the target set], the sesion is done and therefore the certainty gain is the summation of all relevance scores in V_{k-1}". Why is this the certainty gain? It is rooted in the definition in (1) that the certainty gain of a target item sets all items to "checked for certainty" but it is unclear what the basis is for this or whether it's a choice made to simplify the modeling.
- The authors omit discussion of a large area of conversational recommendation focusing on eliciting user feedback via critiques of surfaced attributes [including 1-4]. This area is directly related to the "attribute query" in the online-checking framework of CORE, and merits discussion in the literature review.
- The reference to ChatGPT-3.5-turbo seems gratuitous, as there is little mention of it in the main paper other than a statement on (254) that a pre-trained language model can be used as the Psi_{CO} component.
- The experimental analysis seems relatively sparse. It would be good to see a deeper analysis (case studies, for example, or human evaluations) in the main paper body. It would also help to see a discussion of the limitations, edge cases, or remaining challenges in the space and what other challenging experimental settings were considered.


References:
[1] Antognini, D. "Interacting with Explanations through Critiquing" (2021)
[2] Antognini, D. "Positive and Negative Critiquing for VAE-based Recommenders" (2022)
[3] Li, S. "Self-Supervised Bot Play for Conversational Recommendation with Rationales" (2022)
[4] Wu, G. "Deep language-based critiquing for recommender system" (2019)

**Questions:**

N/A

**Limitations:**

The authors do not really discuss limitations of the work in a substantive way. I would like to see a greater discussion of challenges and limitations.

---

> ### Author Rebuttal · Authors · 2023-08-10
>
> Thanks for your suggestions. Please also see the main responses above.
>
> > Why is this certainty gain from finding an item satisfying user needs in lines 135-136?
>
> As defined in Eq. (2), our objective is to find an item satisfying user needs, and then we assume that the session is finished because the user would click the item to jump to the item page. Therefore, as shown in Definition 1 (lines 95-96), the certainty gain from finding an item satisfying user needs should be the maximum value. Therefore, we set it as the summation of all the unchecked relevance scores. We will further clarify this in our revision.
>
> > More literature on focusing on eliciting user feedback via critiques of surfaced attributes.
>
> Thanks for your suggestion. We plan to add these papers to our revision and discuss the connections between them and our method.
>
> > Little mention ChatGPT-3.5.
>
> Thanks for pointing it out. Our conversational component has two parts: one is how to decide what items or attribute values to query (which corresponds to our online decision tree algorithm), and the other one is how to generate querying texts and analyze user responses. As stated in Section 3.3 and Appendix 4.2, ChatGPT-3.5 is used to scale CORE up to free-text real-world use cases, which indeed is an alternative to other Chatbots such as Llama 2. We will re-organize the main text and the appendix to better explain the utility of ChatGPT here.
>
> > Deeper analysis (e.g., case studies, for example, or human evaluations) is needed.
>
> Thanks for your suggestions. We provide a case study of CORE in Figure A2, and some examples of generated texts can be found in Appendix 4.2. We plan to include more examples and human evaluations of the generated conversations in the revision.
>
> > Further discussion of the limitations, edge cases, or remaining challenges.
>
> Thanks for your suggestion. Tables 1 to 4 (A1 to A4) show that CORE can well address cold-start problem on user side (where RS knows nothing about the user, and therefore offer equivalent treatment on all the items). But, when it comes to cold-start problems on item side (where RS knows nothing about the item), it is still an opening problem of how to use CORE to help RS to address this issue. We will include this edge case in our revision.

---

> > ### Comment · Reviewer_URPy · 2023-08-22
> >
> > Thanks to the authors for their detailed responses to my review & the others. The analysis and limitations remain an area of improvement for me regarding the paper, and my stance is still "weak accept".

---

### Author Rebuttal · Authors · 2023-08-10

We summarize our responses and the results of the suggested experiments here. We also respond to every specific concern of each reviewer as individual comments below.

To summarize our contributions: (i) CORE is a plug-and-play method that can enable any (offline) RS to recommend (i.e., query) items and attribute values online. For this purpose, we develop an offline-training and online-checking paradigm, where we regard RS as an offline estimator and an online conversational agent as an online checker. Our proposed online decision tree algorithm can leverage offline estimations from RS to decide what to query online. (ii) CORE can utilize pre-trained Chatbot APIs, since many powerful Chatbots are too heavy to be jointly optimized with RS, even some of them can be only accessed through APIs such as ChatGPT-3.5. (iii) RL often requires large numbers of training samples (known as data insufficiency) and relatively small action space (corresponding to querying attributes in our paper). Also, without sufficient training data, RL could not well generalize to open-world cases. Therefore, in many real-world RS cases, RL might not get its best performance; while CORE, a learning-free method (our online decision tree does not have any parameters to tune online), could achieve stable performance.

As one of the main concerns lies in the detailed comparisons between CORE and RL, below is a list of new experiments.
1. Comparison with [1] (CRIF) and [2] (UNICORN). Results (Tables R1 and R2) verify that CORE can perform well for querying attribute values, while RL methods excel at querying attributes because querying attribute values holds a huge action space which is not friendly for RL.
2. Comparisons with a new user simulation with a diversity metric. Results (Table R3) show that RL excels at giving diverse recommendations and therefore can well address the Matthew effect when there are multiple rounds in real-world cases, while CORE is good at giving specific recommendations.
3. Ablation studies with different amounts of training data. Results (Table R4) show that CORE is relatively stable with few training samples since RL often requires a huge number of training data.

Tables R1 to R4 can be found in uploaded PDF. Due to time limitations, we only conduct the experiments on LastFM and Amazon datasets, we plan to provide a completed form in our final version.

Another concern lies in whether CORE would be significantly affected by RS. Below is a list of old and new experiments answering this:
1. Tables 1 to 4 (A1 to A4) provide the results of cold-start setting (on user side). In these cases, RS knows nothing about the user, and our results show that CORE also could outperform other baselines. Besides, Lemma 1 also provides a bound for the expected number of turns in the context of cold-start setting. These empirical results and theoretical analysis guarantee our performance when RS performs poorly (i.e., assigning the same scores to all the items).
2. Tables 1 to 4 (A1 to A4) also include different RS methods (with different RS performance). Results show that CORE can consistently benefit their RS compared to other baselines.
3. Table R4 shows the result of ablation studies. Different amounts of training data often lead to different RS performances, and CORE can stably outperform other baselines.

[1] Learning to Infer User Implicit Preference in Conversational Recommendation. 2022.

[2] Unified Conversational Recommendation Policy Learning via Graph-based Reinforcement Learning. 2021.

---

> ### Author Response · Authors · 2023-08-21
> **Update of Rebuttal**
>
> We sincerely thank again all the reviewers for their constructive comments and suggestions. We are aware that comprehensive comparisons between CORE and RL methods are expected, and, as mentioned in our rebuttal, we have designed and conducted new experiments and revised versions addressing all the questions raised by the reviewers.
>
> Here, we want to note that we have already finished all the new experiments (on the same setting as Tables R1 to R4) on all the datasets including Amazon, LastFM, Yelp, Taobao, Tmall, Alipay, Douban Movie, and Douban Book datasets. Our results show the same trend as reported results in Amazon and LastFM datasets. For example, in the context of querying items and attribute values on the tabular dataset, CORE + FM achieves 3.14, 0.84, 3.20, and 0.99 in terms of T@3, S@3, T@5, and S@5 on the Yelp dataset, while UNICORN achieves 4.01, 0.38, 6.12, and 0.76. We will include these results in our revision.
>
> We hope that reviewers will champion our paper for acceptance.

---

### Decision · Program_Chairs · 2023-09-21

**Decision:**

Accept (poster)

**Comment:**

Although scores are ever-so-slightly borderline, they tend toward accepting the paper. The discussion was fairly active and rebuttal had a positive impact on scores.